# Benchmarking Tabular Foundation Models for Conditional Density Estimation in Regression

## Abstract

Conditional density estimation (CDE) – recovering the full conditional distribution of a response given tabular covariates – is essential in settings with heteroscedasticity, multimodality, or asymmetric uncertainty. Recent tabular foundation models, such as `TabPFN` and `TabICL`, naturally produce predictive distributions, but their effectiveness as general-purpose CDE methods has not been systematically evaluated – unlike their performance for point prediction, which is well studied. We benchmark three tabular foundation model variants against a diverse set of parametric, tree-based, and neural CDE baselines on 39 real-world datasets, across training sizes from 50 to 20,000, using six metrics covering density accuracy, calibration, and computation time. Across all sample sizes, foundation models achieve the best CDE loss, log-likelihood, and CRPS on the large majority of datasets tested. Calibration is competitive at small sample sizes but, for some metrics and datasets, lags behind task-specific neural baselines at larger sample sizes. However, we show that a post-hoc PIT recalibration step improves absolute calibration of the foundation models while leaving their accuracy advantage intact. In a photometric redshift case study using SDSS DR18, `TabPFN` exposed to 50,000 training galaxies outperforms all baselines trained on the full 500,000-galaxy dataset. Taken together, these results establish tabular foundation models as strong off-the-shelf conditional density estimators.

**Keywords:** conditional density estimation, tabular foundation models, benchmarking, TabPFN, TabICL, uncertainty quantification, regression

## 1 Introduction

Conditional density estimation (CDE) seeks to estimate the full distribution of a response variable given covariates, rather than only its conditional mean. In this paper, we focus on the standard univariate-response setting, where the goal is to estimate the full conditional density of a scalar response given tabular covariates. This is useful when uncertainty, asymmetry, or multimodality matter, with applications in areas such as photometric redshift estimation, risk analysis, simulation-based inference, and treatment-response modeling (Izbicki et al., 2014; 2017; Schmidt et al., 2020; Izbicki et al., 2017; Koenker, 2005; Hothorn et al., 2014; Izbicki et al., 2019; Cranmer et al., 2020; Dalmasso et al., 2020a; 2024; Cabezas et al., 2025a).

Over the years, CDE has been studied through a wide range of approaches, including classical nonparametric estimators (Rosenblatt, 1969; Hyndman et al., 1996; Izbicki & Lee, 2016), mixture density networks (Bishop, 1994), and flow-based models (Papamakarios et al., 2021). Recent tabular foundation models, such as `TabPFN` and `TabICL`, have shown strong performance for point-predictions in tabular supervised learning and naturally produce predictive distributions (Hollmann et al., 2025; Qu et al., 2025; McElfresh et al., 2023). This makes them plausible candidates for CDE. However, it remains unclear whether these distributional outputs are competitive with purpose-built CDE methods in terms of both density accuracy and calibration. Recent work has begun to evaluate tabular foundation models from a distributional perspective, but existing benchmark efforts still focus mostly on point prediction (Erickson et al., 2025; Landsgesell & Knoll, 2026).

In this paper, we present a broad empirical benchmark of tabular foundation models for CDE. We compare `TabPFN` and `TabICL` with a diverse set of classical and modern CDE baselines on thirty-nine real-world

datasets, across multiple sample sizes and using six evaluation metrics. We find that tabular foundation models are consistently competitive across a broad range of settings, most of the time surpassing purpose-built density estimators, although they sometimes do not excel at calibration.

To illustrate the CDE task and the differences among methods, Figure 1 shows estimated conditional densities for a synthetic process in which two Gaussian components have covariate-dependent means, variances, and mixture weights. Three test instances are shown, each exhibiting a different shapes (including bimodal ones), and columns correspond to increasing training-set sizes. Among the methods compared, `TabPFN-2.5` recovers the mixture structure at $n = 200$, whereas both `Flow-Spline` and `FlexCode-RF` require more data and still exhibit spurious peaks or roughness at moderate sample sizes. This example motivates our systematic evaluation: can tabular foundation models offer a strong out-of-the-box alternative to standard approaches?

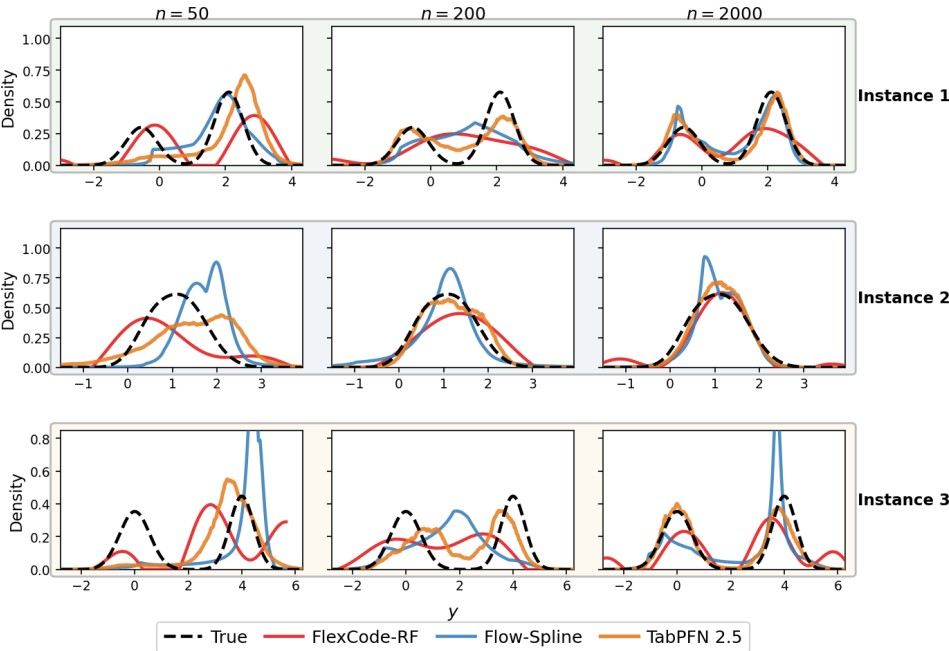

Figure 1: **Illustration of CDE on a two-component Gaussian-mixture synthetic DGP.** Each row corresponds to a different test instance whose true conditional density (dashed black) is a two-component Gaussian mixture with covariate-dependent means, variances, and weights. Columns show three training sizes ($n \in \{50, 200, 2{,}000\}$). `TabPFN-2.5` (orange) already captures the main mixture structure at $n = 200$, while `Flow-Spline` (blue) and `FlexCode-RF` (red) require considerably more data and still show spurious peaks or roughness. This is a controlled synthetic example designed to illustrate differences among methods; see Section 5 for real data analyses.

## 1.1   Relation to Other Work

**Conditional density estimation.** Conditional density estimation has a long history in statistics, going back at least to the nonparametric formulation of Rosenblatt (1969). Classical estimators include kernel-based, local-polynomial, and local-likelihood approaches (Hyndman et al., 1996; Fan et al., 1996; Bashtannyk & Hyndman, 2001; Hyndman & Yao, 2002; Efromovich, 2007). These methods established many of the core ideas in CDE, including kernel smoothing, local estimation, and bandwidth selection. More recent work has developed flexible machine-learning approaches to CDE. Neural methods include mixture density networks (Bishop, 1994), kernel mixture networks (Ambrogioni et al., 2017), and flow-based models built on normalizing flows (Dinh et al., 2014; 2017; Papamakarios et al., 2017; Huang et al., 2018; Durkan et al., 2019; Papamakarios et al., 2021). Another line is FlexCode (Izbicki & Lee, 2017), which casts CDE as a regres-

sion problem. Gaussian-process extensions have also been developed for heteroscedastic and non-Gaussian distributions (Le et al., 2005; Platanios & Chatzis, 2014; Dutordoir et al., 2018). A closely related literature studies probabilistic prediction through parametric families whose distributional parameters depend on covariates. This includes regression quantiles (Koenker & Bassett, 1978), GAMLSS (Rigby & Stasinopoulos, 2005), and distributional regression (Kneib et al., 2023; Klein, 2024). Related ideas also appear in tree-based and ensemble methods such as quantile regression forests (Meinshausen, 2006), transformation forests (Hothorn & Zeileis, 2021), and NGBoost (Duan et al., 2020), as well as in Bayesian approaches including BART-based density regression (Chipman et al., 2010; Orlandi et al., 2021; Li et al., 2023). Overviews of these and related methods are given by Dalmasso et al. (2020b), Kneib et al. (2023), and Klein (2024). Taken together, this motivates the families of baselines considered in our benchmark.

**Tabular foundation models.** Recent tabular foundation models build on the prior-data fitted network (PFN) paradigm, in which a transformer is meta-trained on many synthetic supervised tasks and then applied in context to a new dataset without gradient updates at test time (Müller et al., 2022; Nagler, 2023). Early work in this area focused mainly on point prediction for tabular classification and regression, typically evaluated by accuracy, AUC, RMSE, or $R^2$. In this line, Hollmann et al. (2025) introduced `TabPFN` for small- to medium-sized tables, while Qu et al. (2025) scaled in-context learning to much larger tables with `TabICL`. Benchmark efforts such as Erickson et al. (2025) likewise assess these models mainly through point-prediction metrics. At the same time, at least some original tabular foundation model papers already go beyond point prediction. In particular, Hollmann et al. (2025) explicitly presents density estimation as one of `TabPFN`'s foundation model abilities and shows proof-of-concept density-estimation experiments. Vetter et al. (2025) propose an extension of these density estimators to multivariate responses in the simulation-based inference setting, using an autoregressive factorization, and compare the resulting method with traditional SBI estimators. The broader official `TabPFN` ecosystem also exposes related extension-based capabilities, although the corresponding extensions are explicitly described as experimental and "less rigorously tested than the core `tabpfn` library" (Prior Labs, 2025). More recently, Qu et al. (2026) extend the `TabICL` line to both regression and classification, making distributional prediction a more direct use case. Thus, the main gap is not that tabular foundation models lack distributional outputs, but rather that their quality as *conditional density estimators* has only recently begun to be evaluated systematically. Indeed, Landsgesell & Knoll (2026) note that leading tabular foundation models already produce full predictive distributions, whereas major benchmarks still rely mostly on point-estimate metrics. Our work complements this emerging line by benchmarking tabular foundation models explicitly as CDE methods.

**Benchmarks for tabular learning.** Most benchmark papers in tabular machine learning focus on point prediction. For example, Grinsztajn et al. (2022) showed that tree-based methods remain highly competitive on standard tabular benchmarks. Subsequent efforts broadened the empirical picture in several directions: McElfresh et al. (2023) compared 19 algorithms; Gardner et al. (2023) studied robustness under distribution shift; Rubachev et al. (2025) introduced industry-style benchmarks with time-based splits and feature-engineering effects; Ye et al. (2024) evaluated tabular methods on more than 300 datasets in TALENT; Holzmüller et al. (2024) studied strong default baselines on large collections of regression and classification tasks; and Erickson et al. (2025) proposed a continuously maintained living benchmark. As a result, that literature has generally said more about point-prediction performance than about the quality of the full conditional distribution.

Recent concurrent work has further emphasized the need to evaluate tabular foundation models through distributional metrics rather than only point-prediction metrics. Landsgesell & Knoll (2026) evaluate the distributional outputs of RealTabPFN-v2.5 and TabICL-v2 on OpenML regression datasets using proper scoring rules, showing that leading tabular foundation models produce meaningful predictive distributions. Building on this direction, Landsgesell et al. (2026) introduce ScoringBench, an open benchmark for tabular regression that evaluates foundation models and other predictors under a broader suite of proper scoring rules, including CRPS, CRLS, interval score, energy score, and weighted CRPS.

Our work is complementary to this concurrent track. Rather than benchmarking probabilistic tabular regression in general, we focus specifically on conditional density estimation and compare tabular foundation models with purpose-built CDE baselines from five methodological families—parametric, quantile-based, regression-based, tree-based, and neural. We also expand the experimental scope to 39 datasets, sample sizes

up to 20,000 in the main benchmark and 500,000 in the SDSS case study, and six evaluation metrics including calibration diagnostics. Finally, we provide diagnostic analyses of failure modes, post-hoc recalibration experiments, and a scaling case study that isolates the sample-efficiency advantage of foundation models.

### 1.2 Novelty

Our main contributions are:

- **A broad benchmark of tabular foundation models for CDE:** We evaluate the conditional density outputs of `TabPFN` and `TabICL` against a diverse set of parametric and nonparametric baselines.

- **Large-scale experimental design:** The benchmark covers thirty-nine real-world datasets from OpenML and SDSS DR18, with sample sizes ranging from 50 to 20,000 and covariate dimensions from 5 to 563.

- **Multifaceted evaluation:** We evaluate all methods using six complementary metrics—CDE loss, log-likelihood, CRPS, PIT-based calibration, 90% coverage, and computation time—to assess accuracy, calibration, and efficiency jointly.

## 2 Background

### 2.1 Conditional Density Estimation

Let $(X, Y) \sim P_{XY}$, where $X \in \mathcal{X} \subseteq \mathbb{R}^d$ denotes the covariates and $Y \in \mathcal{Y} \subseteq \mathbb{R}$ a univariate response. Conditional density estimation seeks to estimate the full conditional distribution of $Y$ given $X = x$. When a conditional density exists, it is a function $f(y \mid x)$ such that

$$\mathbb{P}(Y \in A \mid X = x) = \int_A f(y \mid x) \, dy \tag{1}$$

for every measurable set $A \subseteq \mathcal{Y}$. Unlike mean or quantile regression, CDE aims to recover the entire shape of the predictive distribution, including heteroscedasticity, asymmetry, heavy tails, and multimodality. This is useful whenever downstream decisions depend on uncertainty summaries such as predictive intervals, tail probabilities, or quantiles. Throughout the paper, each method receives a training sample $\mathcal{D} = \{(x_i, y_i)\}_{i=1}^n$ and returns either a conditional density $\hat{f}(y \mid x)$, a conditional distribution function $\hat{F}(y \mid x)$, or a set of predictive quantiles from which a density can be derived.

## 3 Methods Compared

### 3.1 Tabular Foundation Models

We evaluate three tabular foundation model variants: `TabPFN-2.5`, `RealTabPFN-2.5`, and `TabICL-Quantiles` (Hollmann et al., 2025; Qu et al., 2025; 2026). All three models are used as pretrained in-context learners, without task-specific gradient updates. That is, no fine tuning is performed.

The two `TabPFN` variants output bar distributions over the response range. We convert these bar distributions into conditional densities as follows: for each bin, we divide the predicted probability mass by the bin width to obtain a density value at the bin center; we then interpolate these bin-center density values onto a fine evaluation grid of 200 equally spaced points using linear interpolation; finally, the interpolated density is renormalized to integrate to one. By contrast, `TabICL-Quantiles` outputs predictive quantiles. We convert these quantiles into a conditional distribution function by interpolation and then differentiate numerically to obtain a density. Full implementation details for these conversions are given in Appendix A.1.

### 3.2 Baselines

We compare tabular foundation models with a diverse set of classical and modern CDE baselines spanning five families:

**Parametric distributional regression.** We include six models that assume a specific parametric family for $Y \mid X = x$ and allow one or more distributional parameters to depend linearly on the covariates: a homoscedastic and a heteroscedastic Gaussian model (`LinearGauss-Homo` and `LinearGauss-Hetero`), a linear model with Student-$t$ residuals (`Student-t`), log-normal models with constant and covariate-dependent scale (`LogNormal-Homo` and `LogNormal-Hetero`), and a gamma GLM with log link (`Gamma-GLM`) (Rigby & Stasinopoulos, 2005; Kneib et al., 2023). Each is also included in a ridge-regularized variant (`-Ridge` suffix) (Hoerl & Kennard, 1970). Parametric methods via ML estimation are not run when the number of parameters is larger than the number of sample points.

**Quantile-based methods.** `Quantile-Tree` fits a gradient-boosted tree independently at 49 quantile levels and converts the resulting quantile function to a density by interpolation and numerical differentiation (Koenker & Bassett, 1978; Chen & Guestrin, 2016).

**Regression-based density estimation.** `FlexCode-RF` expresses the conditional density as a cosine basis expansion whose coefficients are estimated by random-forest regression (Izbicki & Lee, 2017).

**Tree-based probabilistic models.** `BART-Homo` and `BART-Hetero` use XBART to estimate the conditional mean (and, in the heteroscedastic variant, an input-dependent variance), inducing a Gaussian predictive density (Chipman et al., 2010; He et al., 2019).

**Neural density estimators.** `MDN` is a mixture density network with a tuned number of Gaussian components (Bishop, 1994). `Flow-Spline` is a conditional normalizing flow based on rational-quadratic splines (Durkan et al., 2019; Papamakarios et al., 2021). `CatMLP` discretizes the response into bins (in the same spirit as `TabPFN`) and fits a multilayer perceptron with a softmax output using cross-entropy loss.

Detailed implementation choices for each baseline are reported in Appendix A.2.

## 4 Experimental Setup

We evaluate three tabular foundation model variants and the diverse set of baselines on a variety of real-world regression tasks. Experiments vary the dataset and training-set size, and all methods are assessed using proper scoring rules, calibration diagnostics, and computation time. Unless otherwise noted, results are averaged over 5 independent repetitions.

### 4.1 Evaluation Metrics

We evaluate all methods using six complementary metrics covering density accuracy, calibration, sharpness, and computational cost. Let $\hat{f}(y \mid x)$ denote an estimated conditional density and let

$$\hat{F}(y \mid x) = \int_{-\infty}^{y} \hat{f}(t \mid x) \, dt \tag{2}$$

be the associated predictive distribution function. When a method outputs $\hat{F}$ or predictive quantiles directly, we evaluate the corresponding implied distribution.

Our main density-specific metric is the CDE loss (Schmidt et al., 2020),

$$L(\hat{f}) = \int \int \hat{f}(y \mid x)^2 \, dy \, dP_X(x) - 2 \, \mathbb{E}_{(X,Y)}[\hat{f}(Y \mid X)]. \tag{3}$$

This is a proper scoring rule, minimized in expectation by the true conditional density. Note that CDE loss values are typically negative on real data, with more negative values indicating better density estimates. We estimate it on held-out test data. We also report mean test log-likelihood (i.e., the cross-entropy) and the

continuous ranked probability score (CRPS) (Gneiting & Raftery, 2007), which are also proper scoring rules for CDE.

We also evaluate calibration of the CDEs. A CDE is calibrated if events predicted to occur with a given probability occur with approximately that same frequency under the data-generating distribution. In the conditional-distribution setting, a standard notion of probabilistic calibration is that, when $(X, Y)$ is drawn from the test distribution, the random variable $U = \hat{F}(Y \mid X)$ should be approximately uniformly distributed on $[0, 1]$. We summarize marginal probabilistic calibration using the Kolmogorov–Smirnov (KS) statistic of the probability integral transform (PIT) values. We use PIT KS because it is a scale-free marginal calibration diagnostic and is a standard choice in several comparisons of CDE methods (Dalmasso et al., 2020b; Schmidt et al., 2020; Zhao et al., 2021). We also report empirical coverage of 90% predictive intervals as a complementary calibration diagnostic, so our calibration assessment does not rely solely on PIT KS.

Finally, we also evaluate total fit-and-predict wall-clock time.

For each metric, we also test whether each foundation model significantly outperforms every parametric and nonparametric competitor on a given dataset. Specifically, for foundation method $F$ and competitor $C$, we compute a one-sided Welch $t$-test using the conservative variance estimate $\widehat{\mathrm{SE}}^2_{\mathrm{diff}} = \widehat{\mathrm{SE}}^2_F + \widehat{\mathrm{SE}}^2_C$, which ignores the positive correlation induced by shared test sets and therefore yields a conservative test (i.e., it is harder to reject the null). We apply Holm–Bonferroni correction across all comparisons for a given foundation method and dataset, and use $\alpha = 0.1$.

### 4.2 Protocol and Hyperparameter Tuning

Each experiment is repeated over 5 independent train/test splits (except for $n = 50$, in which case we used 50 random splits due to small testing size). In each split, 25% of the available data is held out as the test set. We report the mean of each metric across repetitions together with standard errors. All tabular foundation models are used with their default pretrained settings, and no fine-tuning is performed.

For the baselines, hyperparameters are selected on the training data only. Ridge-regularized parametric models choose the regularization strength by cross-validation; `FlexCode-RF` selects the number of basis functions by 5-fold cross-validation; and the remaining baselines (`MDN`, `Flow-Spline`, `Quantile-Tree`, `BART-Homo`, `BART-Hetero`, and `CatMLP`) are tuned by random search with 3-fold cross-validation drawing using the CDE loss as the selection criterion. After tuning, the selected configuration is refit on the full training set. Complete search spaces and implementation details are reported in Appendix A.3.

### 4.3 Datasets

We include 39 regression datasets from OpenML (Bischl et al., 2025) and SDSS DR18 (SDSS Collaboration, 2023), spanning domains such as housing, robotics, computer hardware, transportation, molecular prediction, retail, music, bioinformatics, energy, and photometric redshift estimation. The covariate dimension $d$ (after one-hot encoding of categorical features) ranges from 5 to 563 across the real-world datasets. For each dataset, we evaluate methods on nested subsamples of size $n \in \{50, 500, 1{,}000, 5{,}000, 10{,}000, 20{,}000\}$ whenever available. Complete dataset details are reported in Appendix B.

## 5 Results

Figures 2–4 present CDE loss raw-value heatmaps across real-world datasets at $n \in \{50, 1{,}000, 20{,}000\}$, with datasets sorted by covariate dimension $d$ (results at other sample sizes are reported in Appendix C). Across all sample sizes, foundation models dominate the top positions in average rank.

At $n = 50$ (Figure 2), `RealTabPFN-2.5` achieves an average rank of 2.2 across the 39 datasets, with `TabPFN-2.5` (2.3) close behind. The third-best average rank, however, is held by the non-foundation `Student-t-Ridge` (6.6), which narrowly outperforms `TabICL-Quantiles` (7.0); thus at this smallest sample size two of the three foundation model variants occupy the top two positions, but `TabICL-Quantiles` lags behind the best parametric baseline. Among the remaining non-foundation competitors, `FlexCode-RF` (7.7)

and `LogNormal-Homo-Ridge` (7.8) follow. Despite `TabICL-Quantiles`' weaker ranking at $n = 50$, a foundation model achieves the best CDE loss on 32 out of 39 datasets (82%), indicating broad dominance even at this extremely small sample size. Significance tests confirm this pattern: a $*$ in the heatmap marks cases where a foundation model significantly outperforms all non-foundation competitors.

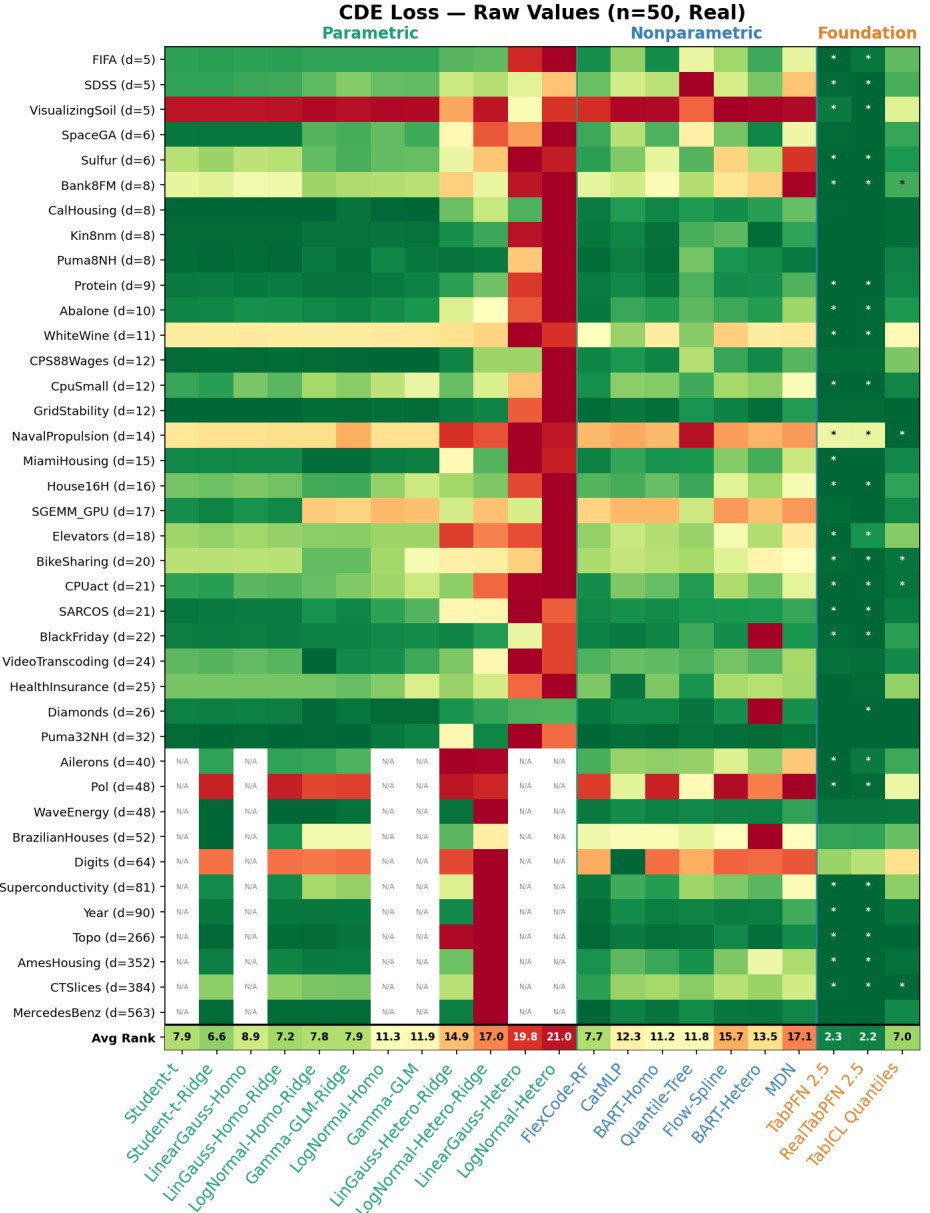

Figure 2: **CDE loss across real-world datasets at** $n = 50$**.** Per-dataset raw CDE loss values (lower/greener is better). Datasets are sorted by covariate dimension $d$. A $*$ marks foundation models that significantly outperform all parametric and nonparametric competitors on that dataset. The two `TabPFN` variants (orange) achieve the top two average ranks (bottom row), while `TabICL-Quantiles` (7.0) is narrowly outranked by `Student-t-Ridge` (6.6). Even at this extremely small sample size, a foundation model achieves the best CDE loss on 82% of datasets.

At $n = 1{,}000$ (Figure 3), all three foundation models occupy the top three average ranks: `RealTabPFN-2.5` leads at 2.1, followed by `TabPFN-2.5` (2.3) and `TabICL-Quantiles` (2.6). The gap to the best non-foundation

baselines is substantial: `Flow-Spline` (8.4), `FlexCode-RF` (8.7), and `Quantile-Tree` and `CatMLP` (both 9.1) trail by a wide margin. Foundation models achieve the best CDE loss on 36 out of 39 datasets (92%) at this sample size, with significant wins ($*$) on many datasets.

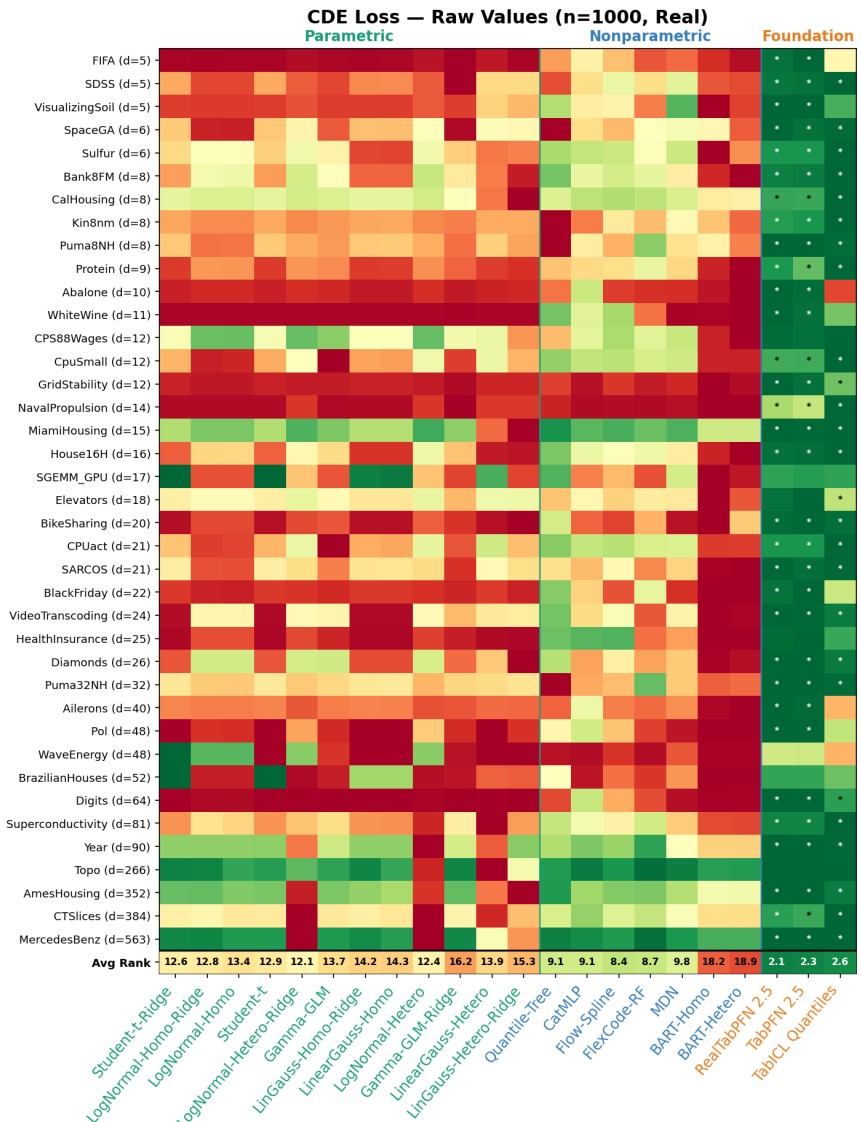

Figure 3: **CDE loss across real-world datasets at** $n = 1,000$**.** Per-dataset raw CDE loss values (lower/greener is better). Datasets are sorted by $d$. A $*$ marks foundation models that significantly outperform all competitors on that dataset. All three foundation models occupy the top three average ranks; `RealTabPFN-2.5` leads at 2.1. Foundation models achieve the best CDE loss on 92% of datasets.

At $n = 20,000$ (Figure 4), which covers the 16 datasets large enough for this subsample, all three foundation models again hold the top three average ranks: `TabPFN-2.5` (2.7), `TabICL-Quantiles` (2.9), and `RealTabPFN-2.5` (3.4). The best non-foundation competitor is `Flow-Spline` (4.4), with `CatMLP` and MDN tied at 5.8. The gap between foundation models and the best nonparametric baselines narrows as $n$ grows (compare the 6+ rank-point gap at $n = 1,000$ with the $\approx$1–2 rank-point gap at $n = 20,000$), consistent with a sample-efficiency advantage that diminishes as non-foundation methods receive more training data. Foundation models achieve the best CDE loss on 12 out of 16 datasets (75%) at this sample size. However, a notable practical limitation emerged at this sample size: on the CTSlices dataset ($d = 384$), both `TabPFN`

variants encountered out-of-memory errors and could not produce predictions (marked with $\times$ in Figure 4) on a NVIDIA GeForce RTX 5070 Ti GPU with 16303 MiB of VRAM. Only `TabICL-Quantiles` ran successfully on this dataset, achieving rank 1. This indicates that the combination of large sample size and high covariate dimension can exceed the memory capacity of current `TabPFN` implementations, a constraint that users should be aware of in high-dimensional settings.

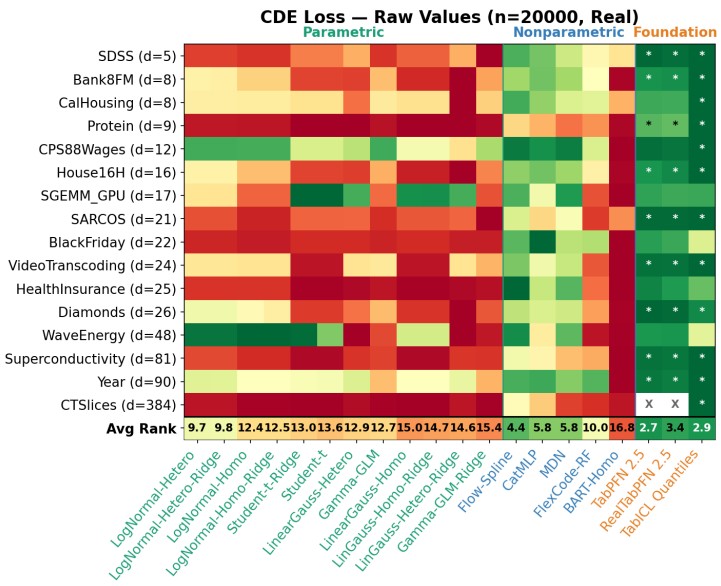

Figure 4: **CDE loss across real-world datasets at** $n = 20,000$**.** Per-dataset raw CDE loss values (lower/greener is better). Datasets are sorted by $d$. A $*$ marks foundation models that significantly outperform all competitors on that dataset; $\times$ marks foundation models that encountered out-of-memory errors. All three foundation models occupy the top three average ranks. On CTSlices, both `TabPFN` variants ran out of memory at this sample size.

The ranking pattern under log-likelihood and CRPS (Appendix C.1) is broadly consistent with the CDE loss results, though with some differences for `TabICL-Quantiles`. Under log-likelihood, the two `TabPFN` variants lead at every sample size (e.g., average ranks of 1.9 at $n = 1,000$), but `TabICL-Quantiles` ranks further behind (3.8 at $n = 1,000$; 8.2 at $n = 50$; 4.3 at $n = 20,000$). At $n = 20,000$, `Flow-Spline` (3.9) surpasses `TabICL-Quantiles` under log-likelihood, so that only two of the three foundation models remain in the top three for this metric. Under CRPS, all three foundation models consistently occupy the top three ranks at every sample size, and `TabICL-Quantiles` performs particularly well, achieving average ranks of 2.1 at $n = 1,000$ and 1.7 at $n = 20,000$—the best overall among all methods. A likely explanation is that `TabICL` outputs predictive quantiles, which must be converted to densities via interpolation and numerical differentiation. CRPS can be computed directly from the quantile function and is thus unaffected by this conversion, whereas CDE loss and log-likelihood evaluate the density pointwise and are sensitive to the roughness it introduces. This suggests that `TabICL-Quantiles`' predictive distributions are high quality, but that the density conversion penalizes density-specific metrics.

Calibration results based on the PIT KS statistic (Appendix C.1) paint a more nuanced picture. Foundation models achieve among the best calibration ranks at small sample sizes (e.g., `TabPFN` variants rank 4.4–4.6 at $n = 50$), and remain in the upper half of methods at $n = 1,000$. However, at $n \geq 5,000$, their calibration ranks deteriorate to mid-table (ranks 7–9 for the `TabPFN` variants), while `MDN` typically achieves the best calibration. This suggests that foundation models' in-context learning mechanism, while highly effective for density accuracy, does not scale as well for calibration as end-to-end training on larger datasets. The 90% empirical coverage results (Appendix C) show a similar trend. This confirms that strong performance on proper scoring rules does not automatically guarantee the best calibration across all settings. We examine whether this gap can be closed by post-hoc recalibration in Section 6. This sample-size dependence is not

uniform across methods: for example, `MDN` remains among the best-calibrated methods at larger sample sizes, whereas the foundation models improve more strongly on density accuracy than on calibration.

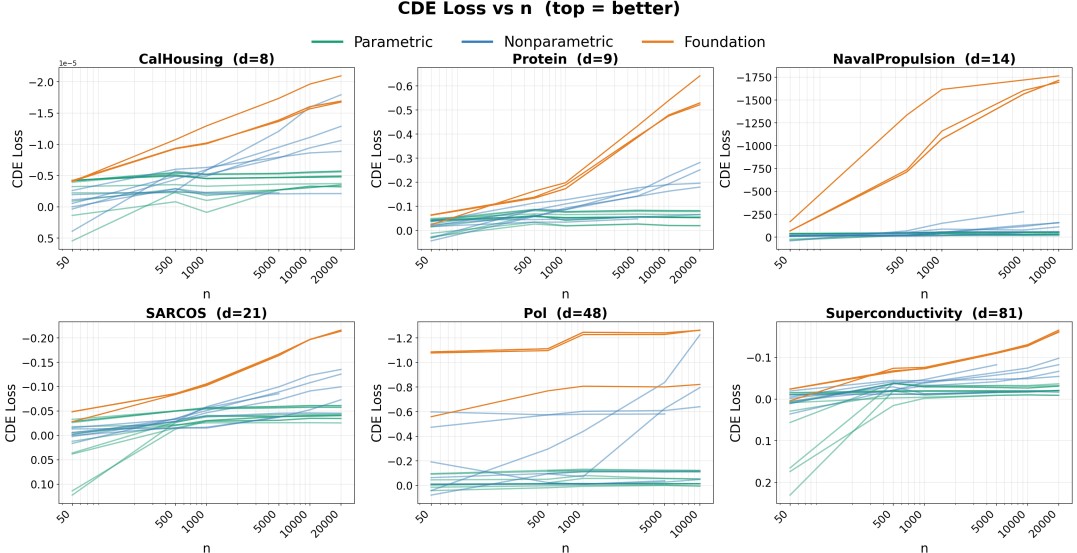

Figure 5: **CDE loss vs. sample size for selected real-world datasets.** Each panel shows one dataset, arranged by increasing $d$. Foundation models (orange, bold) are contrasted against parametric (green, faded) and nonparametric (blue, faded) baselines; the $y$-axis is oriented so that higher values are better. Foundation models are already competitive at $n = 50$ and often exhibit the steepest improvement with growing $n$. Parametric baselines plateau early. Nonparametric methods show greater variability, occasionally matching foundation models in higher-dimensional settings but without consistency.

Figure 5 displays CDE loss as a function of sample size $n$ across six real-world datasets, arranged by increasing covariate dimension, with methods grouped into parametric, nonparametric, and foundation-model families. Foundation models are already competitive at $n = 50$, and their advantage generally widens as $n$ grows— most dramatically in NavalPropulsion ($d = 14$), where they far surpass all other methods at large $n$. The foundation-model advantage persists in higher dimensions: at $d = 81$ (Superconductivity) these models remain the best performers across all sample sizes. At $d = 48$ (Pol), the picture is more mixed, with individual nonparametric methods occasionally matching foundation models at intermediate sample sizes, but no single nonparametric estimator consistently doing so across datasets. Parametric methods are tightly clustered and rarely reach the top regardless of sample size.

Figure 6 highlights counterexamples where foundation models lose to specialized methods. On Digits ($n = 50$, $d = 64$), where responses take only a few discrete values, `CatMLP` outperforms `TabPFN-2.5` by placing probability mass exactly on the support points, though this gap disappears by $n = 500$, suggesting a small-sample issue rather than a fundamental limitation. On VideoTranscoding ($n = 50$, $d = 24$), a simple `LogNormal-Homo-Ridge` model beats `RealTabPFN-2.5` because its strong parametric bias yields better estimates with little data, especially in the right tail; again, the advantage vanishes as $n$ grows. On BlackFriday ($n = 20,000$, $d = 22$), however, `CatMLP` also outperforms `TabPFN-2.5`, and the gap persists even at the largest sample size, indicating a structural disadvantage for the foundation model on quasi-discrete outcomes. These examples suggest that foundation models can be outperformed when the response has structure that is naturally captured by a specialized model—such as discrete or quasi-discrete support, or a well-matched parametric family whose inductive bias doubles as regularization—though in most such cases the advantage is confined to small sample sizes. Notice however that these cases are the exception rather than the rule: across all 39 datasets, only 3–7 (depending on sample size) see a non-foundation method rank first.

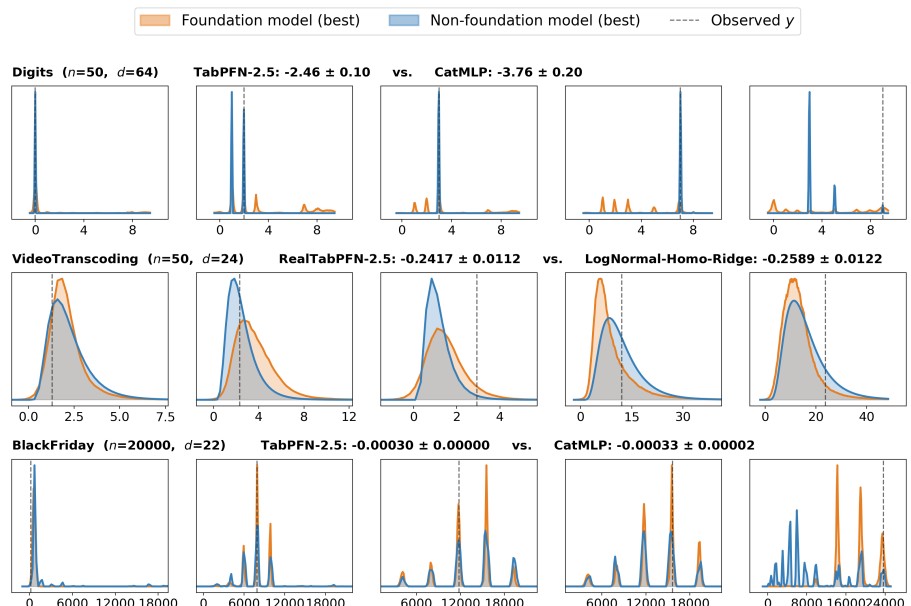

Figure 6: **Diagnostic examples of foundation-model failures on real data.** Each row shows a task where the best foundation model (orange) is beaten in CDE loss by the best non-foundation method (blue); five test cases are shown per row, with dashed lines marking observed outcomes. On Digits, `CatMLP` captures the discrete support more precisely than `TabPFN-2.5`. On VideoTranscoding, `LogNormal-Homo-Ridge` produces sharper tail densities. On BlackFriday, `CatMLP` better captures the quasi-discrete response structure—a gap that persists even at the largest available sample size.

## 6 Post-hoc Recalibration

The benchmark in Section 5 shows that, although foundation models dominate on the proper scoring rules, their calibration can lag behind the best task-specific baselines at larger sample sizes. A natural question is whether this gap can be reduced by a simple post-hoc recalibration step. Since recalibration can be applied to any predictive distribution, we also study whether the same procedure benefits foundation models and baselines to the same degree. We use distribution-free probabilistic recalibration (Kuleshov et al., 2018). Concretely, for a fitted conditional CDF $\hat{F}(\cdot \mid x)$, recall from Section 4.1 that probabilistic calibration corresponds to the PIT values $U = \hat{F}(Y \mid X)$ being uniformly distributed on $[0, 1]$. Recalibration learns a monotone map $\hat{R} : [0, 1] \to [0, 1]$, estimated as the empirical CDF of held-out PIT values, and replaces $\hat{F}$ by $\tilde{F} = \hat{R} \circ \hat{F}$. Because we also evaluate density-based losses, we recalibrate the full density rather than only the CDF. By the chain rule, $\tilde{f}(y \mid x) = \hat{R}'(\hat{F}(y \mid x)) \hat{f}(y \mid x)$. We fit $\hat{R}$ as a smooth monotone PCHIP interpolant of the empirical PIT CDF on a coarse probability grid, pinned at $(0, 0)$ and $(1, 1)$, so that $\hat{R}'$ is well defined and $\tilde{f}$ integrates to one. The same procedure is applied to every method.

Figure 7 reports mean PIT KS and mean $|\text{coverage} - 0.90|$ before and after recalibration. Both statistics improve for nearly every method at every sample size: recalibration lowers the PIT KS statistic on 92–100% of method–dataset cells, the foundation models' PIT KS falls by roughly 40%, and their 90%-coverage error is more than halved at $n = 20,000$.

The improvement is largest for the methods that were worst calibrated before recalibration. Across all metrics, the initially miscalibrated baselines move most, while the already well-calibrated foundation models and `MDN` move less. Thus, recalibration acts as an equalizer that compresses calibration differences.

Moreover, recalibrating all methods does not remove the foundation models' lead on the proper scoring rules. Table 1 reports the fraction of datasets on which a foundation model is the single best method, before and after recalibration. On CDE loss, log-likelihood, and CRPS this fraction is essentially unchanged, and slightly higher at large $n$ for CDE loss. On the calibration metrics, a foundation model is rarely the single

best, both before and after recalibration, with `MDN` often attaining the best value. Figure 8 corroborates this at the level of average ranks: the foundation models hold the best ranks on the proper scoring rules both before and after recalibration, while on the calibration metrics recalibration compresses the field so that their *relative* rank can slip even as their *absolute* calibration improves.

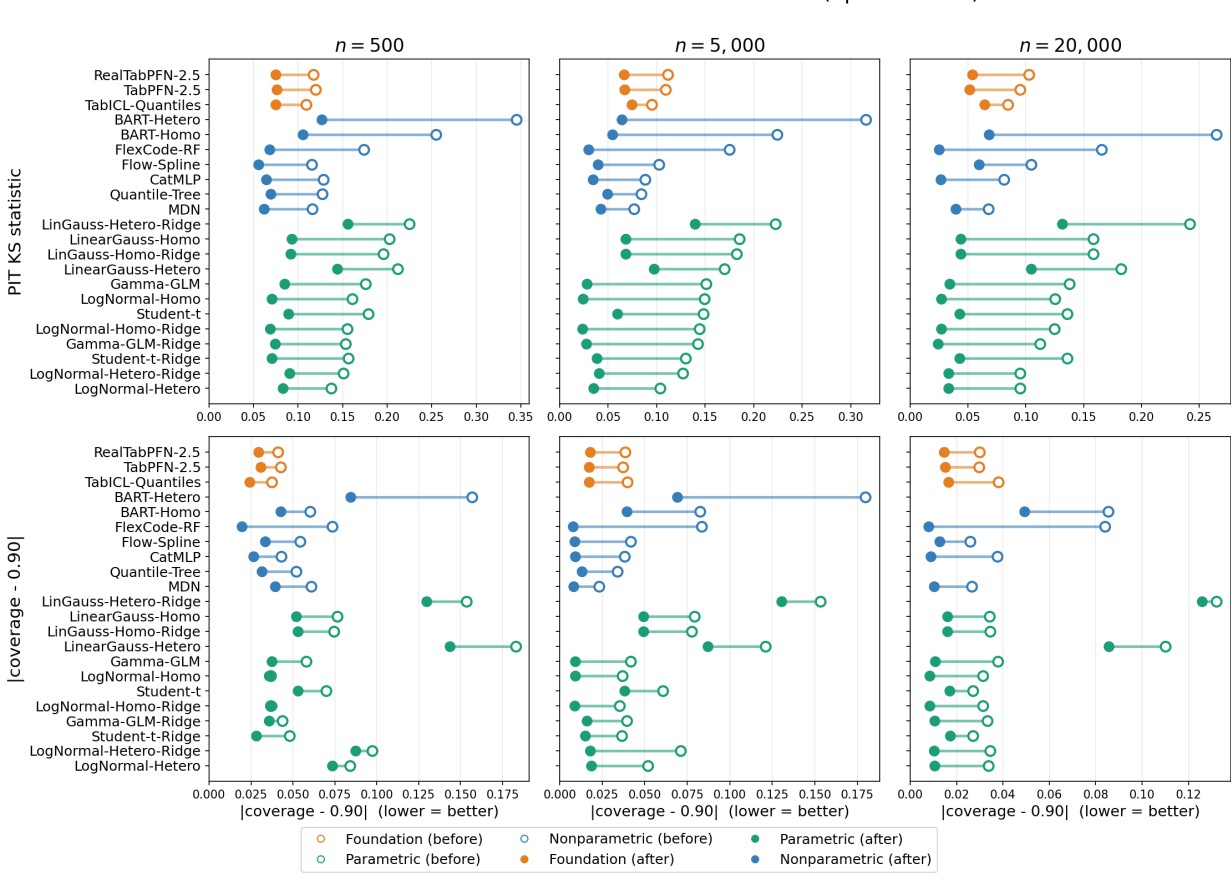

Figure 7: **Calibration before and after post-hoc PIT recalibration.** Mean PIT KS statistic (top) and mean $|\text{coverage} - 0.90|$ (bottom) across datasets (lower is better), per method at each $n$. Each segment runs from *before* (open marker) to *after* (filled marker) recalibration. Both metrics improve for nearly every method at every $n$; the largest gains accrue to the initially worst-calibrated baselines.

Table 1: **A foundation model stays best after recalibration.** Fraction of datasets on which the best foundation model beats the best non-foundation method, shown as *before→after* recalibration. The accuracy advantage is preserved for CDE loss, log-likelihood, and CRPS; on the calibration metrics, a single baseline typically attains the best value both before and after.

| Metric | $n = 500$ | $n = 1,000$ | $n = 5,000$ | $n = 10,000$ | $n = 20,000$ |
|---|---|---|---|---|---|
| CDE loss | 92→90% | 92→92% | 88→94% | 80→84% | 75→81% |
| Log-likelihood | 90→92% | 87→82% | 88→88% | 80→76% | 81→81% |
| CRPS | 97→95% | 97→95% | 97→97% | 96→100% | 94→94% |
| PIT KS | 23→36% | 41→13% | 15→9% | 36→20% | 19→19% |
| $|\text{cov} - 0.90|$ | 5→0% | 5→5% | 3→15% | 20→12% | 12→12% |

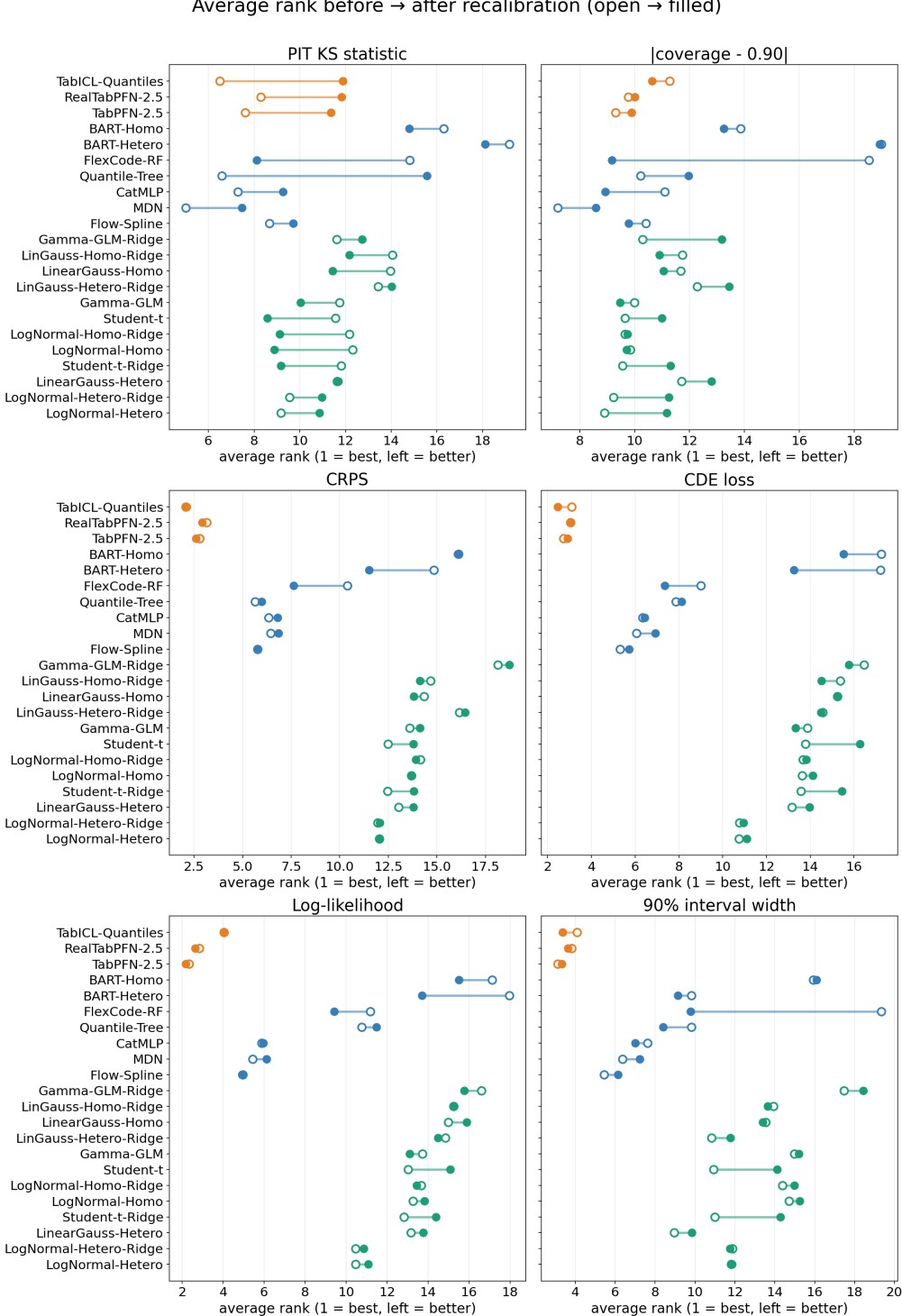

Figure 8: **Average rank (1 = best) of each method before (open markers) and after (filled markers) post-hoc recalibration**; foundation models in orange, parametric baselines in green, nonparametric baselines in blue. On the proper scoring rules (CRPS, CDE loss, log-likelihood) the foundation rows remain at the best ranks, confirming that recalibrating every method does not erode their accuracy advantage. On the calibration metrics (PIT KS, coverage) recalibration compresses the field—the initially worst-calibrated baselines improve most—so the foundation models' *relative* calibration rank can slip even as their *absolute* calibration improves substantially (Figure 7).

# 7 Case Study: Photometric Redshift Estimation with SDSS

The benchmark results in Section 5 evaluate all methods at sample sizes up to $n = 20{,}000$, which aligns with the context-length limits of current tabular foundation models. An important question is how these models behave in a large-scale setting, and in particular how their performance at moderate $n$ compares to classical methods that can exploit larger training sets. To address this, we conduct a scaling study on photometric redshift estimation using the Sloan Digital Sky Survey (SDSS) Data Release 18 (Almeida et al., 2023).

The SDSS dataset was assembled via a SQL query to the DR18 SkyServer (SDSS Collaboration, 2023), selecting 500,000 galaxies with reliable spectroscopic redshifts (`zWarning` $= 0$, $0 < z < 1$) and clean $ugriz$ photometry in the range $14 \leq r \leq 22$; for each galaxy we retrieved the five $ugriz$ model magnitudes and their photometric uncertainties as covariates and the spectroscopic redshift $z$ as the response. Estimating the full conditional density $f(z \mid \text{photometry})$ is a standard problem in astrophysics, as downstream cosmological analyses depend on accurate predictive distributions rather than point estimates (Schmidt et al., 2020; Izbicki et al., 2017; Freeman et al., 2017). This makes SDSS a natural testbed for evaluating conditional density estimators at scale.

We construct a sequence of nested subsamples $n \in \{500, 1\text{k}, 10\text{k}, 50\text{k}, 100\text{k}, 250\text{k}, 500\text{k}\}$, where each smaller dataset is a strict subset of the next. All experiments are repeated over 5 independent train/test splits. Tabular foundation models are constrained by context length: **TabPFN** variants are evaluated up to $n = 50{,}000$ (on GPU), while **TabICL-Quantiles** is evaluated up to $n = 10{,}000$ under default configurations. All other methods are run at all sample sizes up to 500,000 whenever the total fit-and-predict time remains below one hour. In addition to the methods considered in the main benchmark, we include **FlexZBoost**, a sharpening-enhanced variant of **FlexCode** based on gradient-boosted regressors (Dalmasso et al., 2020b), developed specifically for the photometric redshift estimation task (Schmidt et al., 2020).

Figure 9 reports CDE loss, CRPS, and log-likelihood as a function of training set size. The dominant pattern is the strong sample efficiency of tabular foundation models. At $n = 50{,}000$—the largest dataset they can process—the **TabPFN** variants achieve a CDE loss of approximately $-10.8$, surpassing all baselines trained on the full $n = 500{,}000$ dataset. The same qualitative behavior is observed for CRPS and log-likelihood. This comparison isolates an important trade-off: foundation models operate under strong context constraints, yet extract substantially more predictive signal per training example. In this setting, conditioning on only 10% of the available data yields better conditional density estimates than conventional methods trained on the entire dataset. Calibration diagnostics tell a similar story. PIT-based KS statistics and empirical coverage (Appendix C.2) show that foundation models at $n = 50{,}000$ remain competitive with the best baselines, indicating that their gains in accuracy are not achieved at the expense of probabilistic calibration. Finally, Figure 10 shows that tabular foundational models have low computation time when compared to competitive nonparametric models. For instance, **TabPFN-2.5** processes 50,000 galaxies in approximately 39 seconds on GPU, while **Flow-Spline**—the strongest non-foundation competitor at $n = 500{,}000$—requires roughly 19 minutes on CPU and still achieves a weaker CDE loss. The foundation model thus achieves superior CDE performance in a fraction of both the data and the wall-clock time, though this comparison involves different hardware (GPU vs. CPU).

From a practical perspective, these results suggest that tabular foundation models can substantially reduce data requirements in workflows where labeled data is expensive. In this case, achieving comparable CDE performance with classical methods would require roughly an order of magnitude more data.

# 8 Final Remarks

We presented a broad empirical benchmark of tabular foundation models for conditional density estimation in tabular regression. To summarize the main outcomes of the benchmark, the principal takeaways are as follows:

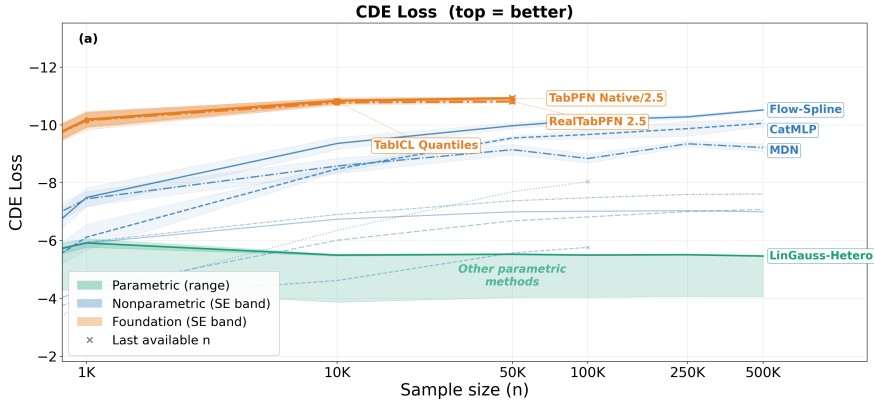

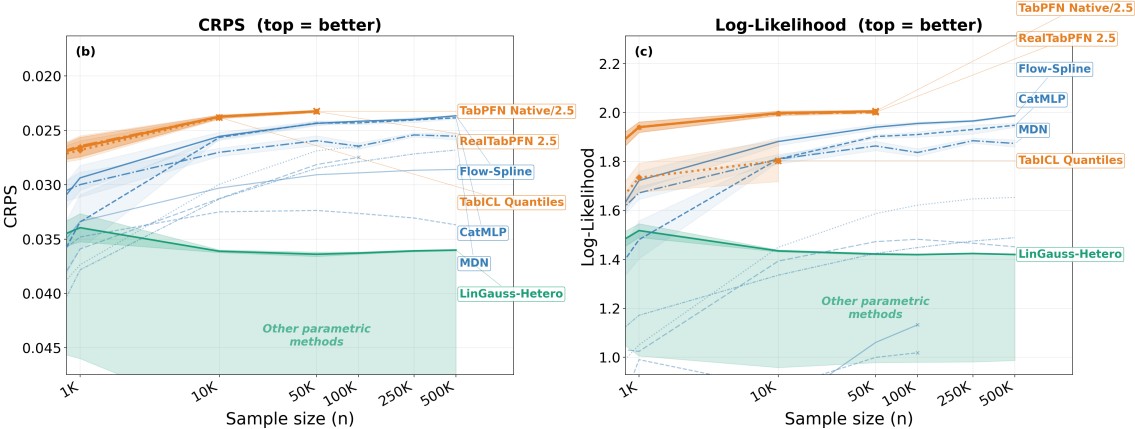

Figure 9: **CDE loss (panel a), CRPS (panel b), and log-likelihood (panel c) vs. training sample size on the SDSS photometric redshift dataset.** All $y$-axes are oriented so that top = better (CDE loss and CRPS are inverted; log-likelihood is shown in natural order). Parametric methods (green): the shaded band spans the full range across all parametric models; the best-performing parametric model (`LinearGauss-Hetero`) is highlighted as a solid line. Ridge-regularized variants are omitted from the band as they are uniformly worse and would distort the $y$-axis scale. Nonparametric methods (blue): each method is shown as a separate line; shaded $\pm 1$ SE bands are displayed only for the three best-performing nonparametric methods to avoid visual clutter. A terminal $\times$ marker indicates the last sample size at which a method was evaluated. Key finding: across all three metrics, `TabPFN` trained on 50k galaxies outperforms all methods trained on 500k galaxies, demonstrating a ten-fold data efficiency advantage.

- **Tabular foundation models are strong off-the-shelf CDE methods.** Across 39 real-world datasets, training sizes from 50 to 20,000, and six complementary evaluation metrics, the foundation-model variants were the strongest overall performers for density estimation. In particular, the `TabPFN` variants achieved the best results on CDE loss, log-likelihood, and CRPS on most datasets, while `TabICL-Quantiles` also ranked among the top-performing methods on most metrics.

- **Their main empirical advantage is sample efficiency.** Foundation models were already competitive at very small sample sizes and often maintained their advantage as $n$ increased. The SDSS photometric redshift case study illustrates this pattern most clearly: `TabPFN` trained on 50,000 galaxies outperformed all competing methods trained on the full 500,000-example dataset, indicating that these models can extract substantially more distributional information per labeled example.

- **Calibration is competitive but not uniformly best.** The calibration results are more nuanced than the proper-scoring-rule results. Foundation models performed especially well at small sample sizes, but

at larger $n$ they were sometimes overtaken by task-specific neural baselines such as `MDN`. Post-hoc PIT recalibration substantially improved their absolute calibration while preserving their advantage on CDE loss, log-likelihood, and CRPS.

- **The methods remain constrained by context length and memory.** The `TabPFN` variants are limited by the amount of data that can be processed in context and typically require GPU resources at larger sample sizes. In our experiments, the combination of large $n$ and high covariate dimension led to out-of-memory failures on CTSlices, showing that direct application to very large or high-dimensional datasets may require subsampling, batching, or other approximation strategies.

- **Specialized models can still win when their inductive bias matches the response structure.** Although such cases were the exception rather than the rule, diagnostic examples showed that non-foundation methods can outperform foundation models for discrete or quasi-discrete responses, or when a simple parametric family provides a particularly well-matched regularizing assumption.

- **Practical recommendation: calibrate, and check computational feasibility.** For practitioners, the results suggest using tabular foundation models as strong default conditional density estimators when the dataset fits within the model's context and memory limits. In applied workflows, we recommend evaluating calibration diagnostics and applying post-hoc PIT recalibration when calibrated uncertainty is important. Users should also account for the joint effect of sample size $n$, covariate dimension $d$, available memory, and total fit-and-predict time, since these factors can determine whether a foundation model is practical on a given problem. This is an active research area, and the computational trade-offs should be revisited as newer tabular foundation models become faster and more memory efficient.

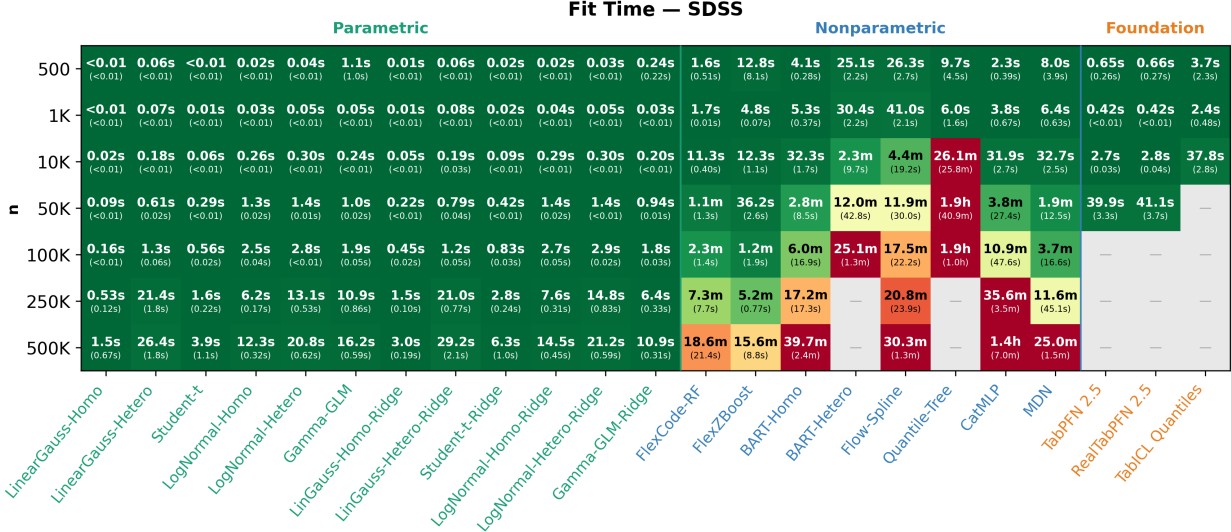

Figure 10: **Fit time (training + prediction) for each method across sample sizes $n$ on the SDSS photometric redshift dataset.** Each cell reports the mean fit time (top) and standard error in parentheses. Cells are colour-coded within each method column from green (fastest) to red (slowest); grey cells indicate that the method was not run at that sample size due to computational limits. Fit time includes both training and inference on the test set. Key finding: Foundation models are substantially faster than neural baselines (`Flow-Spline`, `MDN`) at the same sample sizes, while parametric baselines are the cheapest overall but sacrifice density accuracy.

**Limitations.** Despite these strong results, several caveats apply. First, all `TabPFN` variants are subject to context-length constraints (here capped at $n = 50,000$) and typically require GPU for such sample sizes. Moreover, the combination of large $n$ and high covariate dimension can trigger out-of-memory errors well below the context-length limit: in our experiments, both `TabPFN` variants failed on CTSlices ($d = 384$)

already at $n = 20,000$. These constraints limit direct applicability to very large or high-dimensional datasets without subsampling or batching strategies. Second, while foundation models are broadly competitive on marginal calibration, we observed that for some datasets their calibration is worse than some baselines, although we showed this gap can be reduced via post-hoc recalibration. Third, our benchmark is restricted to the univariate-response setting; it is unclear how these conclusions extend to multivariate or structured outputs. Finally, as illustrated by the diagnostic counterexamples in Figure 6, when the response has structure that is naturally captured by a specialized model—such as discrete support or a well-matched parametric family—the task-specific inductive bias can be more informative than the general-purpose prior acquired during pre-training. In such regimes, current foundation models' pre-training priors, while broadly effective, do not yet fully subsume the knowledge encoded in well-chosen structural assumptions about the response.

**Future directions.** This work opens several avenues for further investigation. Combining tabular foundation models with post-hoc covariate-dependent recalibration (such as Cabezas et al. 2024; Dey et al. 2025; Cabezas et al. 2025b; Cucuzzella et al. 2026) may improve even further calibration gaps observed on some datasets. Moreover, extending the evaluation to multivariate conditional density estimation, structured outputs, and downstream decision-making tasks that depend on the full predictive distribution is a natural next step.

We hope this benchmark helps shift the evaluation of tabular learning beyond point prediction and toward a fuller assessment of distributional quality. Code to reproduce all experiments, as well as tables with all metrics and standard errors, is available at `https://anonymous.4open.science/r/comp-7260/`.

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

# A   Implementation Details

## A.1   From Foundation-Model Outputs to Densities

**TabPFN-2.5.** This variant uses the explicit v2.5 default regression checkpoint from the **TabPFN** release (Hollmann et al., 2025). In regression mode, it outputs a bar distribution: a discrete probability mass over bins that partition the response range. We convert this bar distribution to a conditional density as follows: for each bin, the probability mass is divided by the bin width to obtain a density value at the bin center; these bin-center density values are then linearly interpolated onto a fine evaluation grid of 200 equally spaced points spanning the training range (plus a 5% margin on each side); finally, the interpolated density is clamped to be non-negative and renormalized to integrate to one. This procedure yields a smooth approximation to the underlying piecewise-constant bar density rather than evaluating the exact step function.

**RealTabPFN-2.5.** This variant uses the `RealTabPFN` regression checkpoint from the **TabPFN-2.5** release (Hollmann et al., 2025). The prediction interface is the same as for **TabPFN-2.5**, but the model is additionally fine-tuned on curated real-world tabular datasets. As with the other **TabPFN** variants, we use its native bar distribution as the starting point for the CDE output, following the same interpolation procedure described above.

**TabICL-Quantiles.** **TabICL** also follows an in-context learning paradigm for tabular prediction, but uses a different architecture and pretraining strategy (Qu et al., 2025; 2026). We use an ensemble of four estimators. Its regression variant outputs predictive quantiles. We extract 199 quantiles at levels $\alpha \in \{0.005, 0.010, \ldots, 0.995\}$, enforce monotonicity by sorting, and interpolate the resulting quantile function onto a regular grid to obtain the CDF. The conditional density is then computed as the numerical gradient of the interpolated CDF, clamped to be non-negative and renormalized.

### A.2 Baseline Implementations

#### A.2.1 Parametric baselines

These methods assume a parametric family for $Y \mid X = x$ and allow one or more distributional parameters to depend on the covariates.

**LinearGauss-Homo.** A Gaussian linear model with mean linear in $x$ and constant variance: $Y \mid X = x \sim \mathcal{N}(\beta^\top x, \sigma^2)$, where $\hat{\sigma}^2$ is estimated from training residuals with a degrees-of-freedom correction (Rigby & Stasinopoulos, 2005; Kneib et al., 2023).

**LinearGauss-Hetero.** A heteroscedastic Gaussian model in which both the mean and the log-variance depend on $x$: $Y \mid X = x \sim \mathcal{N}(x^\top\beta, \exp(x^\top\gamma))$ (Rigby & Stasinopoulos, 2005; Kneib et al., 2023). The parameters $\beta$ and $\gamma$ are estimated *jointly* by maximizing the Gaussian log-likelihood

$$\ell(\beta, \gamma) = -\tfrac{1}{2} \sum_{i=1}^{n} \big[x_i^\top\gamma + (y_i - x_i^\top\beta)^2 \exp(-x_i^\top\gamma)\big],$$

via L-BFGS-B (with analytic gradient), initialized from two-step OLS estimates of $\beta$ and a least-squares fit of log-squared residuals for $\gamma$. The predictive density is $\hat{f}(y \mid x) = \mathcal{N}\big(y; x^\top\hat{\beta}, \exp(x^\top\hat{\gamma})\big)$.

**Student-t.** A linear model with Student-$t$ conditional distribution (Rigby & Stasinopoulos, 2005; Kneib et al., 2023; Klein, 2024). The conditional mean is estimated by linear regression, $\hat{\mu}(x) = x^\top\hat{\beta}$. The degrees of freedom $\hat{\nu}$ and scale $\hat{\sigma}$ are then estimated by profile maximum likelihood on the training residuals, optimizing $\nu$ over $[2.01, 200]$ and setting $\hat{\sigma} = \sqrt{\bar{\varepsilon^2}(\hat{\nu} - 2)/\hat{\nu}}$ when $\hat{\nu} > 2$.

**LogNormal-Homo and LogNormal-Hetero.** These models assume a log-normal conditional distribution with linear structure in the log-scale mean (Rigby & Stasinopoulos, 2005; Kneib et al., 2023). When the response is not strictly positive, all training values are shifted by a constant $c = -\min(y_{\text{train}}) + 0.01 \cdot \text{range}(y_{\text{train}})$ so that the shifted response $\tilde{Y} = Y + c > 0$. The model is then $\log\tilde{Y} \mid X = x \sim \mathcal{N}(x^\top\beta, \sigma^2)$, with either constant $\sigma$ (-Homo) or input-dependent $\log\sigma^2(x) = x^\top\gamma$ estimated jointly with the mean via L-BFGS-B (-Hetero), analogously to **LinearGauss-Hetero**. The density is evaluated on the original scale and renormalized.

**Gamma-GLM.** A gamma generalized linear model with a log link for the conditional mean (Rigby & Stasinopoulos, 2005; Kneib et al., 2023; Klein, 2024). As with the log-normal models, when the response is not strictly positive the training values are shifted by $c = -\min(y_{\text{train}}) + 0.01 \cdot \text{range}(y_{\text{train}})$. On the shifted data, $\log\hat{\mu}(x) = x^\top\hat{\beta}$ is estimated by regressing $\log\tilde{y}$ on $x$, and a constant shape parameter $\hat{a}$ is estimated from the variance of the log-scale residuals as $\hat{a} = 1/\text{Var}(\log\tilde{y} - x^\top\hat{\beta})$. The predictive density is $\hat{f}(\tilde{y} \mid x) = \text{Gamma}(\tilde{y}; a = \hat{a}, \text{scale} = \hat{\mu}(x)/\hat{a})$, evaluated on the original scale and renormalized.

For each parametric model above, we also include a ridge-regularized variant (-Ridge suffix), where the regression coefficients are estimated by ridge regression with the regularization strength selected by leave-one-out cross-validation over 20 log-spaced values in $[10^{-4}, 10^4]$. For the heteroscedastic variants, the same ridge penalty is applied to both $\beta$ and $\gamma$ in the joint likelihood.

#### A.2.2 Nonparametric and flexible baselines

**FlexCode-RF.** FlexCode represents the conditional density through a cosine orthonormal basis expansion on the normalized response: $\hat{f}(z \mid x) = \sum_{i=0}^{I-1} \hat{\beta}_i(x)\varphi_i(z)$, where $\varphi_0 \equiv 1$ and $\varphi_i(z) = \sqrt{2}\cos(i\pi z)$ for $i \geq 1$ (Izbicki & Lee, 2017). Each coefficient function $\hat{\beta}_i(x)$ is estimated by regressing $\varphi_i(Y)$ on $X$ using a random forest with 100 trees and maximum depth 8. The number of basis functions is selected by cross-validation, and the estimated density is projected to be non-negative and renormalized.

**FlexZBoost.** FlexZBoost is a variant of FlexCode that replaces the random forest regression engine with XGBoost (100 trees, maximum depth 4, learning rate 0.1) and adds a post-hoc sharpening step (Dalmasso

et al., 2020b). After fitting the basis expansion, the estimated density is raised to a power $\alpha > 0$ and renormalized; the sharpening parameter $\alpha$ is tuned by 5-fold cross-validation minimizing the CDE loss, over a grid of 16 values in $[0.5, 2.0]$. This method has been shown to perform well on photometric redshift estimation tasks (Schmidt et al., 2020).

**BART-Homo.** This method uses XBART to model the conditional mean and pairs it with a constant residual variance estimated from the posterior mean of the XBART $\sigma$ draws, averaged over post-burn-in sweeps, inducing a Gaussian predictive density (Chipman et al., 2010; He et al., 2019).

**BART-Hetero.** A two-stage XBART-based approach: one XBART model estimates the conditional mean and a second XBART model is fit to the log-squared residuals to estimate the input-dependent variance, yielding a Gaussian conditional density with heteroscedastic variance (Chipman et al., 2010; He et al., 2019).

**Quantile-Tree.** A gradient-boosted tree model (XGBoost with the quantile error objective) fit independently at 49 quantile levels $\alpha \in \{0.02, 0.04, \ldots, 0.98\}$. The fitted quantiles are sorted to enforce monotonicity, linearly interpolated to obtain the CDF, and numerically differentiated to obtain the density, which is then clamped to be non-negative and renormalized (Koenker & Bassett, 1978; Meinshausen, 2006; Chen & Guestrin, 2016).

**Flow-Spline.** A conditional neural spline flow (Durkan et al., 2019; Papamakarios et al., 2021). The model learns an invertible transformation from a standard Gaussian base distribution to the conditional distribution of $Y \mid X = x$, parameterized as a composition of rational-quadratic spline layers. Each layer's spline parameters are produced by a two-hidden-layer MLP conditioned on $x$. Training minimizes the negative log-likelihood with Adam in mini-batches (batch size 512), with gradient clipping (max norm 5.0) and early stopping on a 10% validation split (patience 12).

**MDN.** A mixture density network with a tuned number of Gaussian mixture components (selected from $\{2, 3, 5\}$ by cross-validation) (Bishop, 1994). The architecture consists of one hidden layer with ReLU activation mapping the input to the mixture weights, component means, and component log-standard deviations. Training minimizes the negative mixture log-likelihood with Adam, with early stopping based on a 10% held-out validation split (patience of 30 epochs).

**CatMLP.** A categorical MLP that discretizes the response variable into $n_{\text{bins}}$ equal-width bins and fits a two-hidden-layer MLP (with ReLU activations) to predict the bin probabilities via a softmax output and a cross-entropy training objective. The predicted bin probabilities are converted to a density by dividing by the bin width, then linearly interpolated onto the 200-point evaluation grid and renormalized. Training uses Adam with early stopping on a 10% validation split (patience 30). Hyperparameters (number of bins, hidden units, learning rate, and training epochs) are tuned by random search.

### A.3 Hyperparameter Search

The foundation models `TabPFN-2.5`, `RealTabPFN-2.5`, and `TabICL-Quantiles`) are used without any hyperparameter tuning.

For the parametric baselines, the main tuning choice is whether to use ordinary least squares or ridge regression. In the ridge variants, the regularization strength is selected from 20 log-spaced candidates in $[10^{-4}, 10^4]$ via leave-one-out cross-validation.

For `FlexCode-RF` and `FlexZBoost`, the number of cosine basis functions $I$ is selected by 5-fold cross-validation on the training set, minimizing the CDE loss, with $I_{\max} = \min\bigl(30, \max(15, \lfloor\sqrt{n}\rfloor)\bigr)$. For `FlexZBoost`, the sharpening exponent $\alpha$ is additionally selected by 5-fold cross-validation over 16 values in $[0.5, 2.0]$.

For the remaining baselines (`MDN`, `Flow-Spline`, `Quantile-Tree`, `BART-Homo`, `BART-Hetero`, and `CatMLP`), hyperparameters are tuned by random search with 3-fold cross-validation, drawing 8 configurations and selecting the configuration that minimizes the mean CDE loss across folds.

Table 2: **Random-search spaces for tuned baselines.**

| Method | Search space |
|---|---|
| `MDN` | number of components $\in \{2, 3, 5\}$; hidden units $\in \{16, 32, 64\}$; learning rate $\in \{0.005, 0.01, 0.02\}$; epochs $\in \{300, 500, 800\}$ |
| `Flow-Spline` | spline bins $\in \{4, 8, 12\}$; layers $\in \{2, 3, 4\}$; hidden units $\in \{32, 64, 128\}$; learning rate $\in \{10^{-3}, 2 \times 10^{-3}, 5 \times 10^{-3}\}$; weight decay $\in \{10^{-6}, 10^{-5}, 10^{-4}\}$ |
| `Quantile-Tree` | number of boosting rounds $\in \{50, 100, 200\}$; max depth $\in \{3, 4, 6\}$; learning rate $\in \{0.05, 0.1, 0.2\}$ |
| `BART-Homo`, `BART-Hetero` | number of trees $\in \{20, 30, 50\}$; number of sweeps $\in \{40, 60, 80\}$ |
| `CatMLP` | number of bins $\in \{30, 50, 100\}$; hidden units $\in \{32, 64, 128\}$; learning rate $\in \{0.005, 0.01, 0.02\}$; epochs $\in \{300, 500, 800\}$ |

After tuning, the selected configuration is refit on the full training set and used for final prediction on the corresponding test split.

# B    Dataset Details

Table 3 lists the 39 real-world regression datasets used in the benchmark, together with their source (Vanschoren et al., 2013; SDSS Collaboration, 2023), maximum available sample size, and application domain. The covariate dimension $d$ reported is the number of features after one-hot encoding of categorical variables. Datasets are included at all sample sizes $n \in \{50, 500, 1{,}000, 5{,}000, 10{,}000, 20{,}000\}$ for which the dataset has at least $n$ observations.

Table 3: **Real-world datasets.**

| Dataset | Source | Max $n$ | Domain |
|---|---|---|---|
| SpaceGA | OpenML | 3,107 | Spatial analysis |
| Elevators | OpenML | 16,599 | Control systems |
| Kin8nm | OpenML | 8,192 | Robotics |
| Puma8NH | OpenML | 8,192 | Robotics |
| Bank8FM | OpenML | 22,784 | Finance |
| CpuSmall | OpenML | 8,192 | Computer hardware |
| CPUact | OpenML | 8,192 | Computer hardware |
| CalHousing | OpenML | 20,640 | Housing |
| Diamonds | OpenML | 53,940 | Gemology |
| Abalone | OpenML | 4,177 | Marine biology |
| Ailerons | OpenML | 13,750 | Control systems |
| BikeSharing | OpenML | 17,379 | Transportation |
| AmesHousing | OpenML | 2,930 | Housing |
| Digits | OpenML | 5,620 | Digit recognition |
| House16H | OpenML | 22,784 | Housing |
| Sulfur | OpenML | 10,081 | Chemistry |
| BrazilianHouses | OpenML | 10,692 | Housing |
| Pol | OpenML | 15,000 | Politics |
| MercedesBenz | OpenML | 4,209 | Manufacturing |
| Protein | OpenML | 45,730 | Bioinformatics |
| VisualizingSoil | OpenML | 8,641 | Soil science |
| Year | OpenML | 515,345 | Music |
| SGEMM_GPU | OpenML | 241,600 | GPU performance |
| BlackFriday | OpenML | 166,821 | Retail |
| Superconductivity | OpenML | 21,263 | Physics |
| WaveEnergy | OpenML | 72,000 | Energy |
| VideoTranscoding | OpenML | 68,784 | Computing |
| SARCOS | OpenML | 44,484 | Robotics |
| NavalPropulsion | OpenML | 11,934 | Engineering |
| GridStability | OpenML | 10,000 | Energy systems |
| MiamiHousing | OpenML | 13,932 | Housing |
| HealthInsurance | OpenML | 22,272 | Healthcare |
| CPS88Wages | OpenML | 28,155 | Economics |
| WhiteWine | OpenML | 4,898 | Food science |
| FIFA | OpenML | 18,063 | Sports |
| Puma32NH | OpenML | 8,192 | Robotics |
| CTSlices | OpenML | 53,500 | Medical imaging |
| Topo | OpenML | 8,885 | Topology |
| SDSS | SDSS DR18 | 500,000 | Astrophysics |

## C  Additional Results

This section displays the heatmaps of all metrics for all experiments.

### C.1  Real Datasets Experiments

We start with the results for the real data experiments.

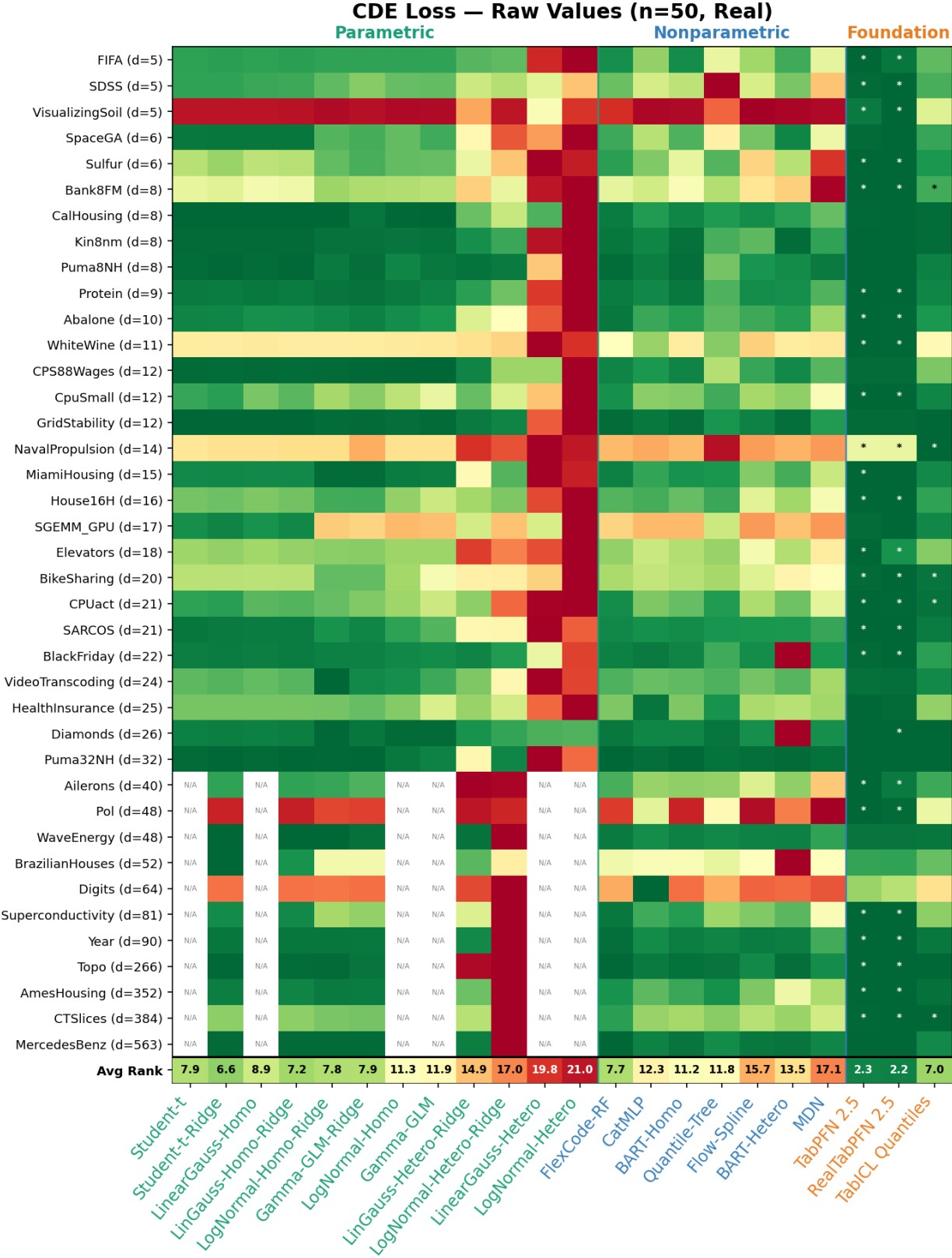

Figure 11: Raw CDE Loss – $n = 50$, real data.

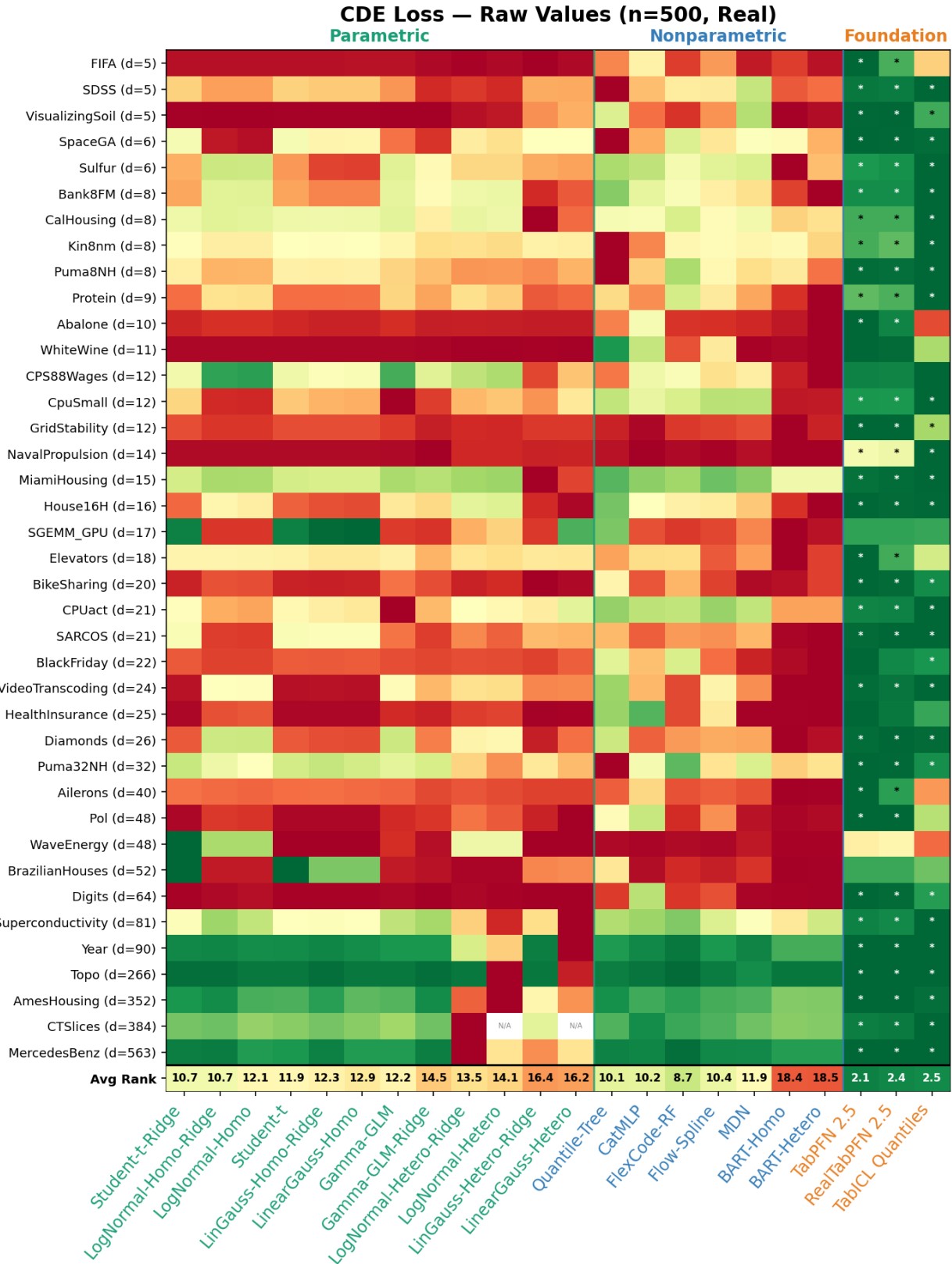

Figure 12: Raw CDE Loss – $n = 500$, real data.

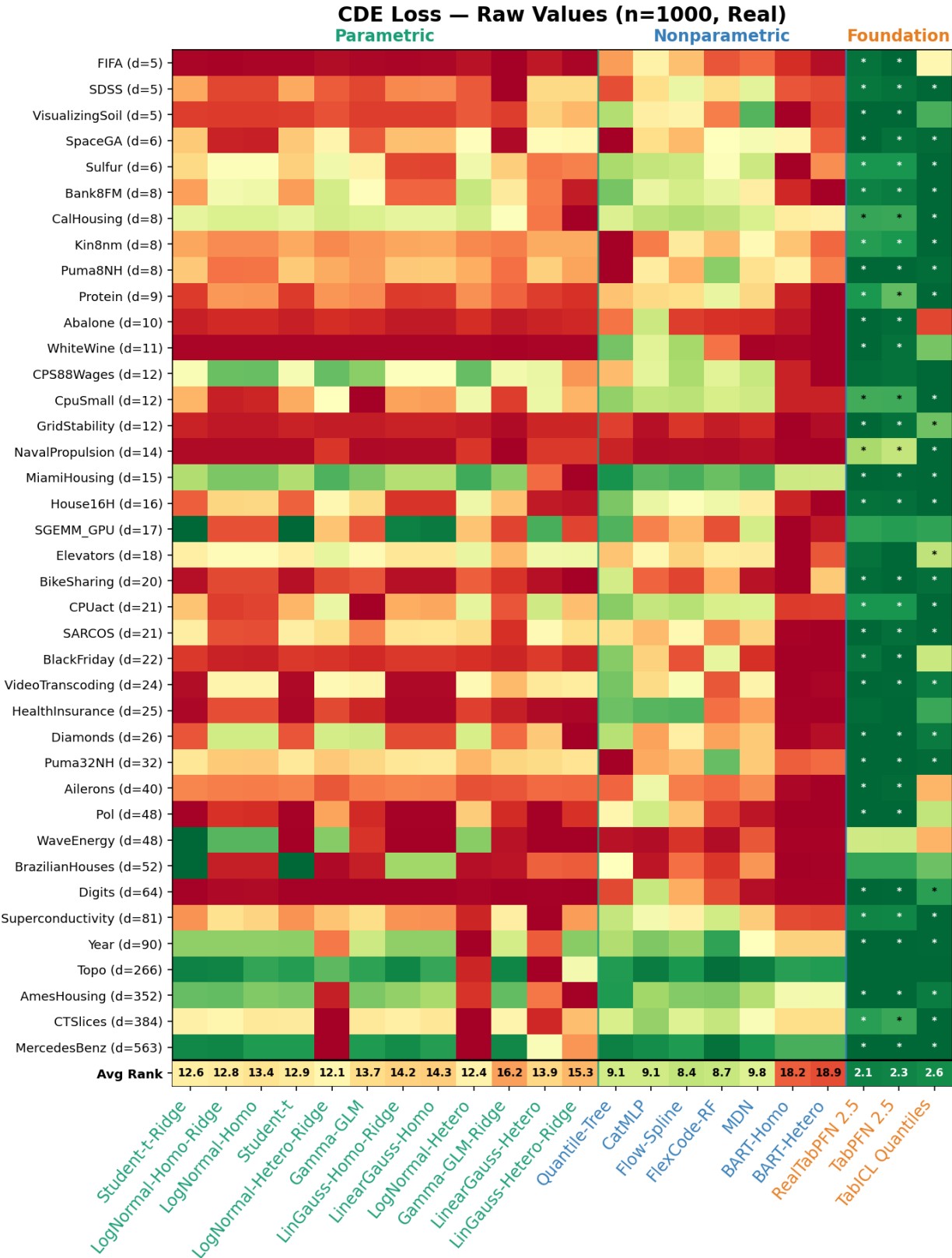

Figure 13: Raw CDE Loss − $n = 1000$, real data.

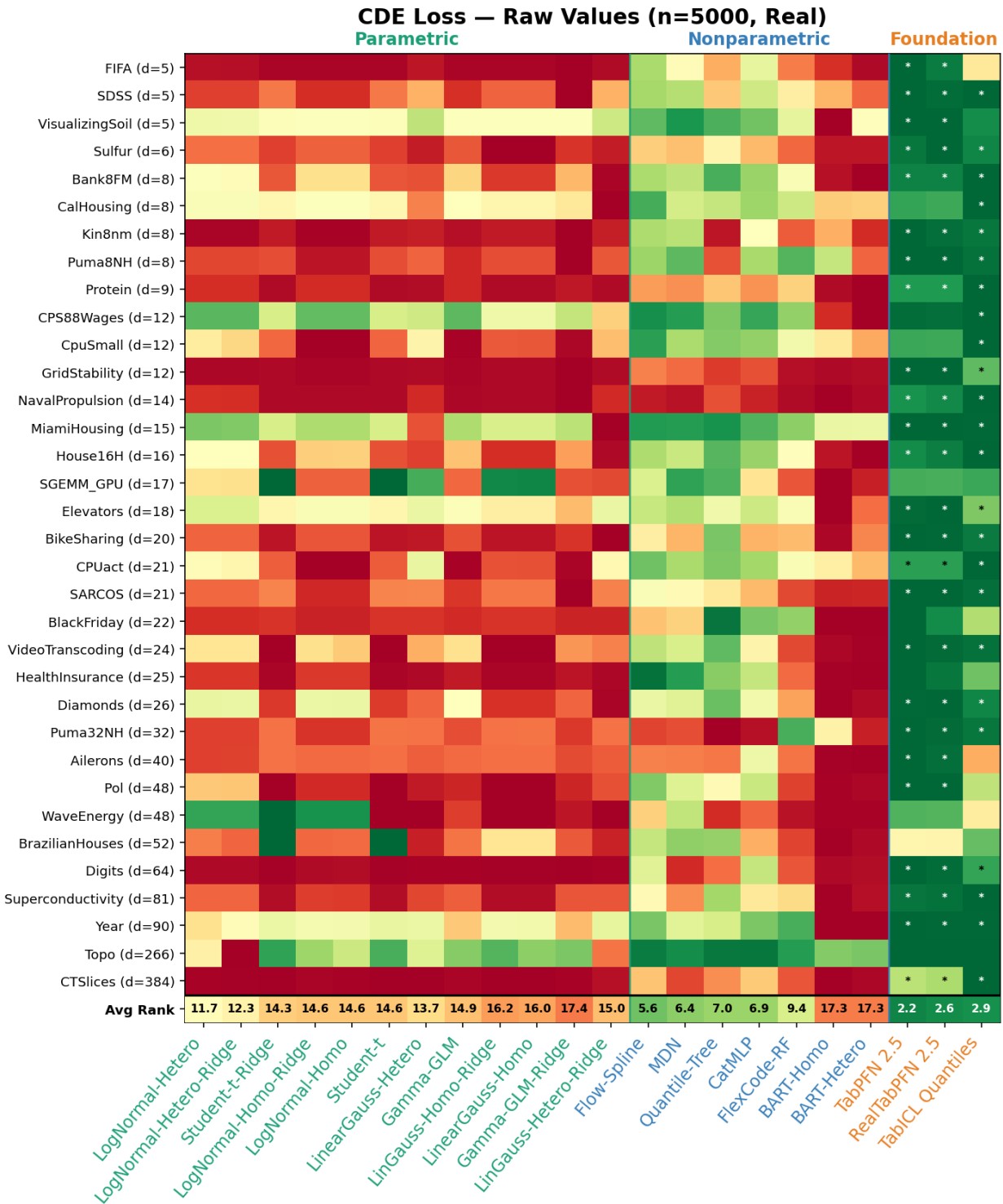

Figure 14: Raw CDE Loss – $n = 5000$, real data.

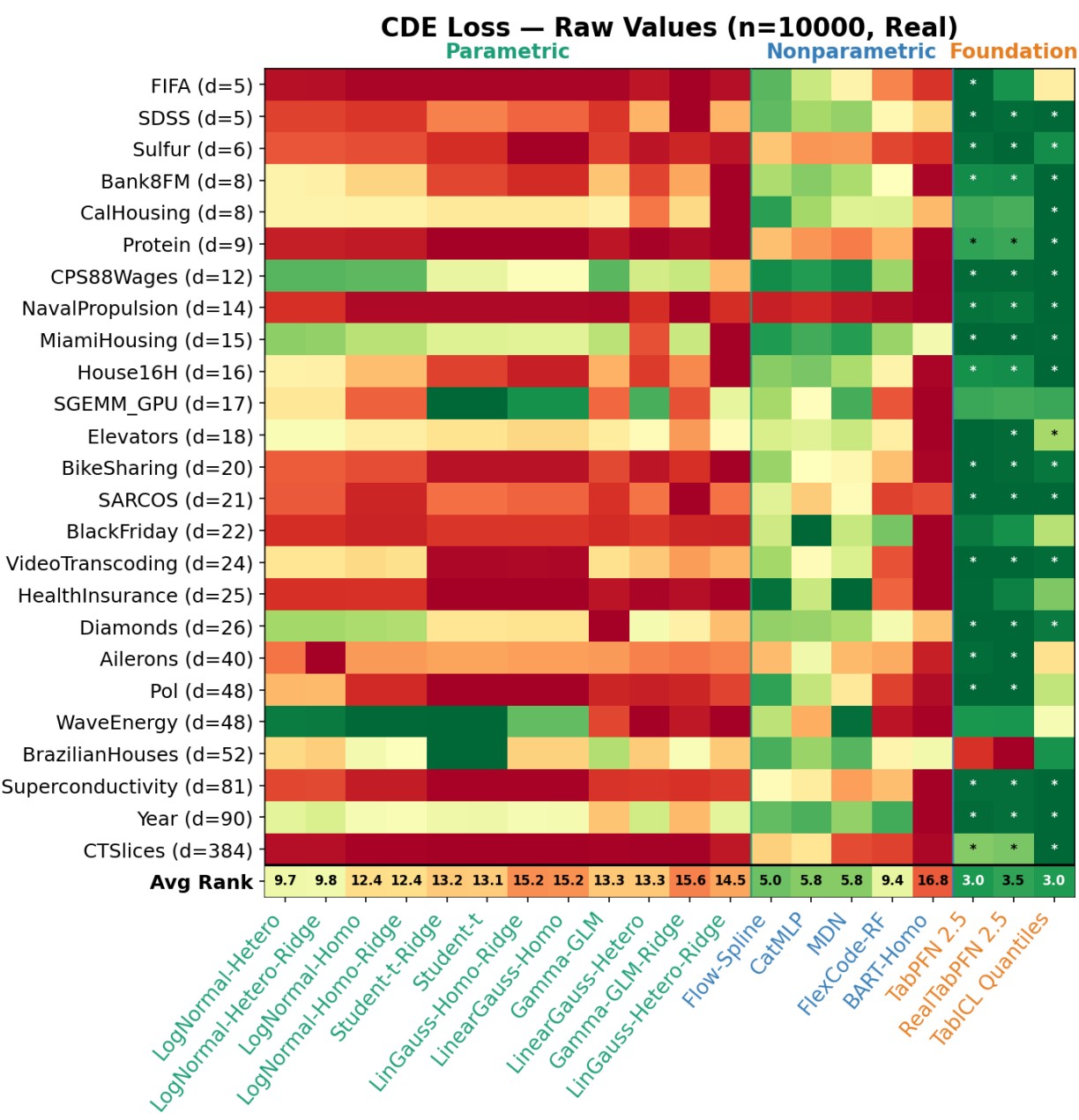

Figure 15: Raw CDE Loss – $n = 10000$, real data.

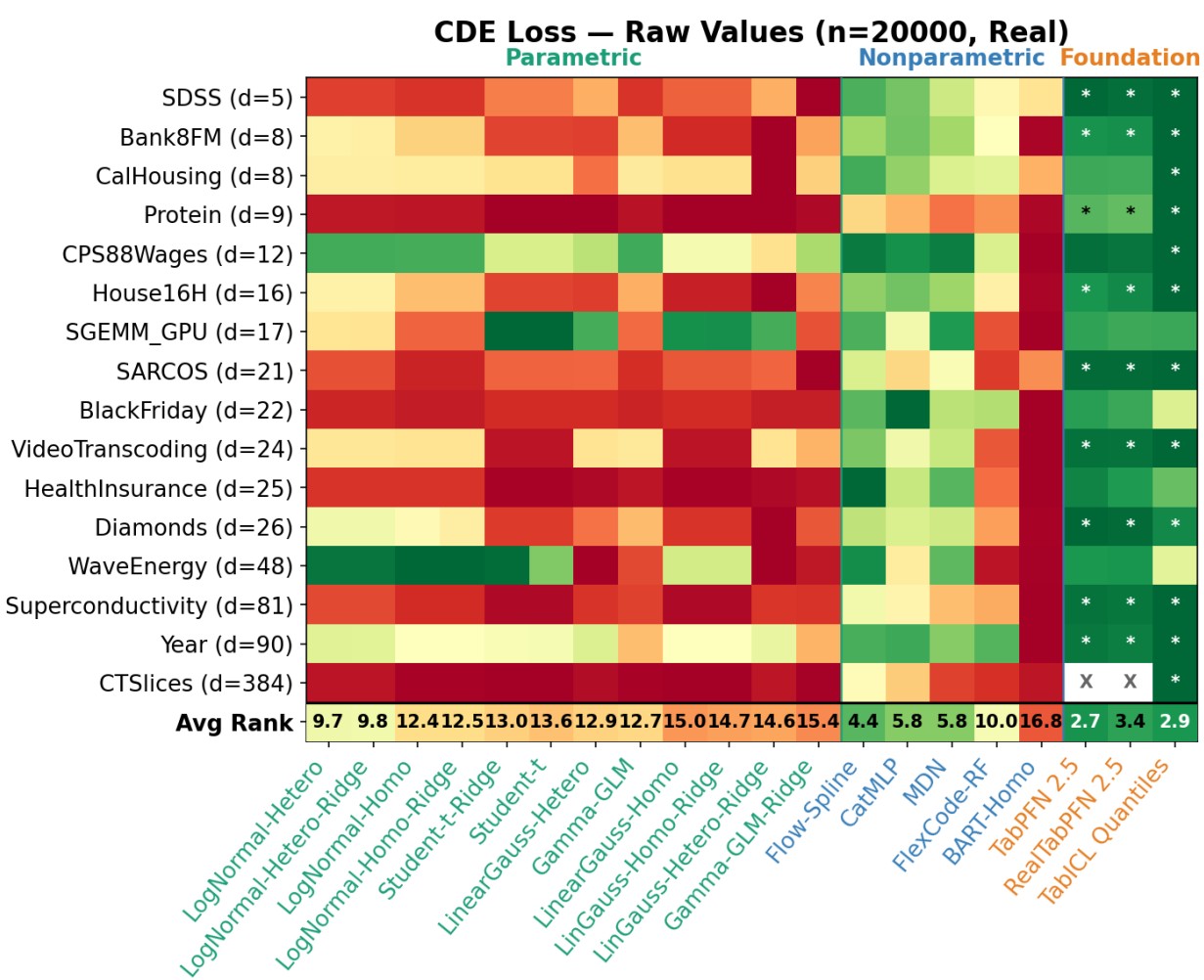

Figure 16: Raw CDE Loss – $n = 20000$, real data.

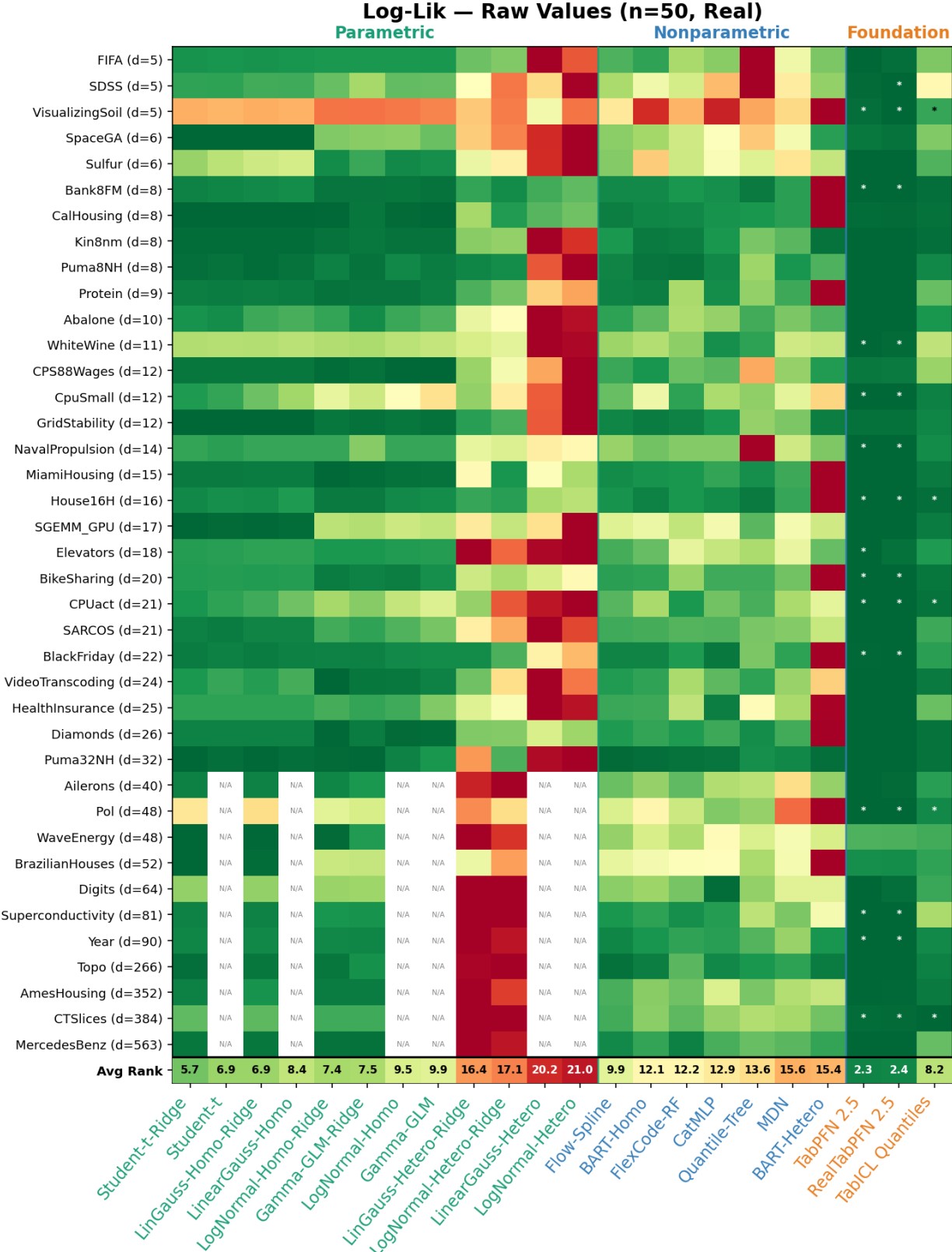

Figure 17: Raw Log-Likelihood – $n = 50$, real data.

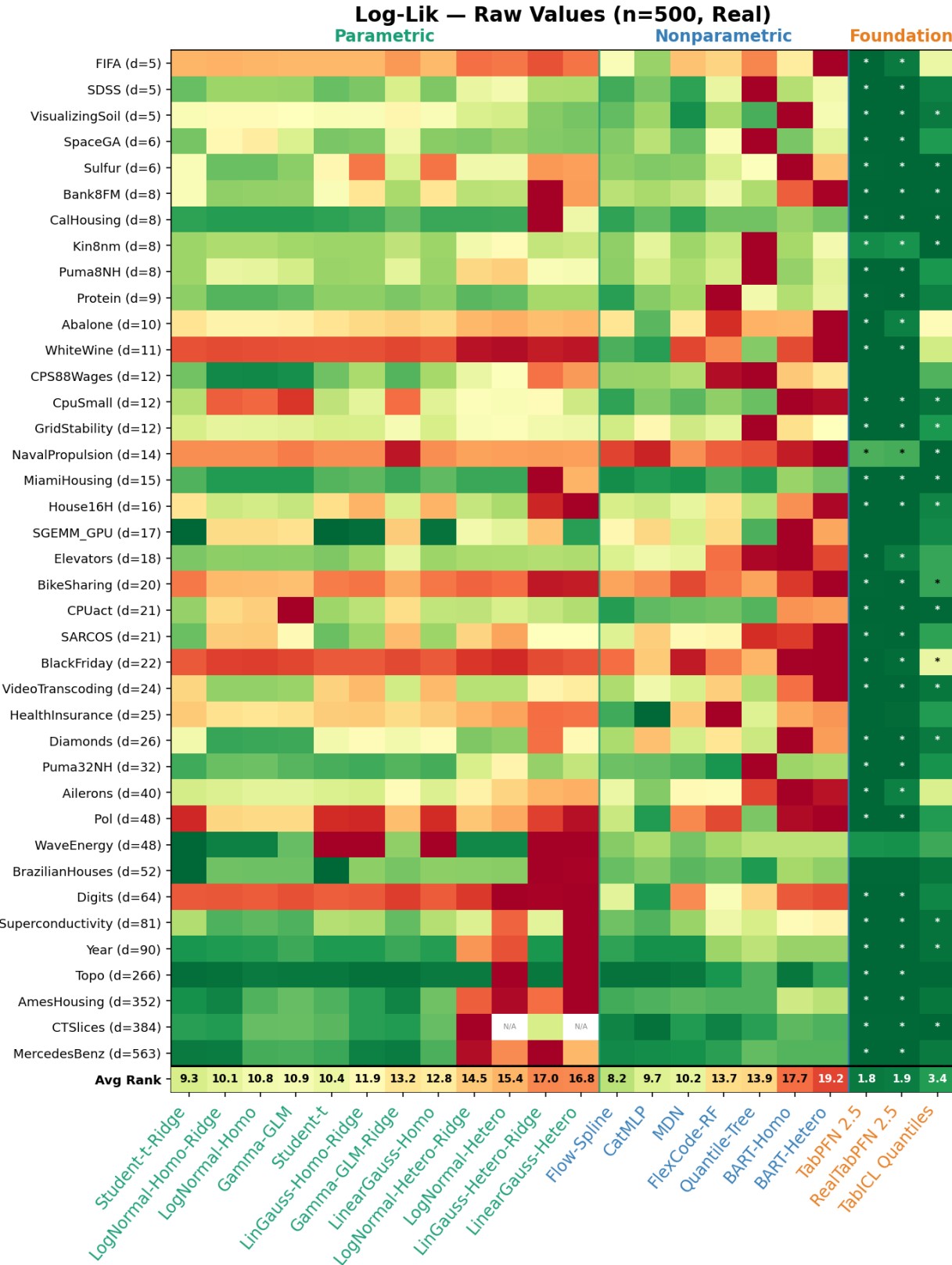

Figure 18: Raw Log-Likelihood – $n = 500$, real data.

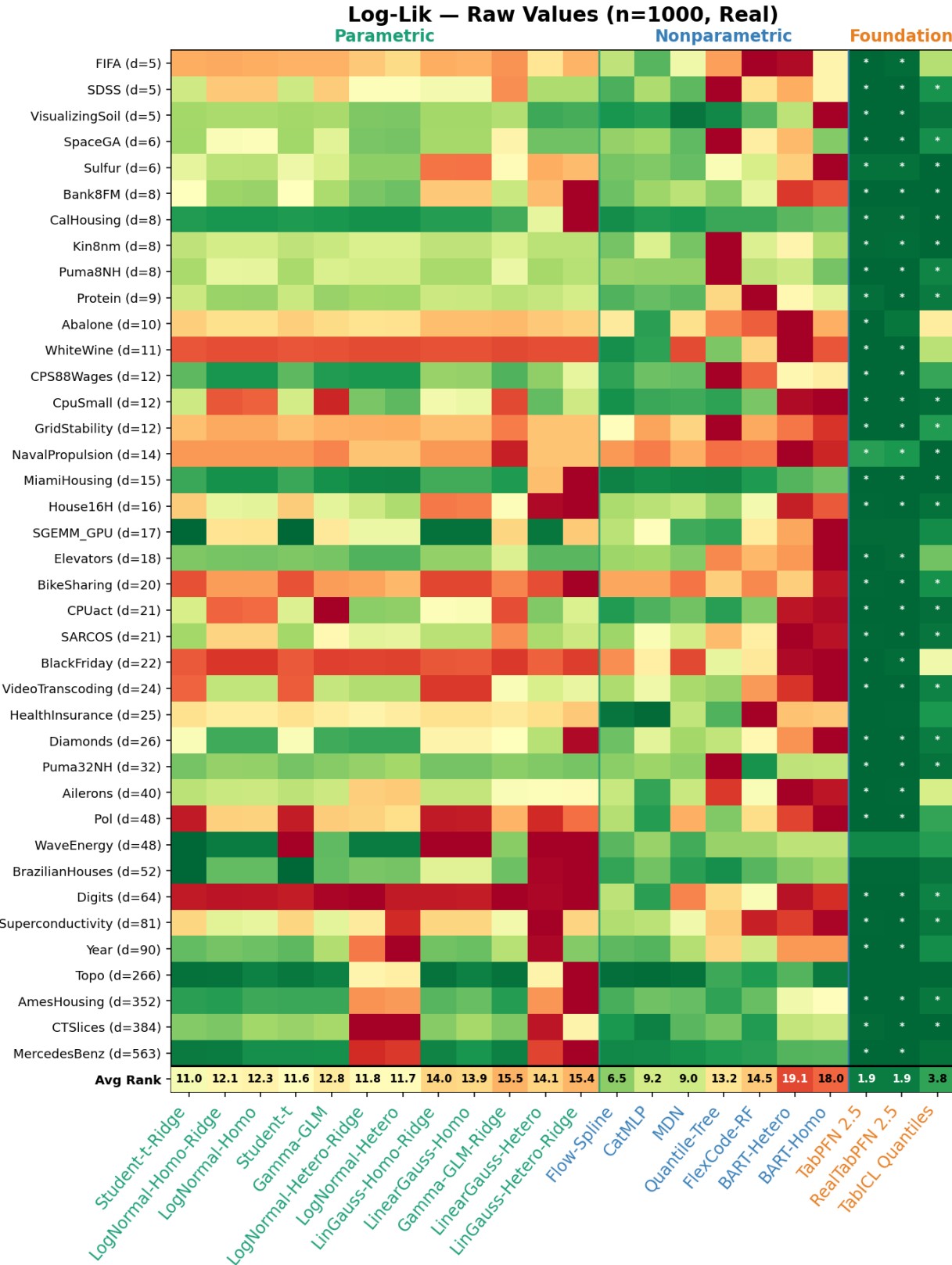

Figure 19: Raw Log-Likelihood – $n = 1000$, real data.

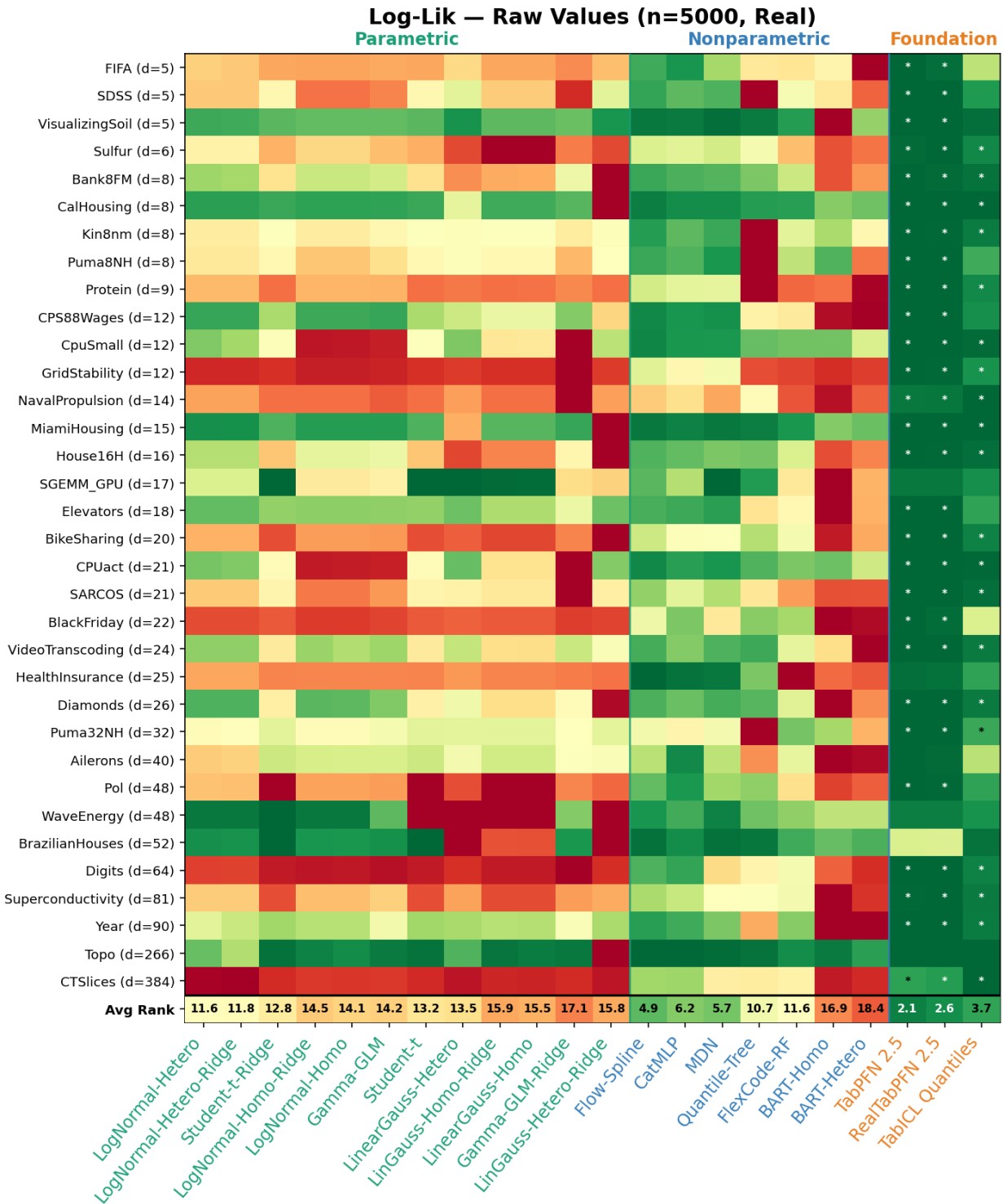

Figure 20: Raw Log-Likelihood – $n = 5000$, real data.

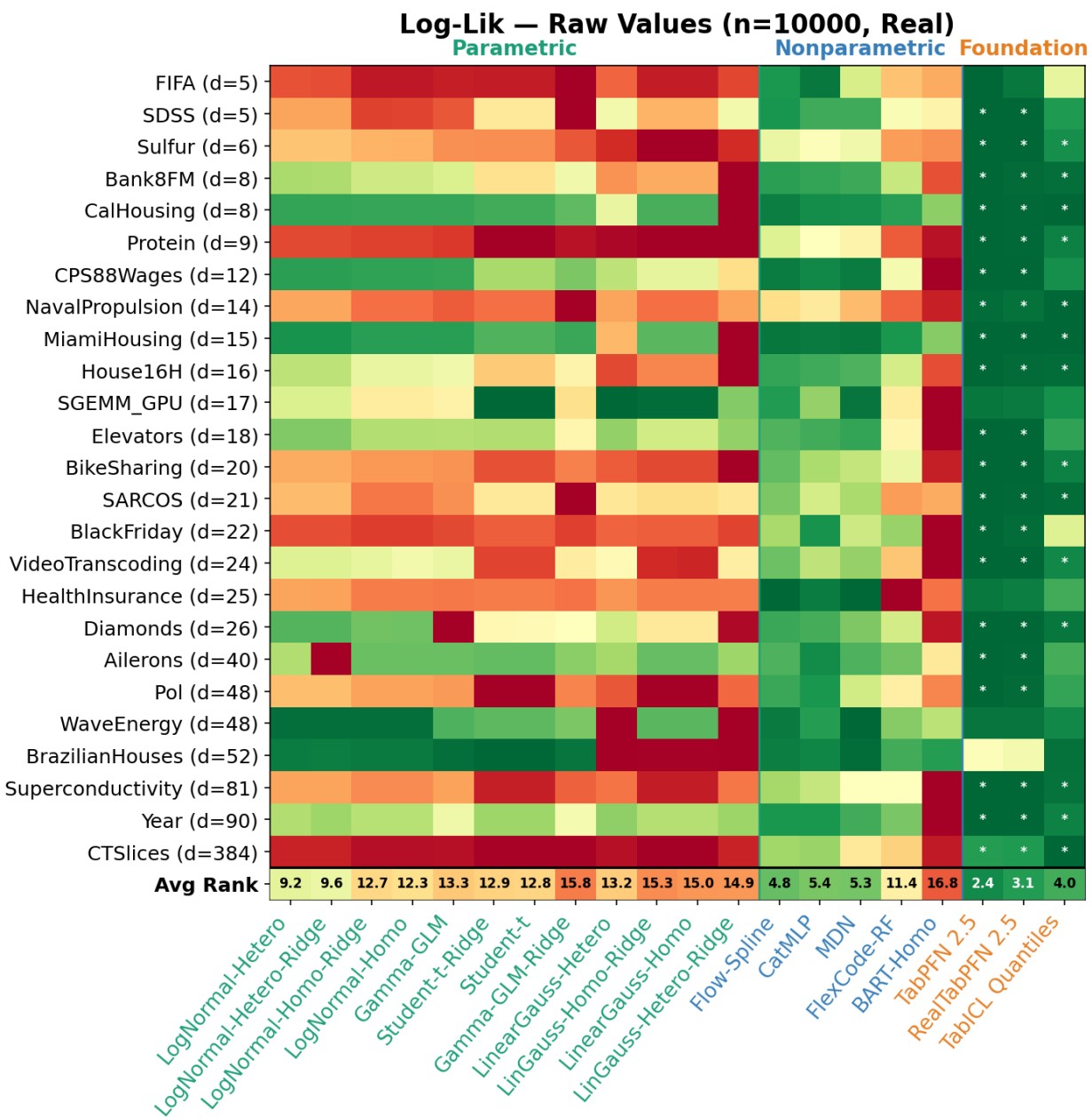

Figure 21: Raw Log-Likelihood – $n = 10000$, real data.

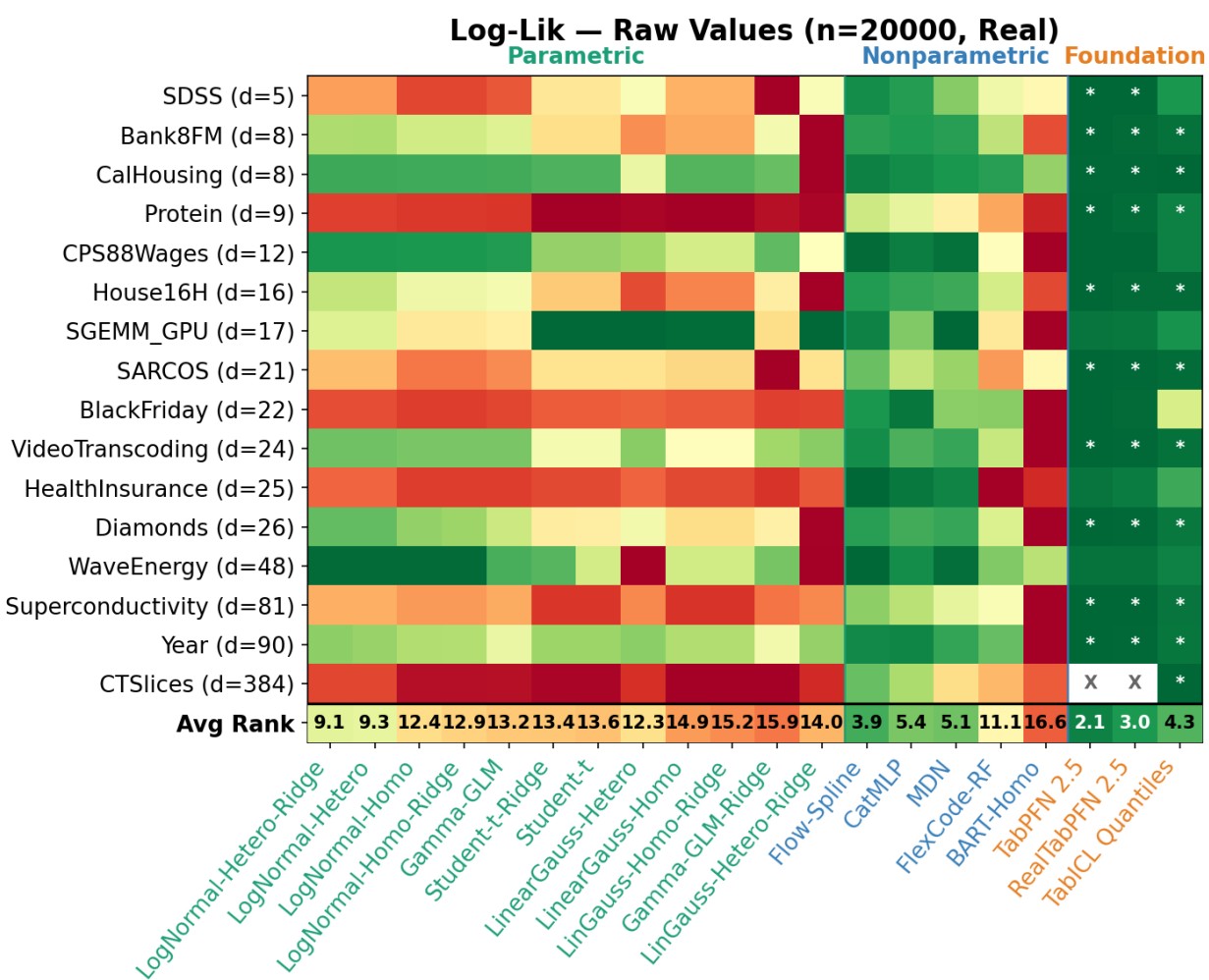

Figure 22: Raw Log-Likelihood – $n = 20000$, real data.

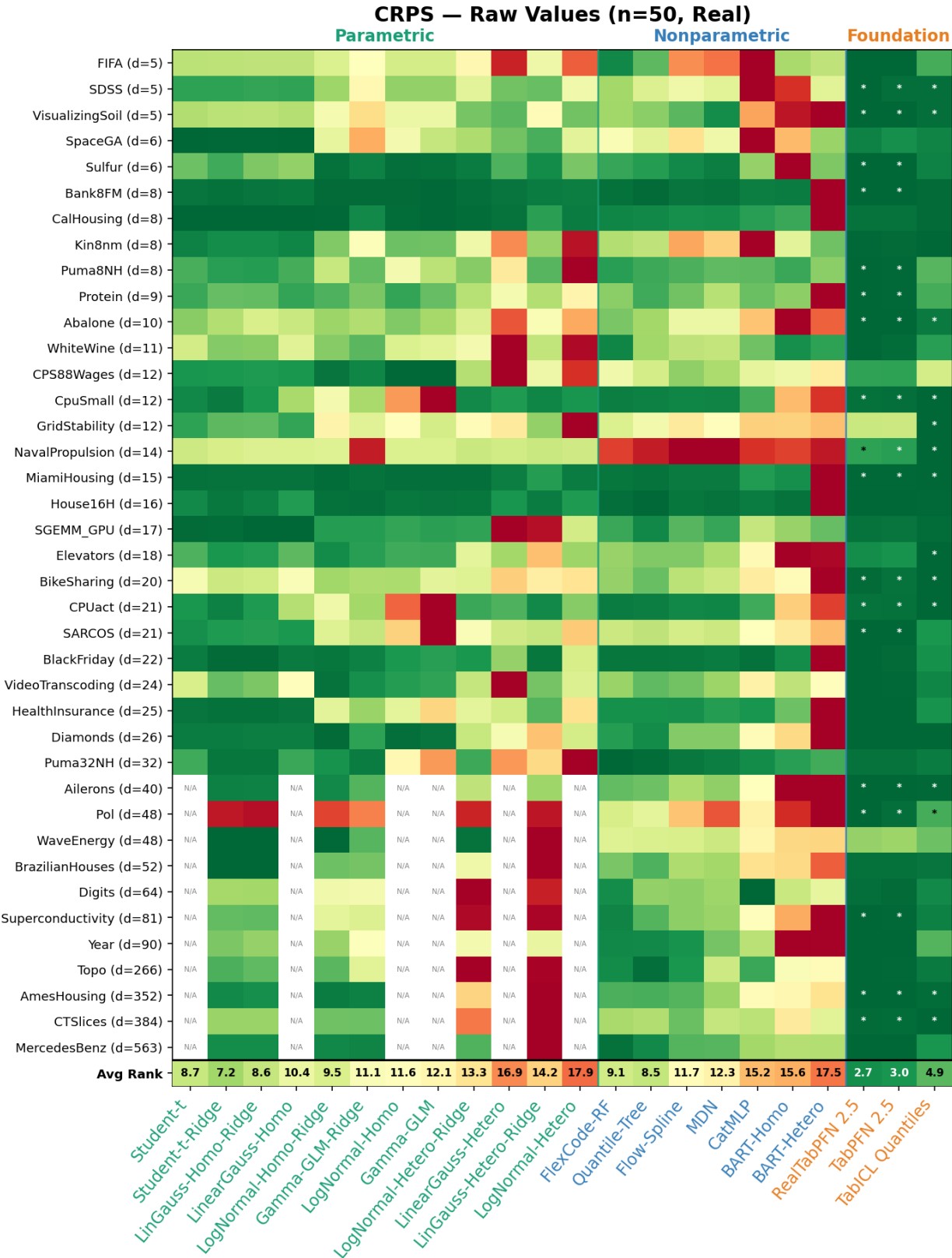

Figure 23: Raw CRPS – $n = 50$, real data.

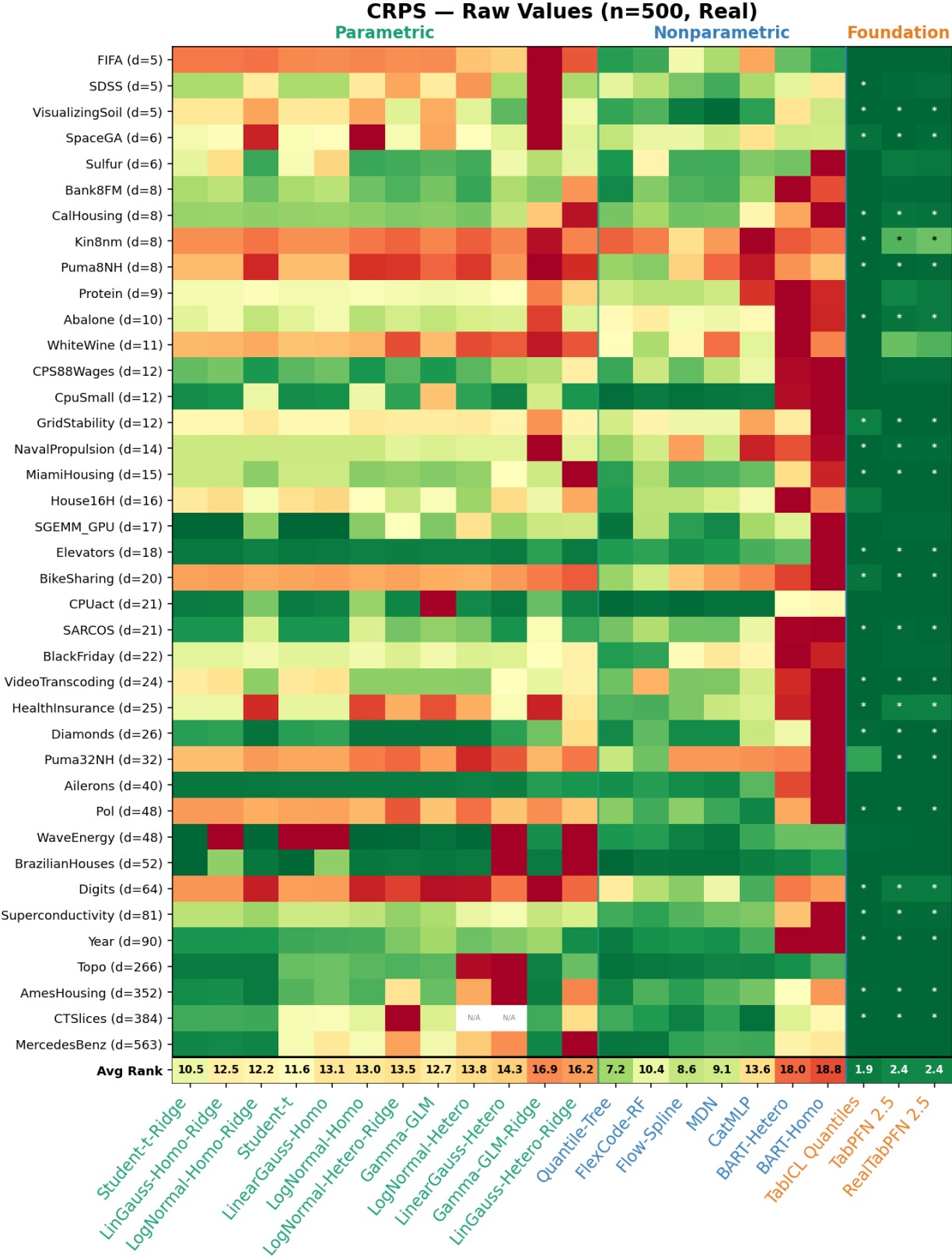

Figure 24: Raw CRPS – $n = 500$, real data.

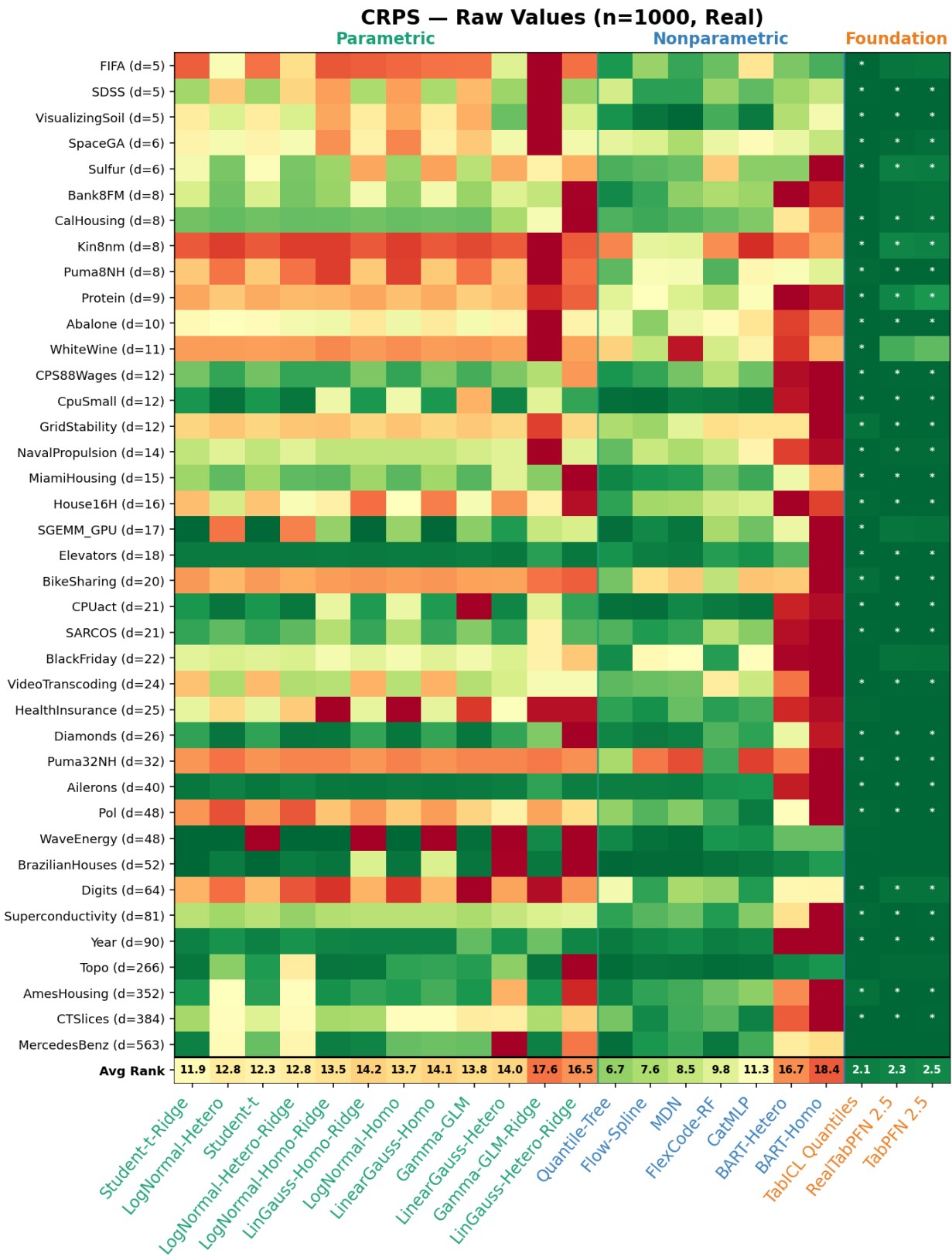

Figure 25: Raw CRPS – $n = 1000$, real data.

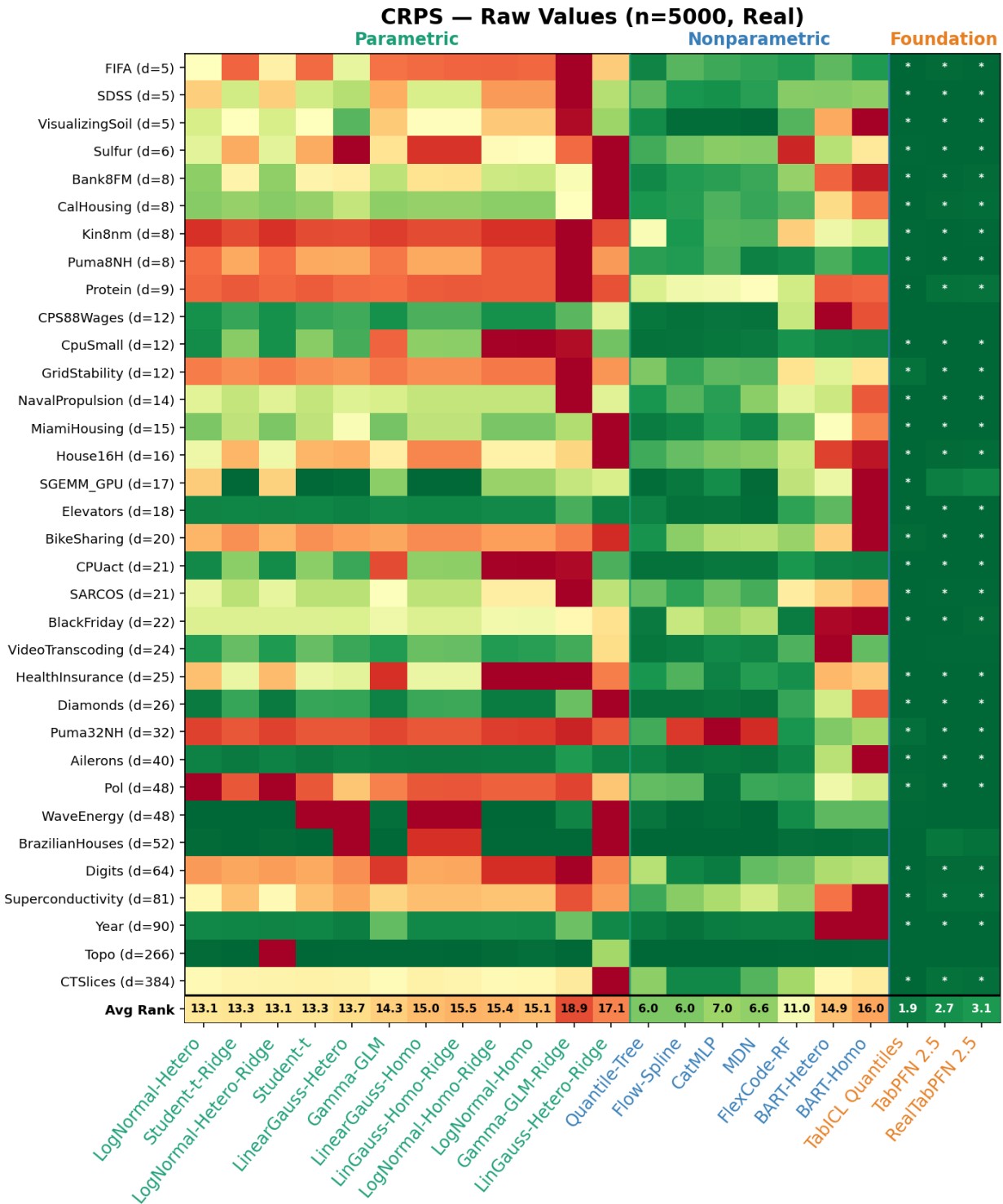

Figure 26: Raw CRPS $-$ $n = 5000$, real data.

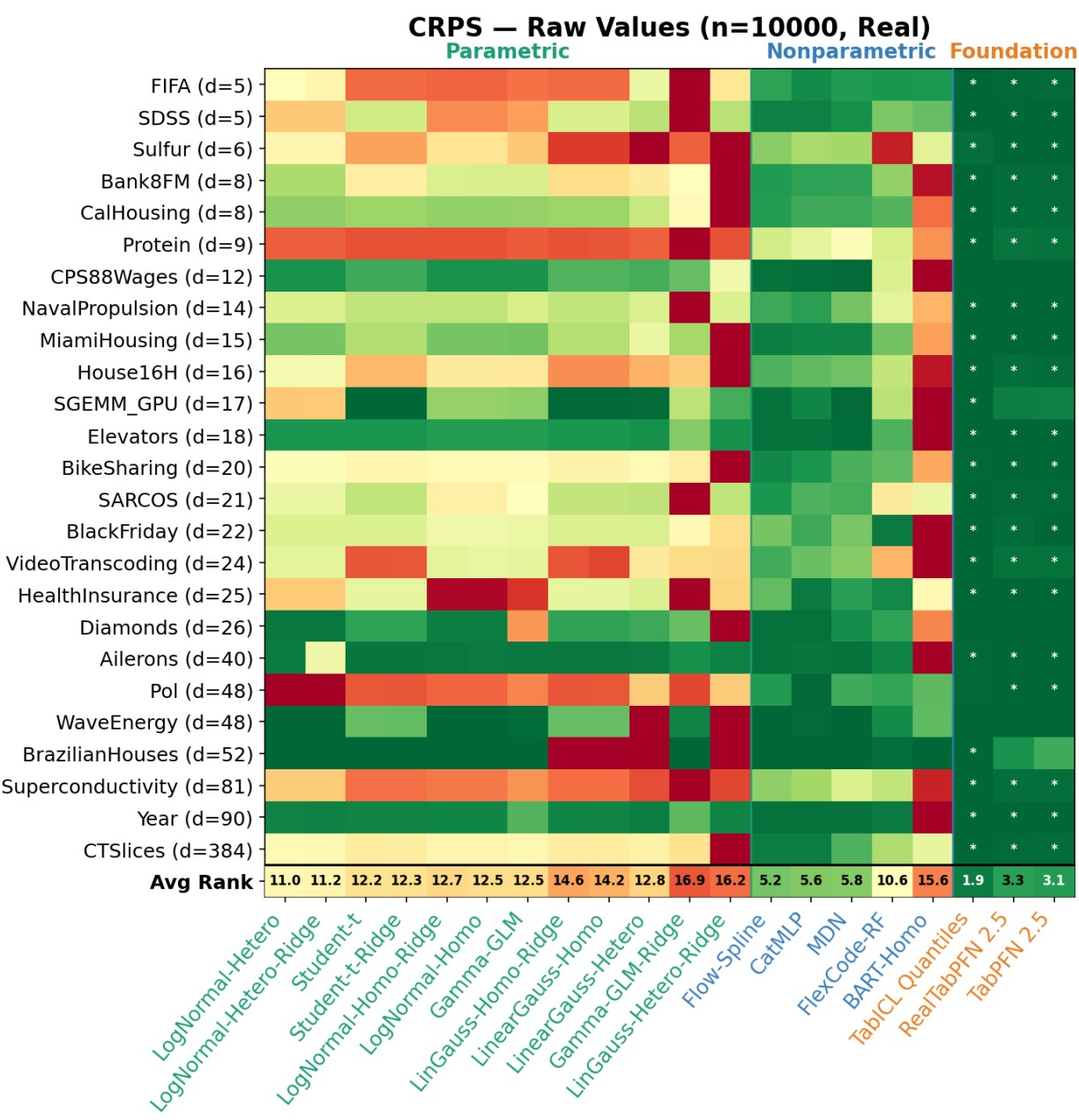

Figure 27: Raw CRPS – $n = 10000$, real data.

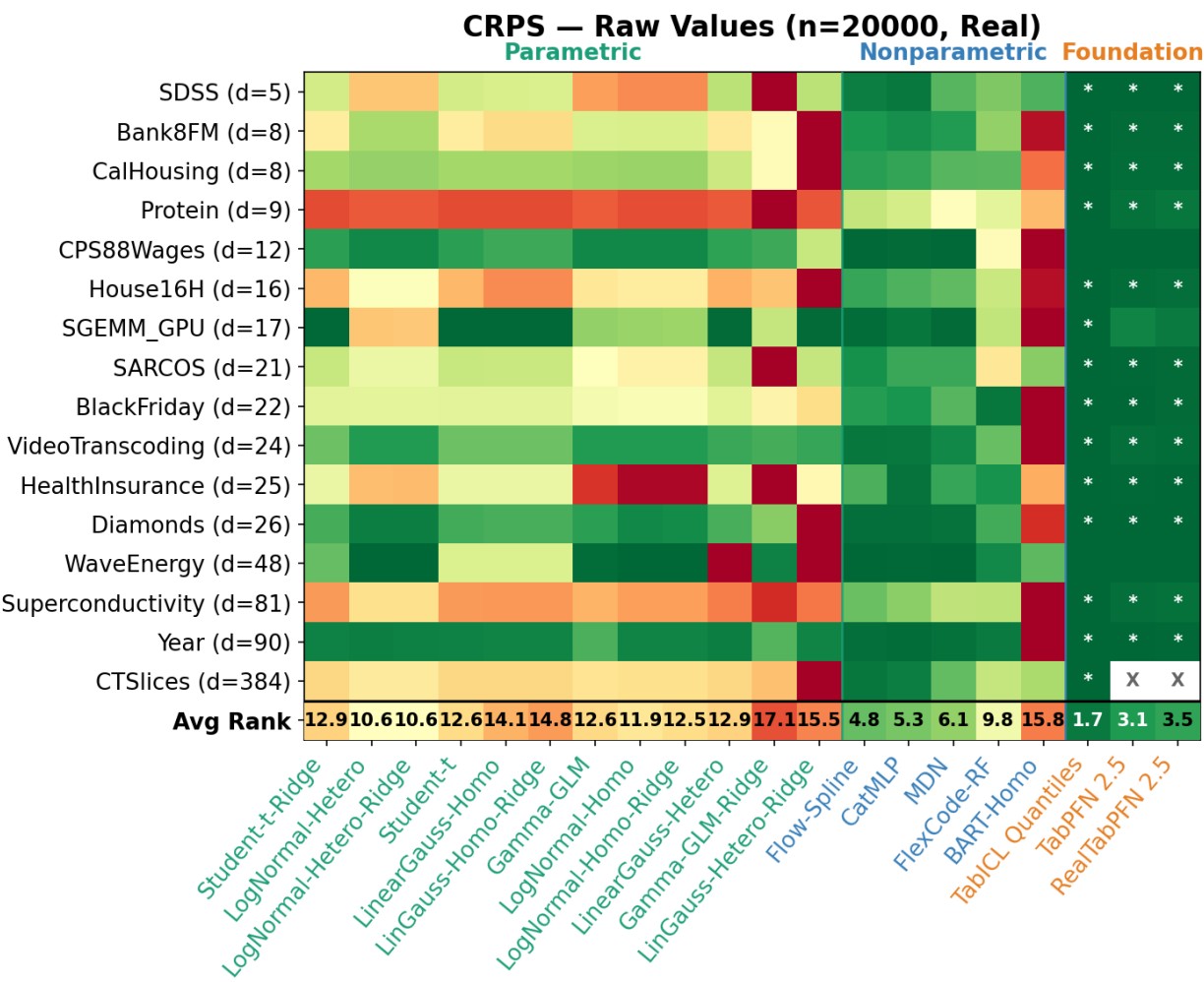

Figure 28: Raw CRPS – $n = 20000$, real data.

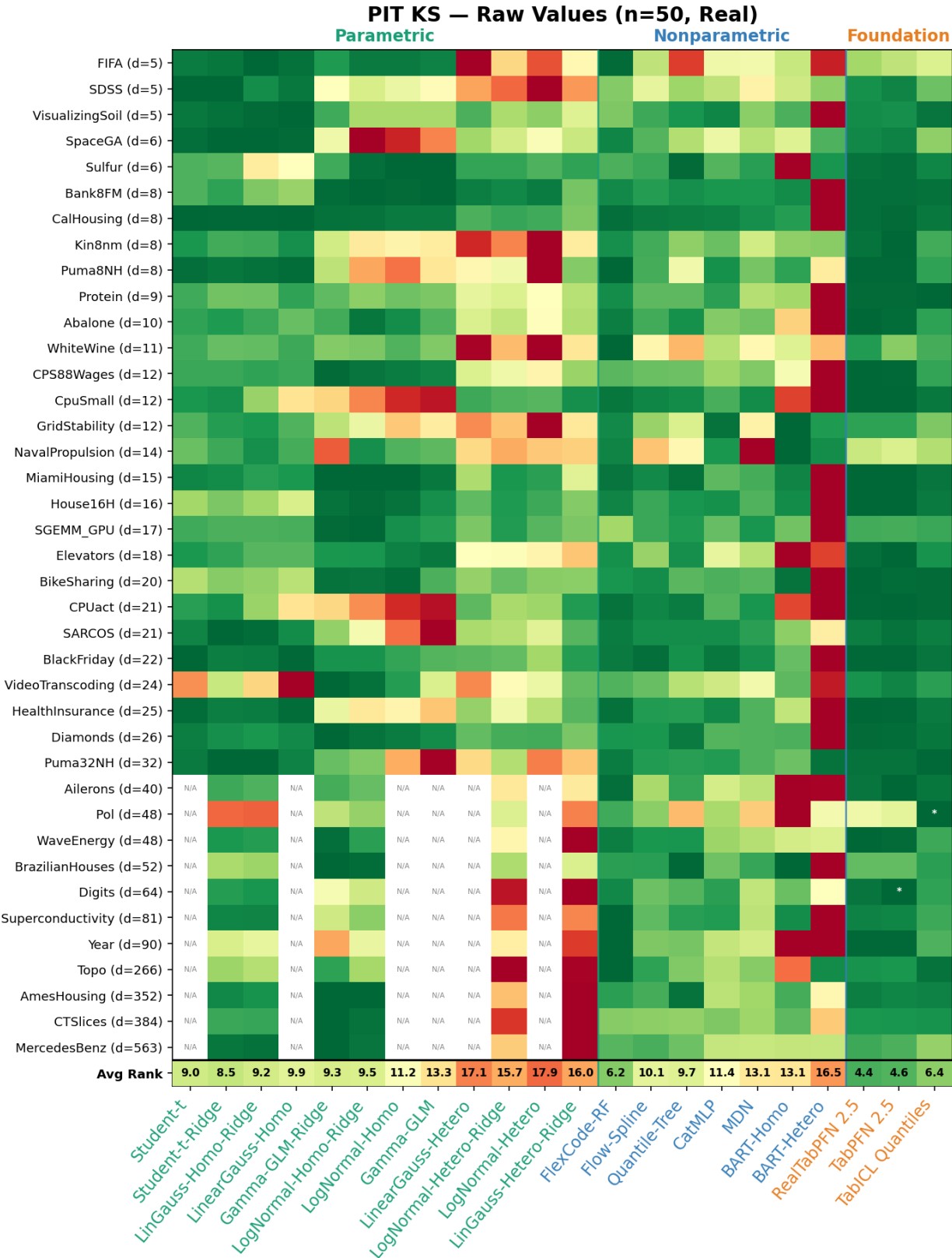

Figure 29: Raw PIT KS – $n = 50$, real data.

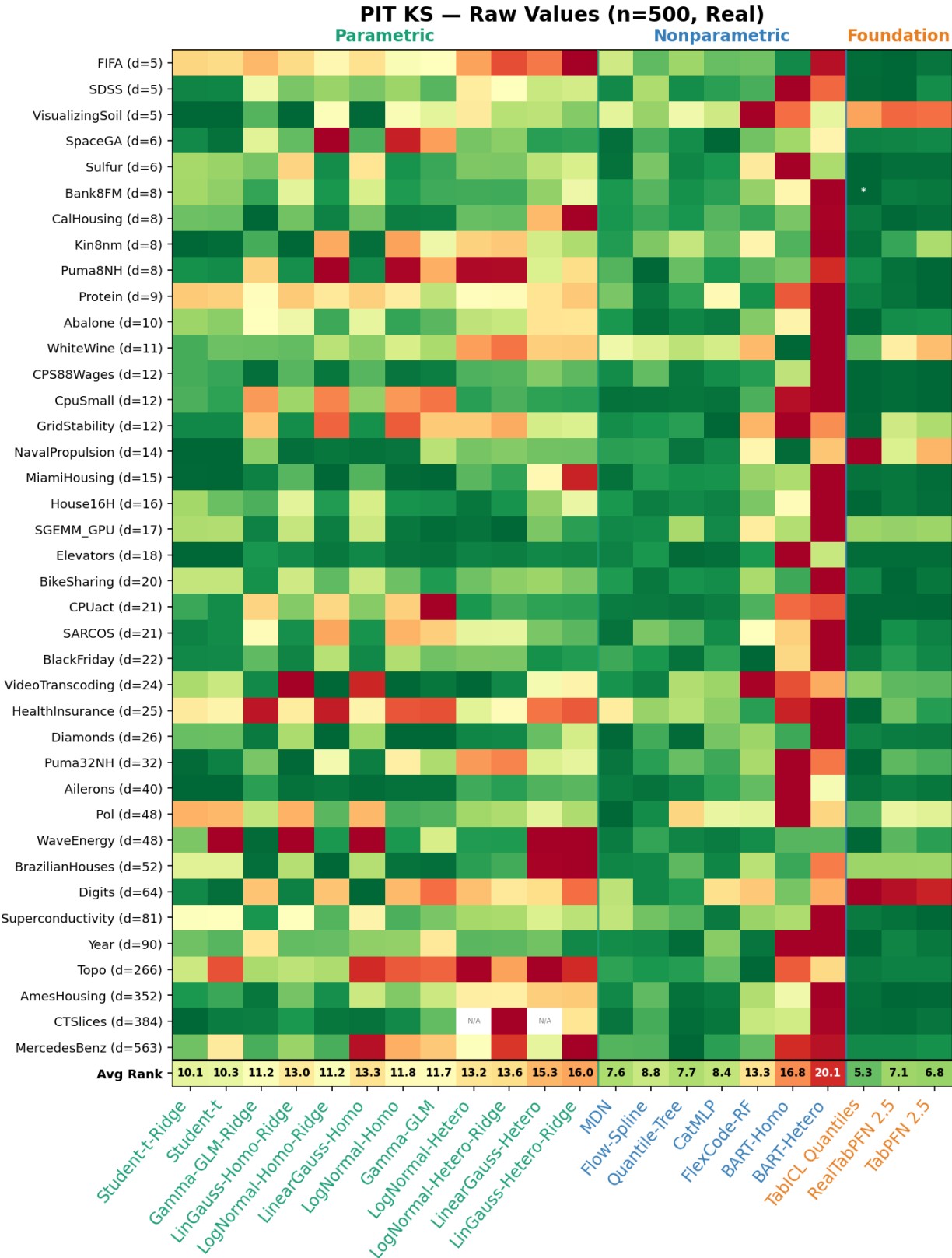

Figure 30: Raw PIT KS – $n = 500$, real data.

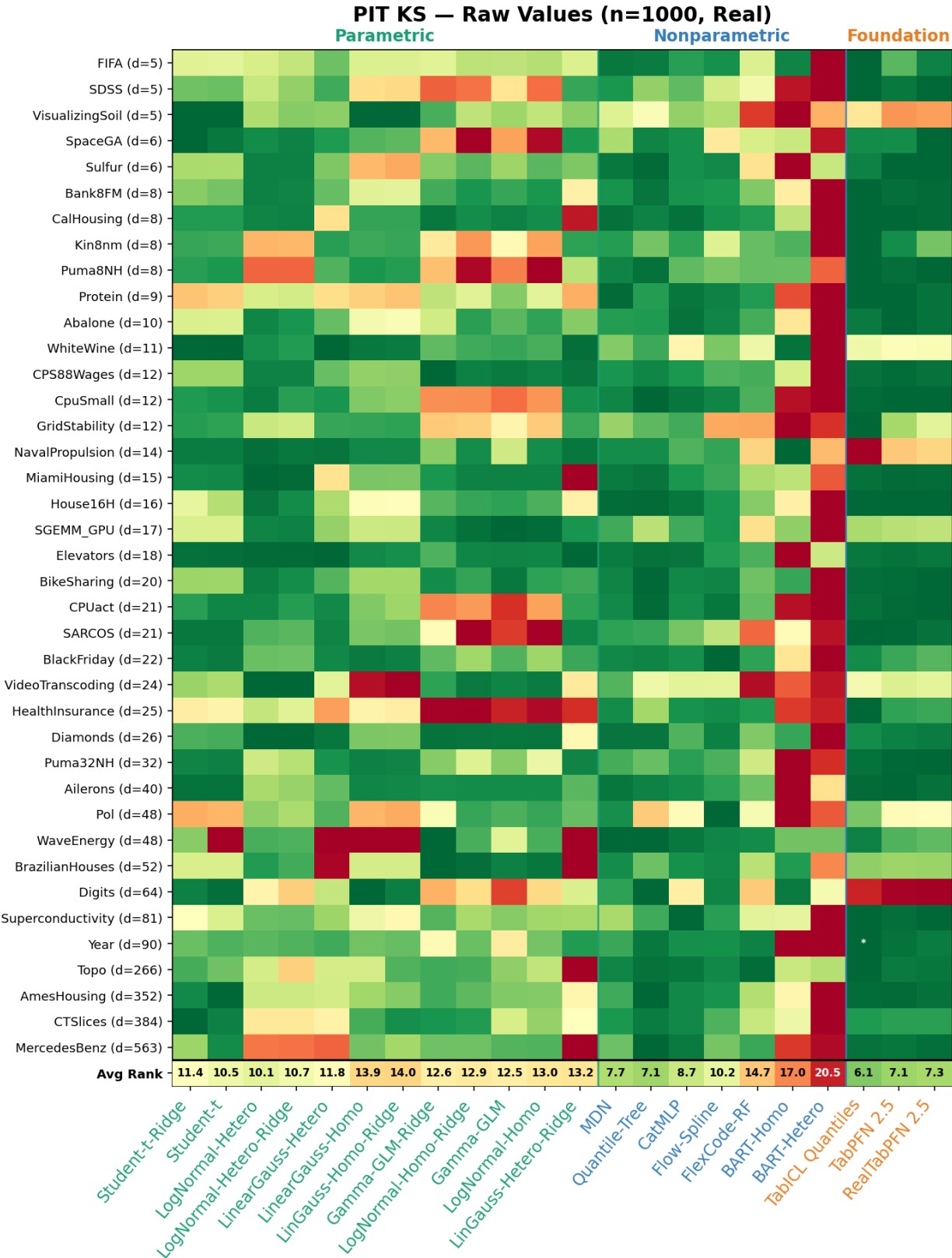

Figure 31: Raw PIT KS – $n = 1000$, real data.

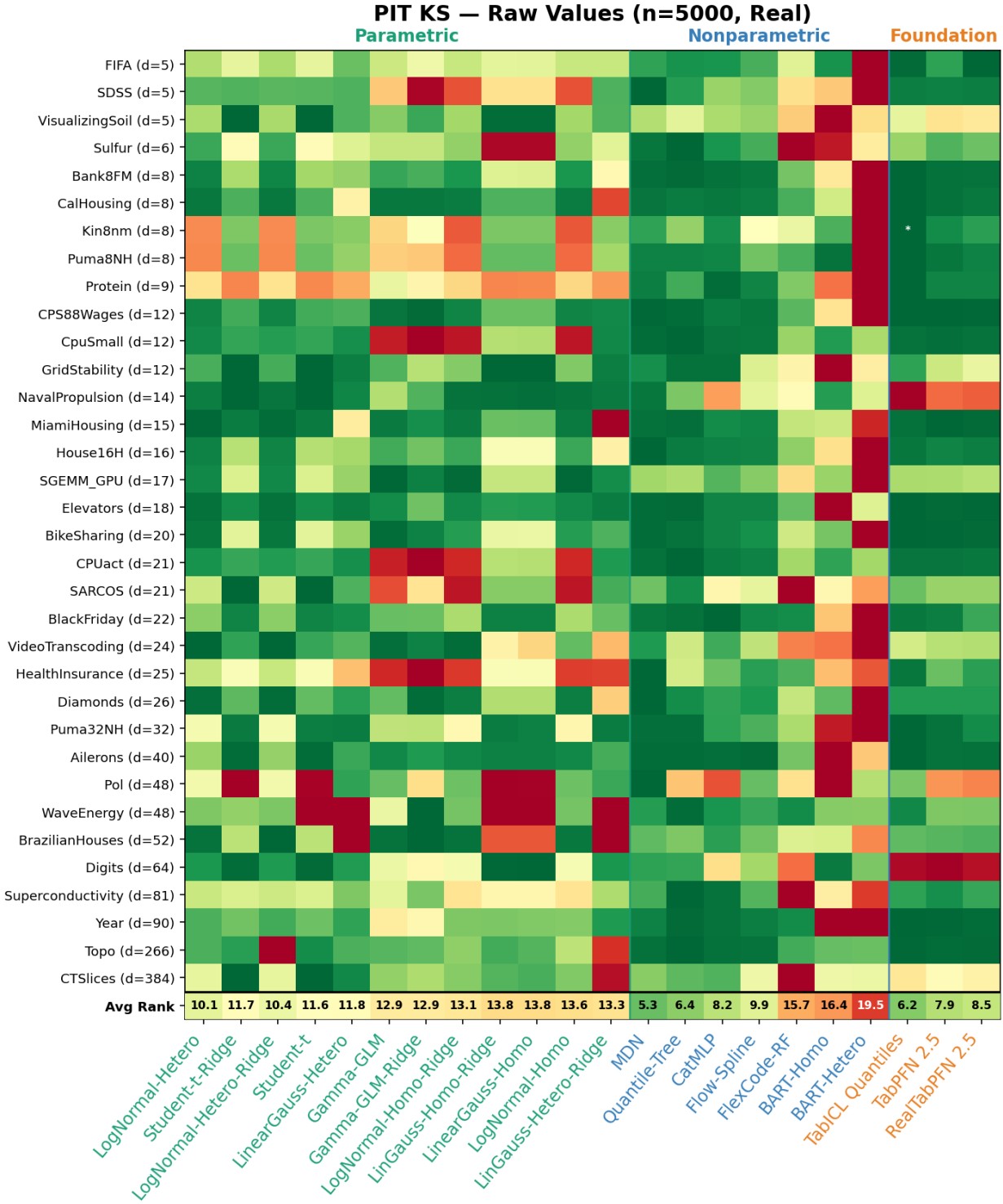

Figure 32: Raw PIT KS – $n = 5000$, real data.

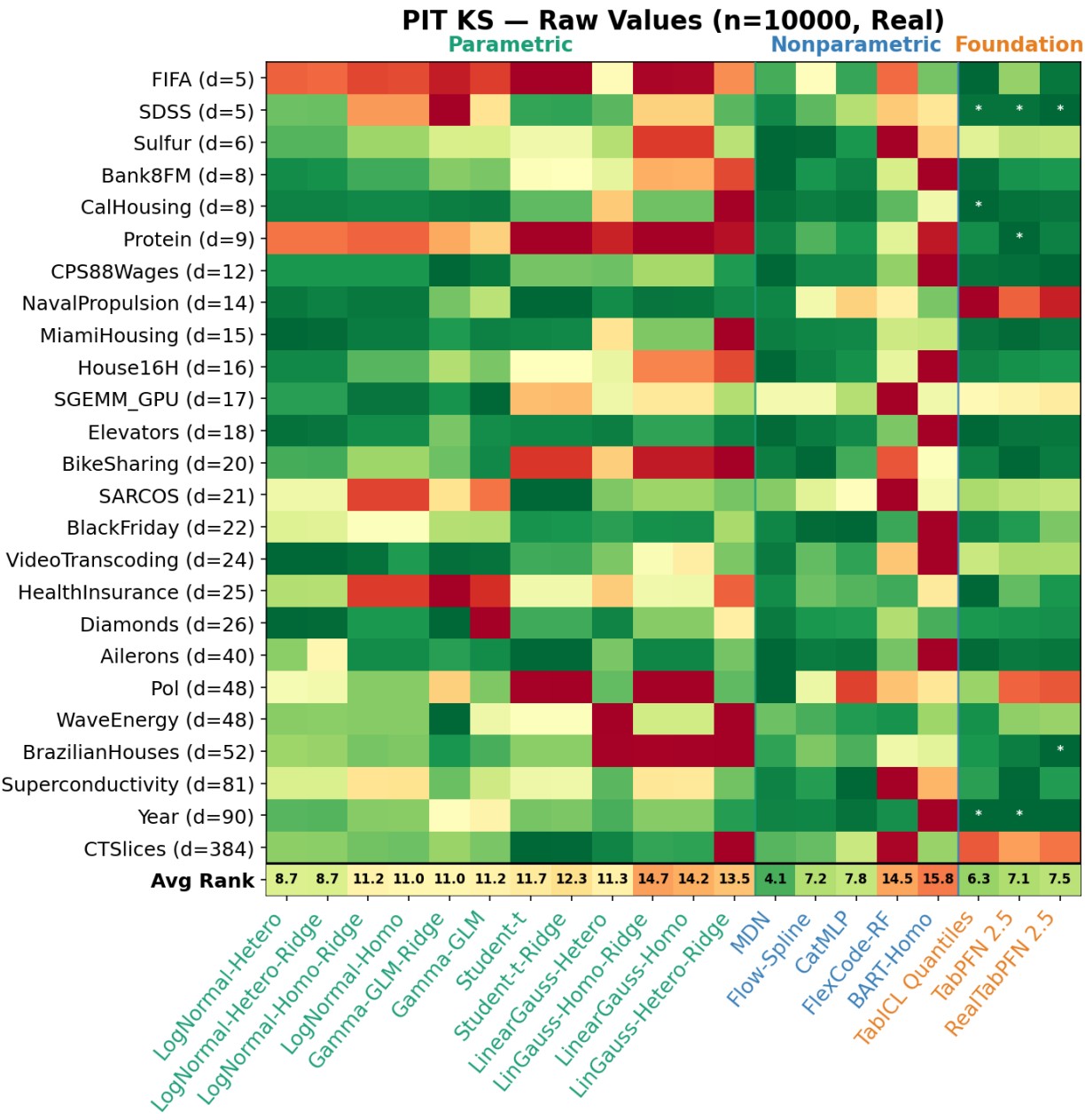

Figure 33: Raw PIT KS – $n = 10000$, real data.

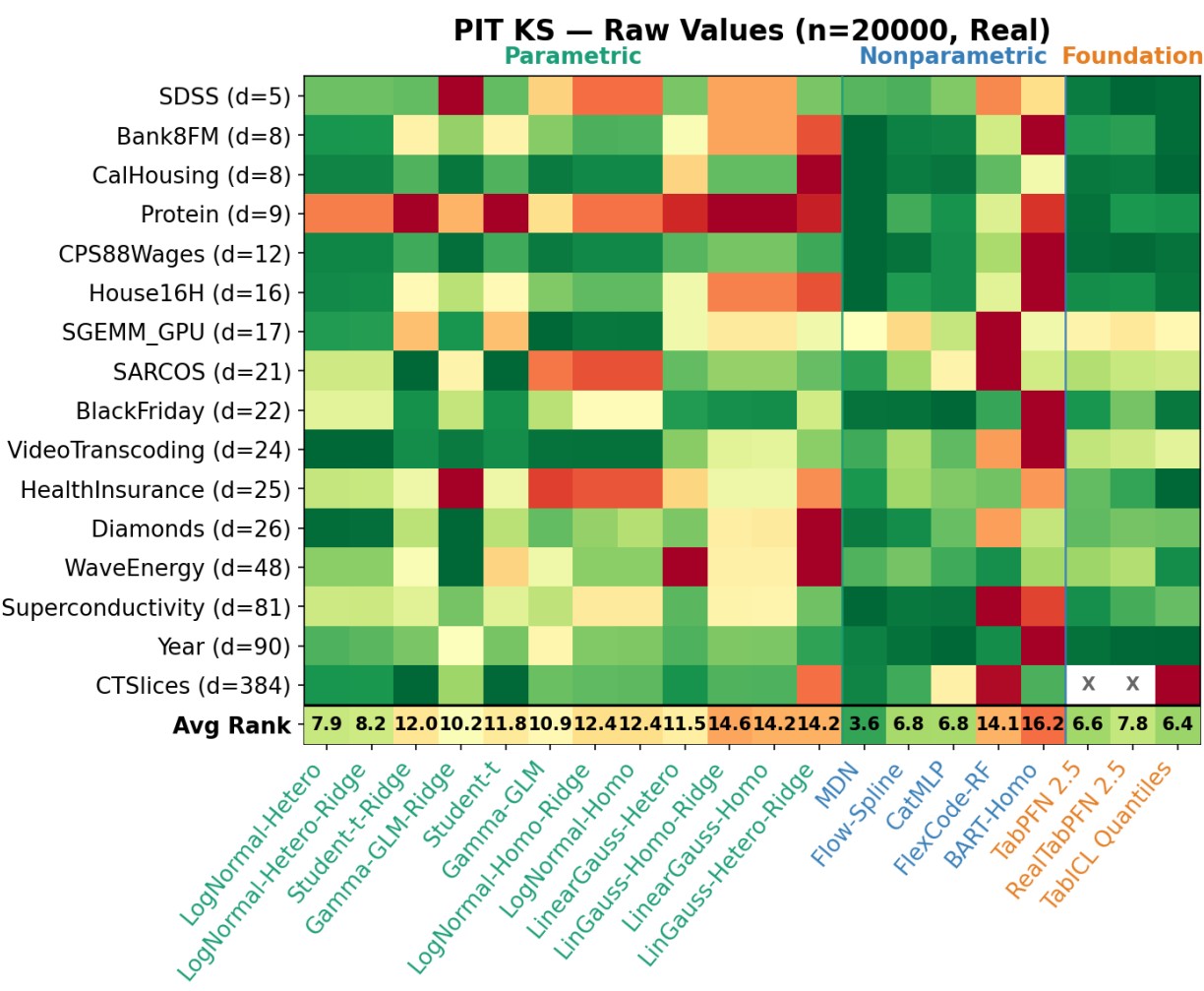

Figure 34: Raw PIT KS – $n = 20000$, real data.

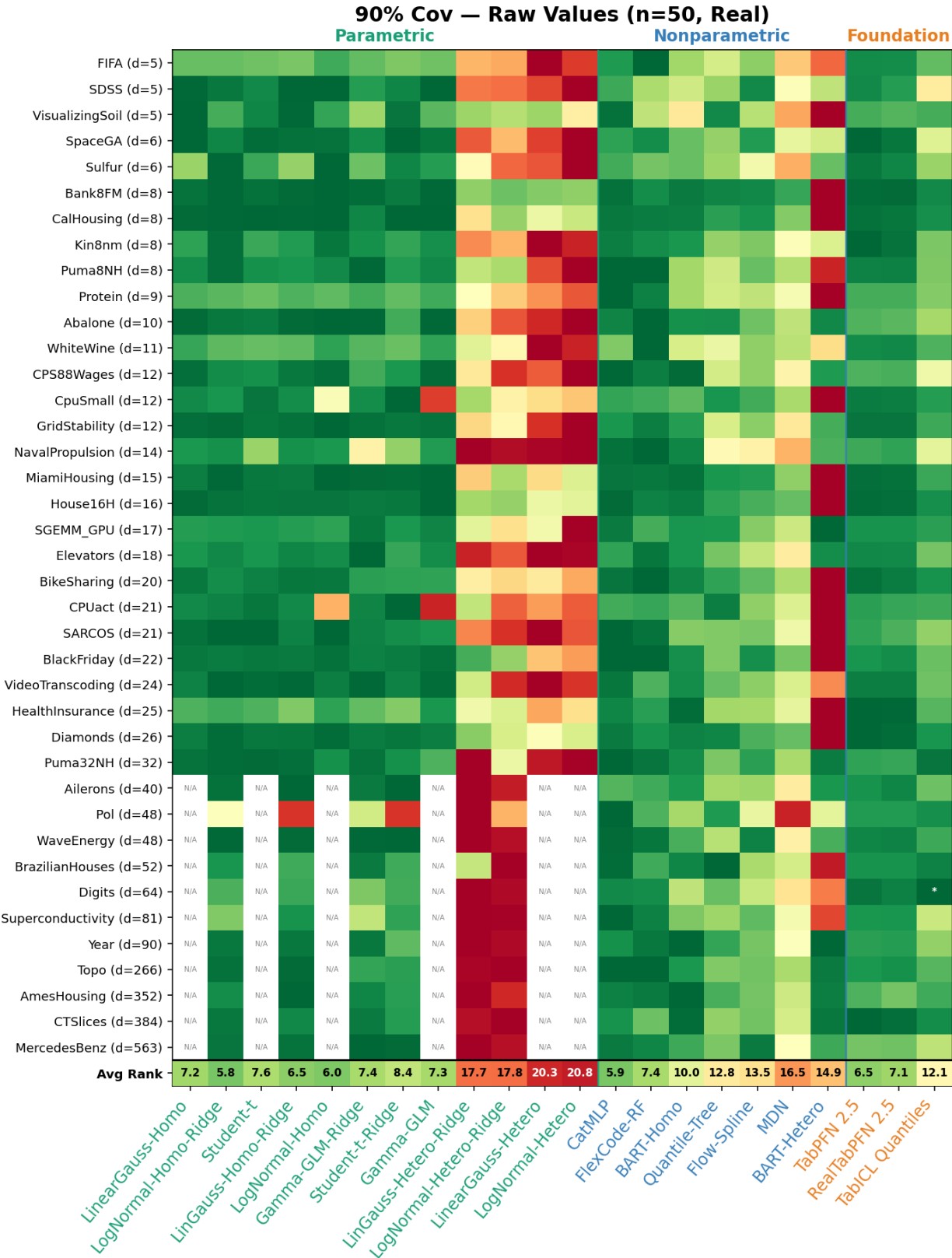

Figure 35: Raw 90% Coverage – $n = 50$, real data.

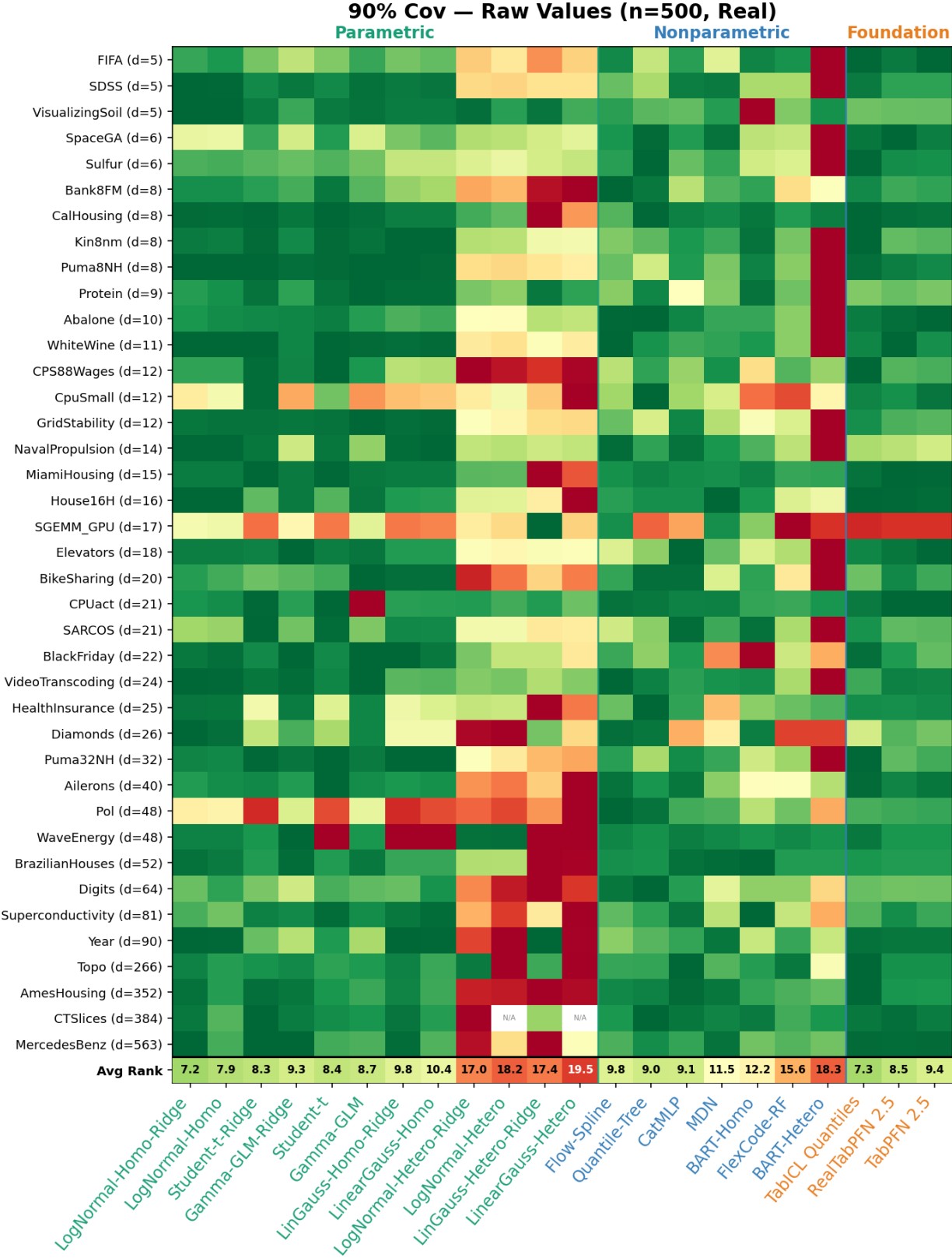

Figure 36: Raw 90% Coverage – $n = 500$, real data.

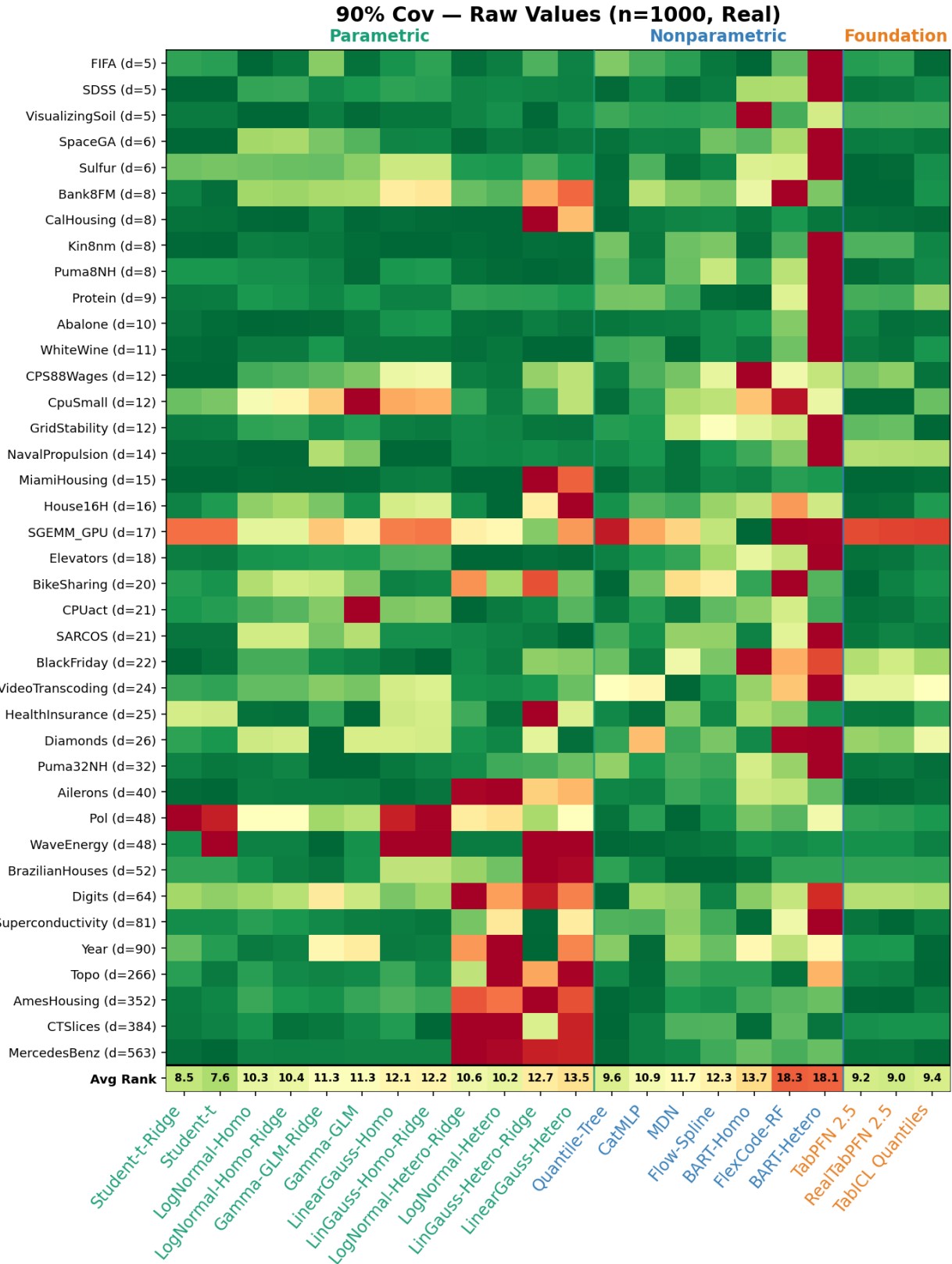

Figure 37: Raw 90% Coverage – $n = 1000$, real data.

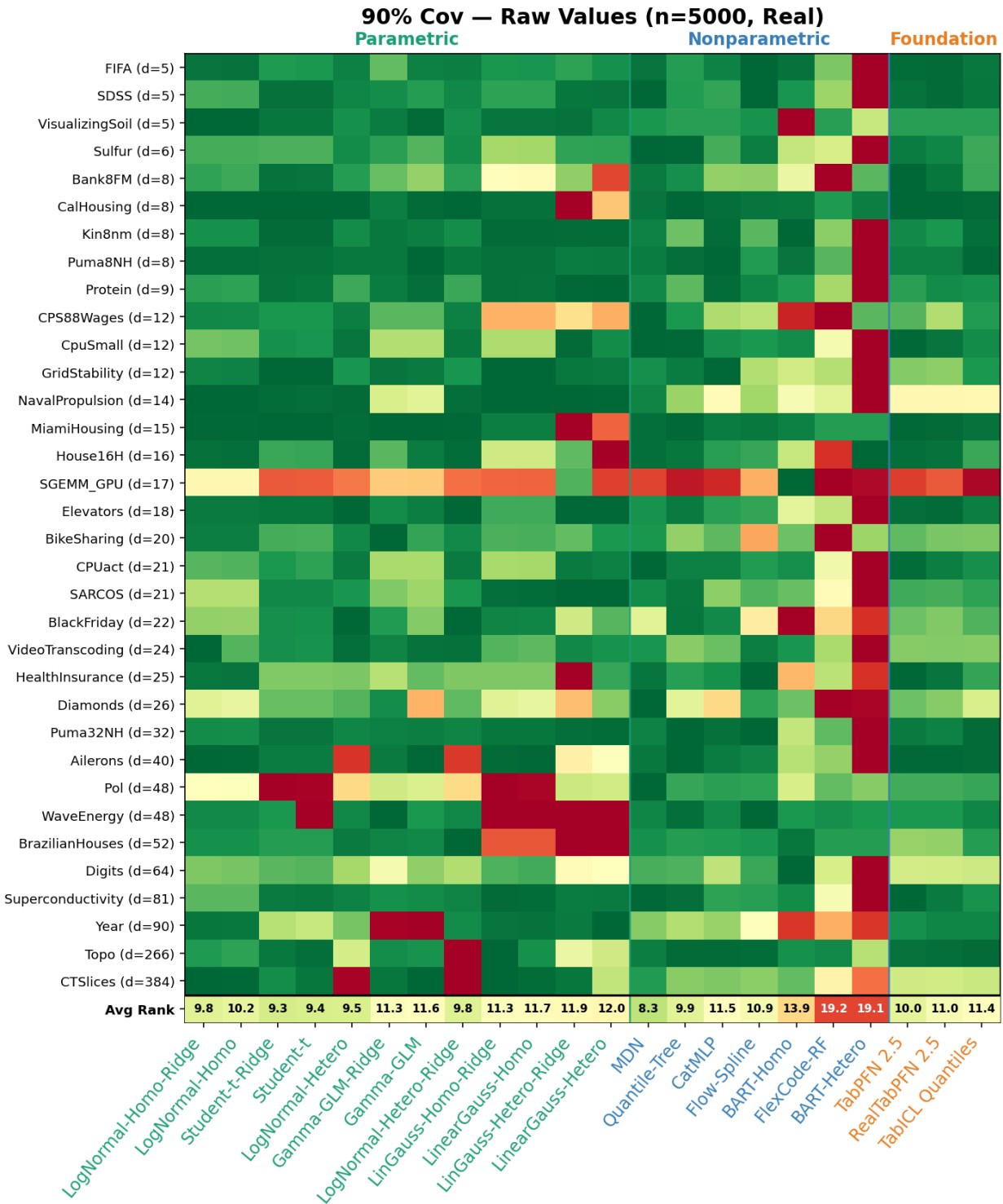

Figure 38: Raw 90% Coverage – $n = 5000$, real data.

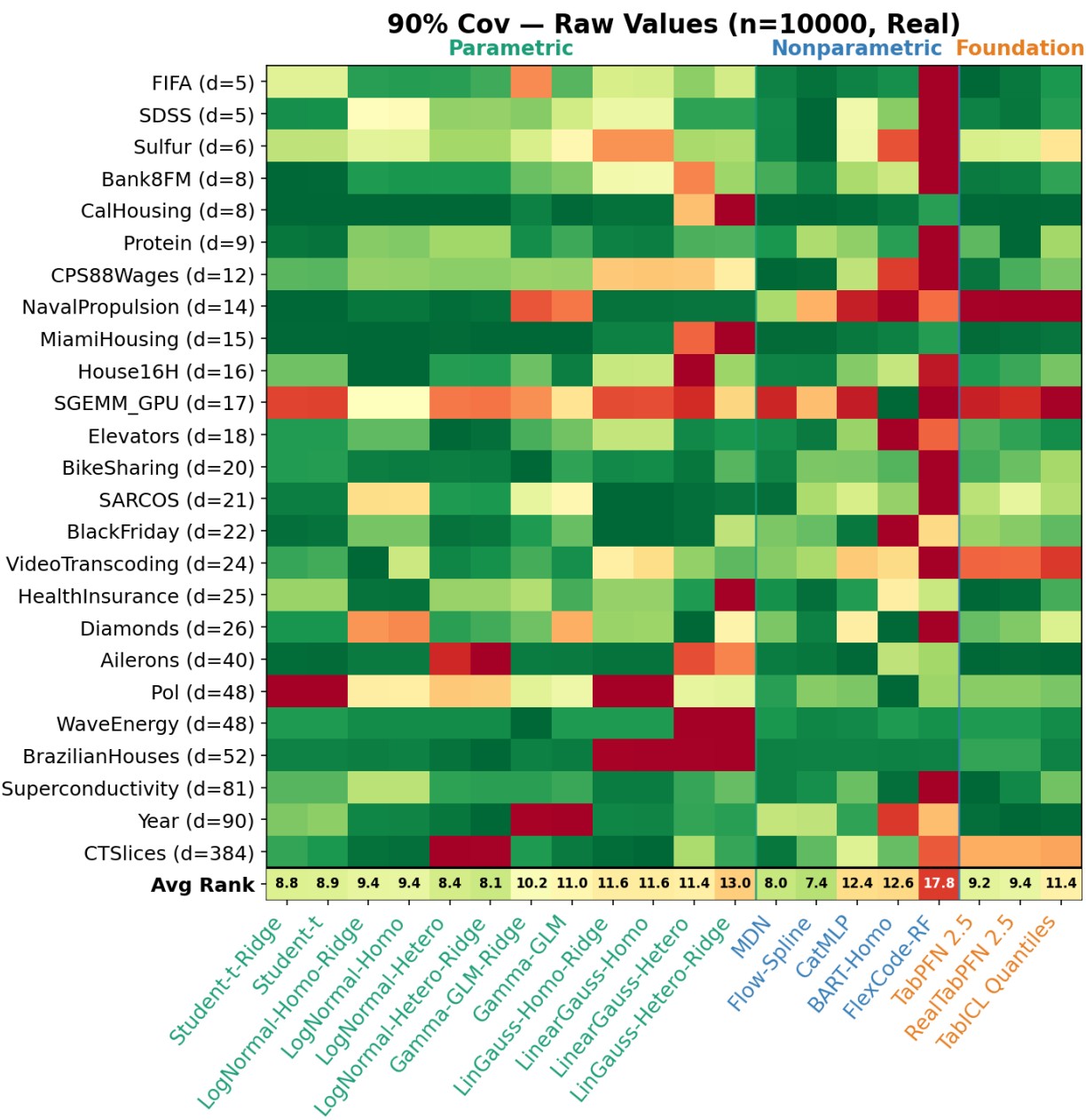

Figure 39: Raw 90% Coverage – $n = 10000$, real data.

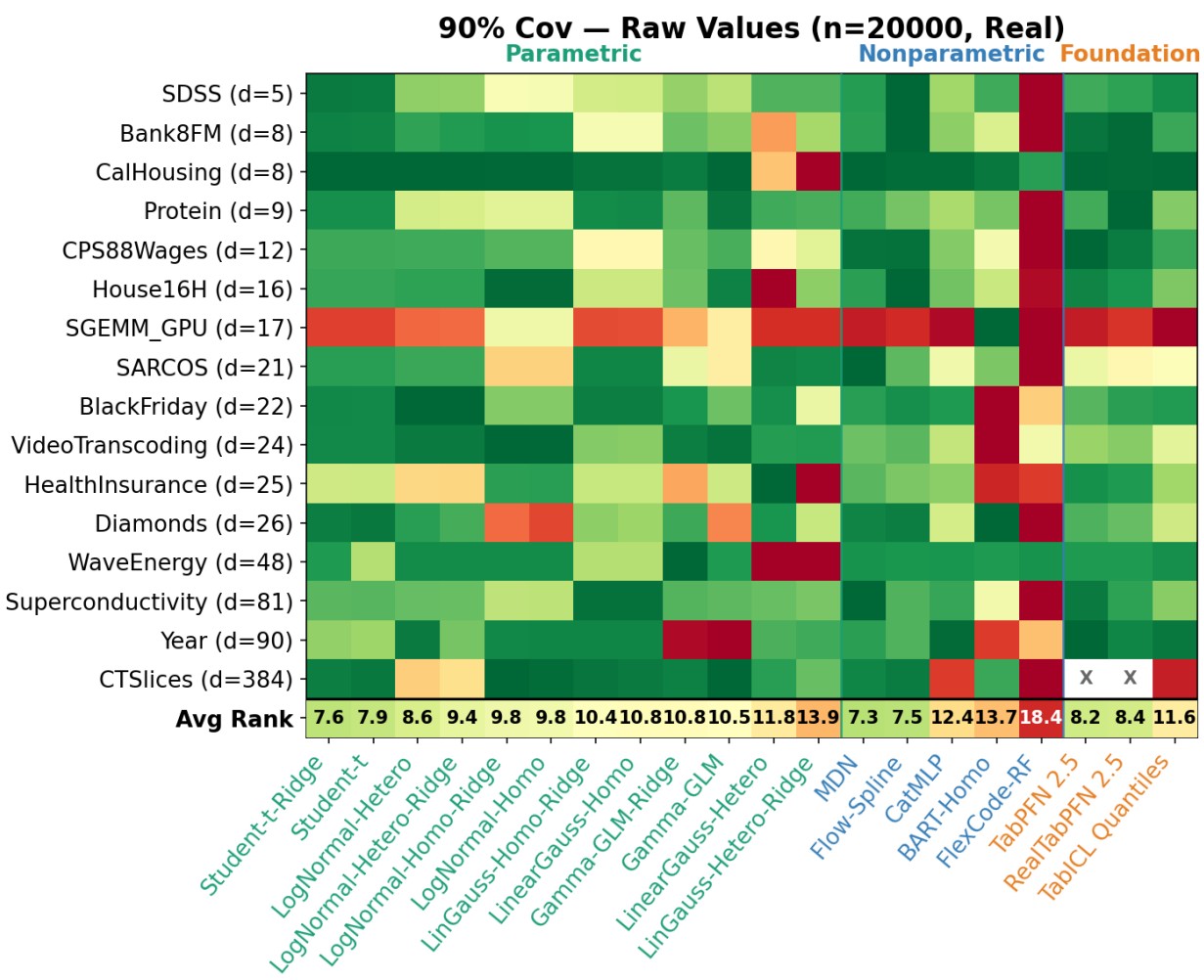

Figure 40: Raw 90% Coverage – $n = 20000$, real data.

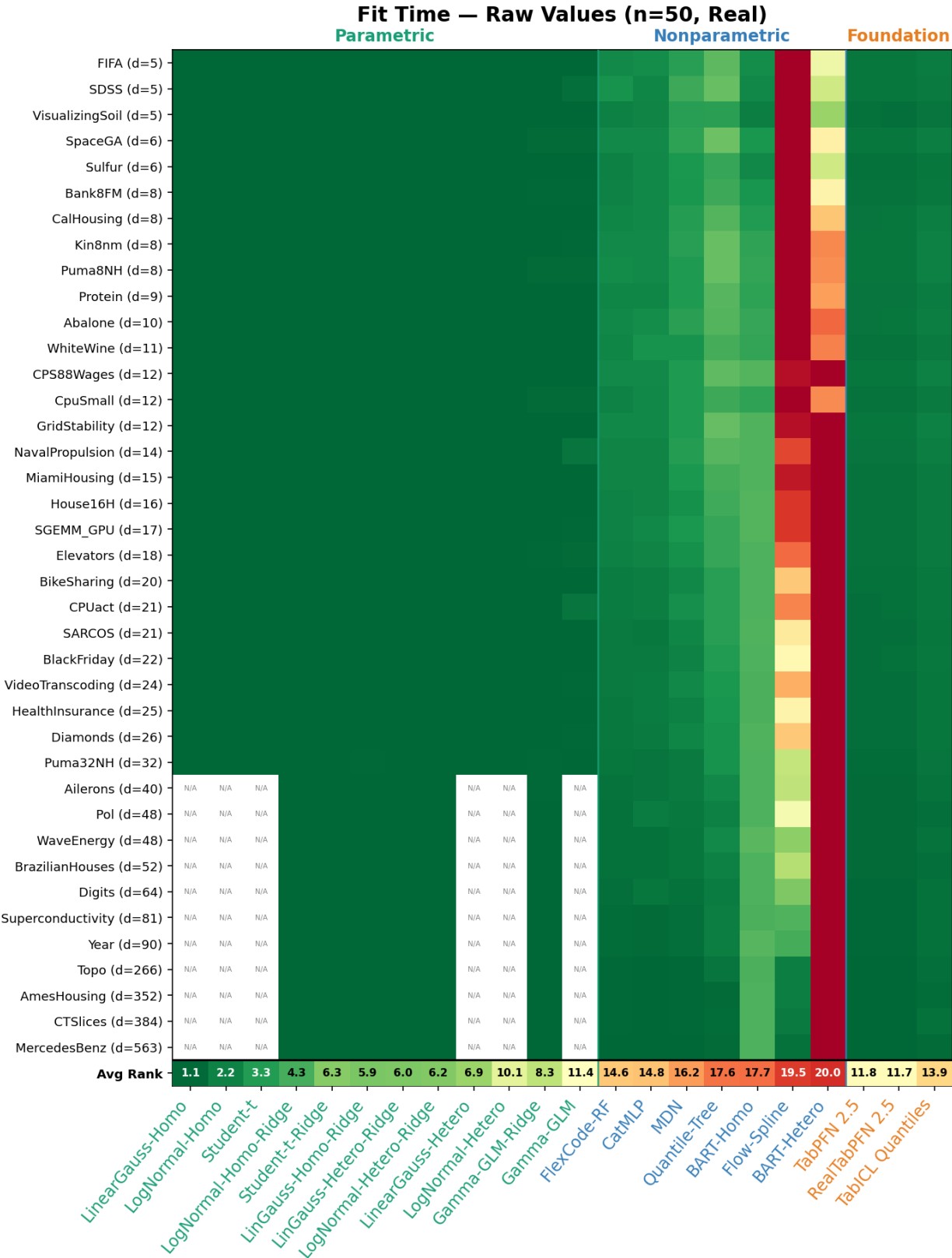

Figure 41: Raw Fit Time – $n = 50$, real data.

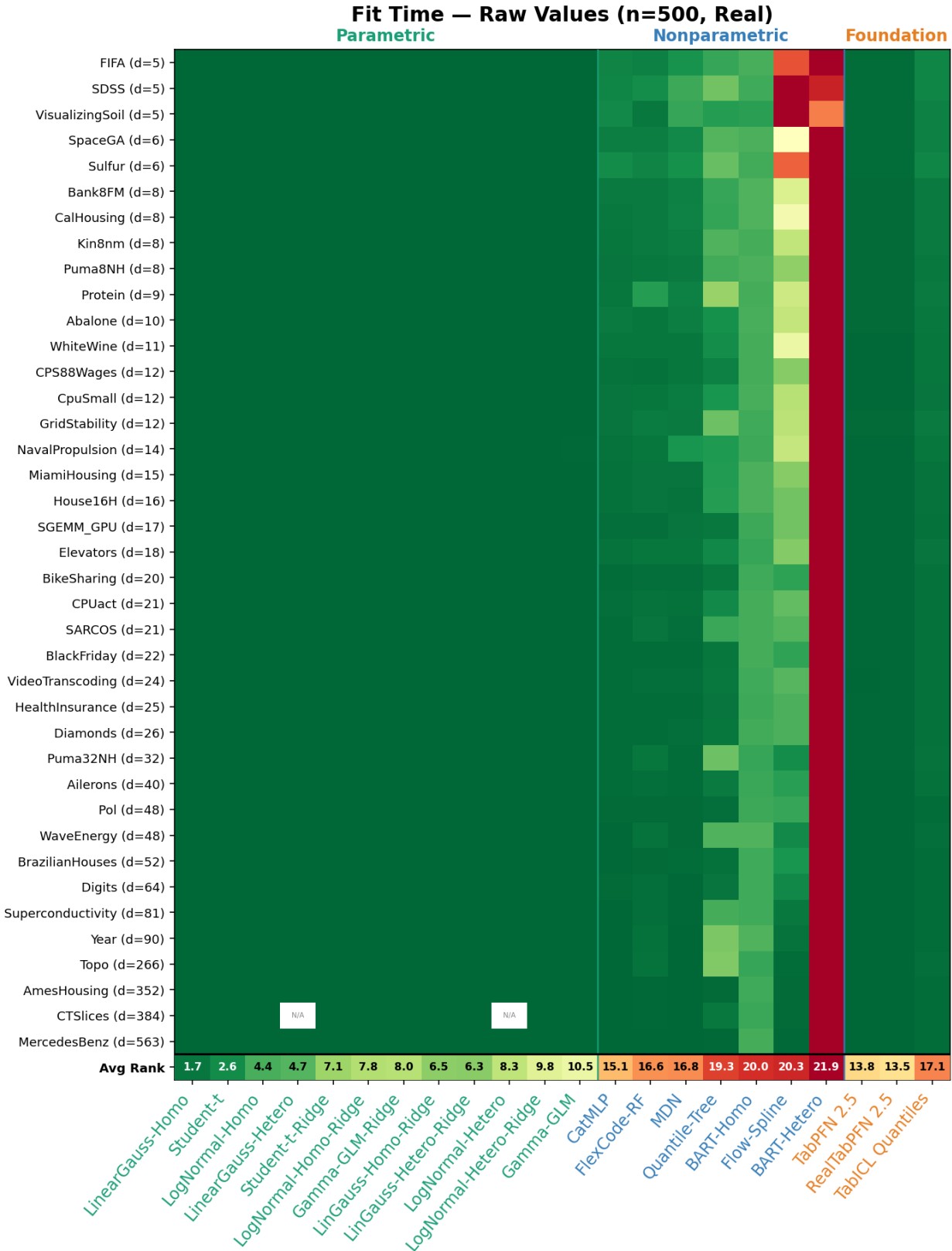

Figure 42: Raw Fit Time − $n = 500$, real data.

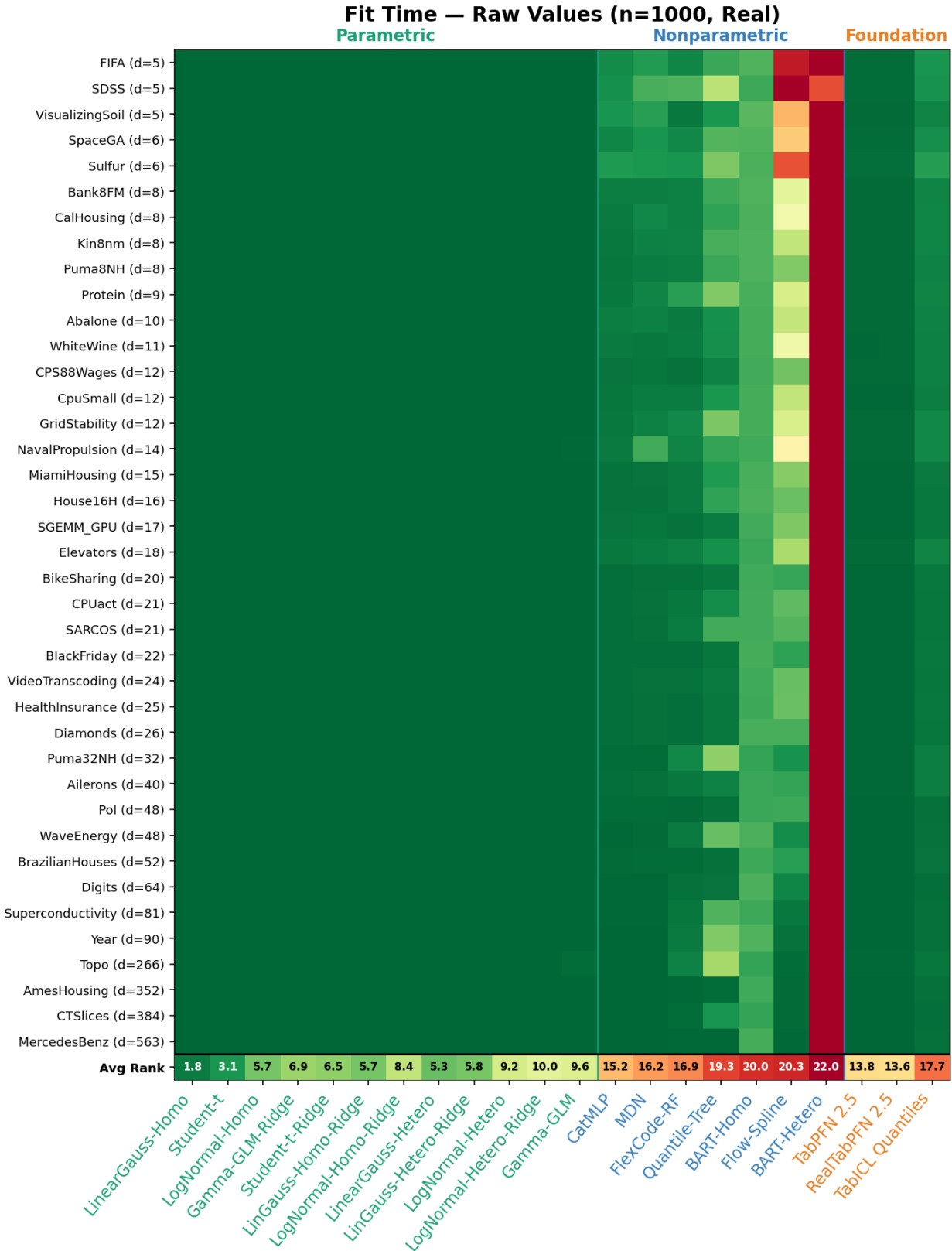

Figure 43: Raw Fit Time – $n = 1000$, real data.

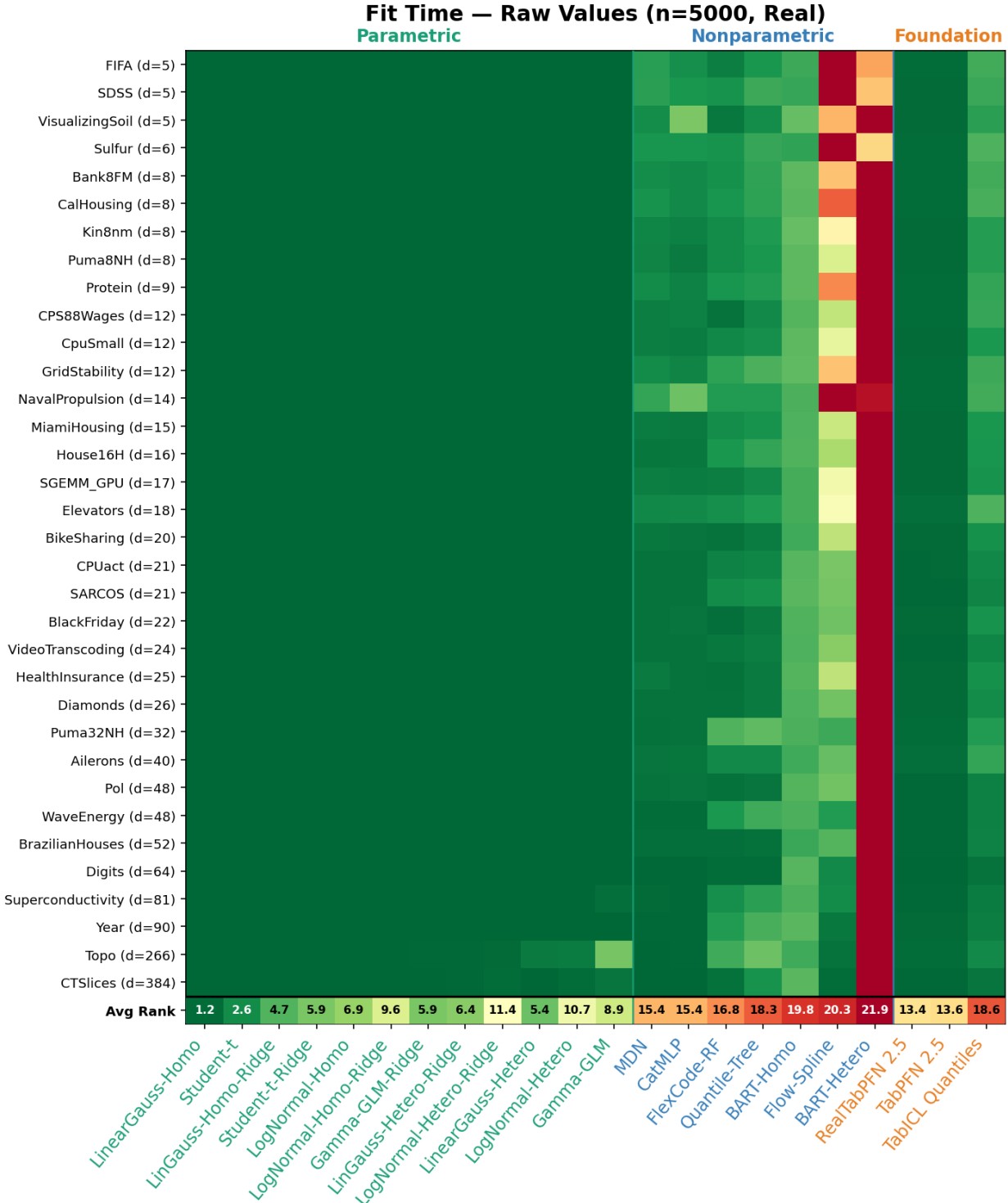

Figure 44: Raw Fit Time – $n = 5000$, real data.

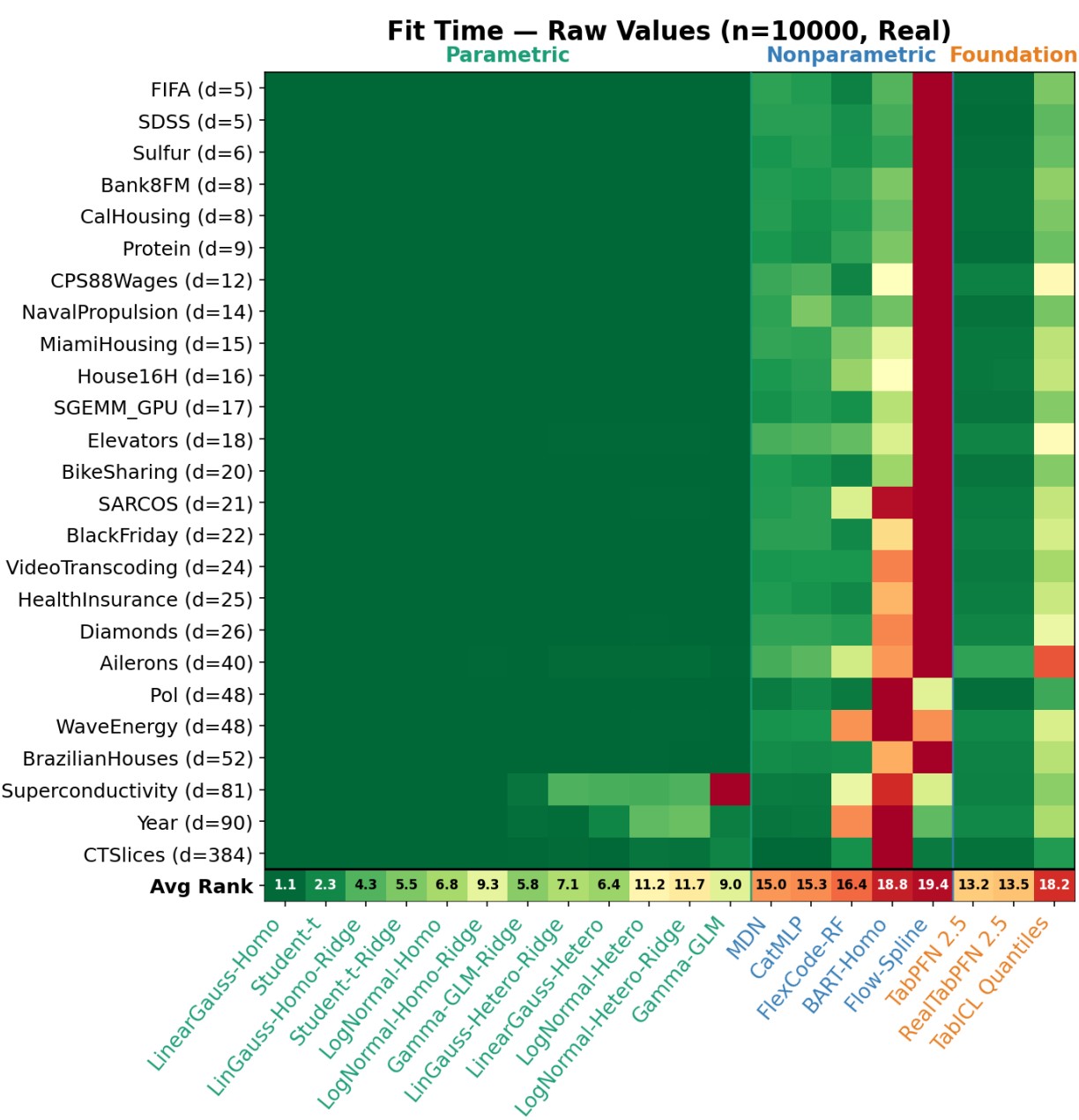

Figure 45: Raw Fit Time – $n = 10000$, real data.

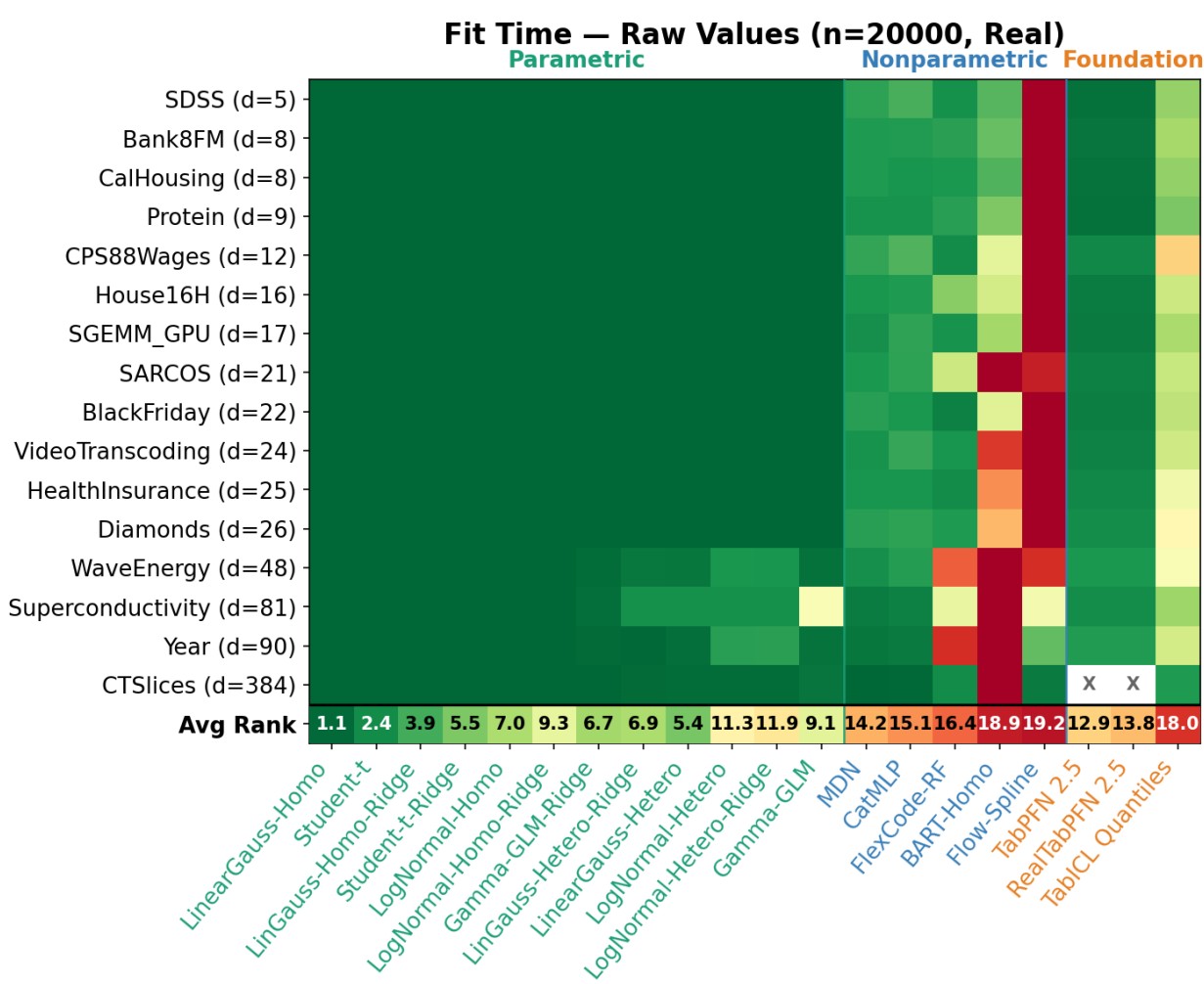

Figure 46: Raw Fit Time – $n = 20000$, real data.

## C.2   SDSS

This section shows all results for the SDSS experiment.

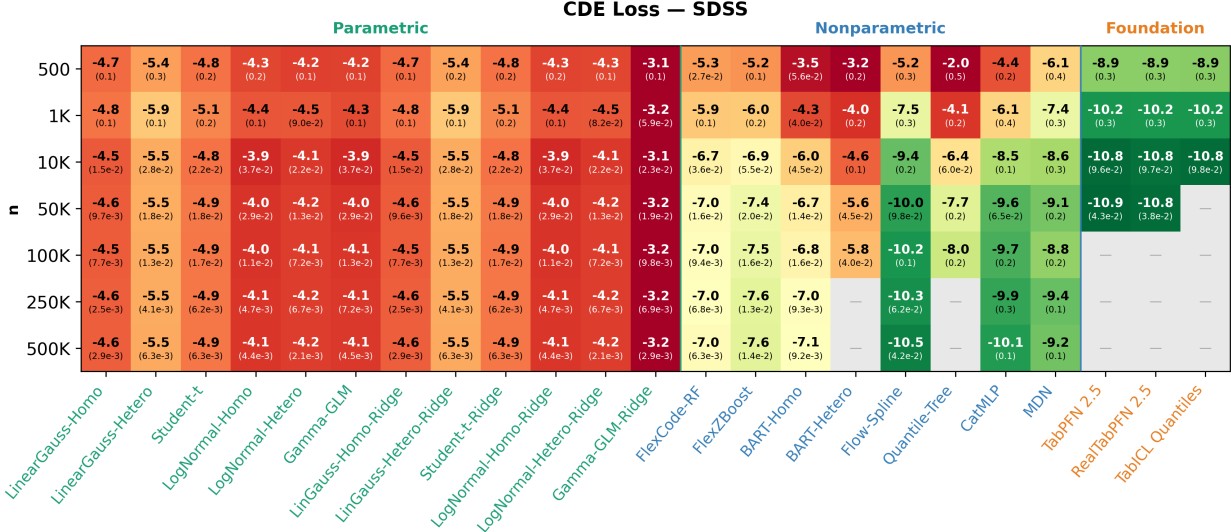

Figure 47: Raw CDE Loss – SDSS, real data.

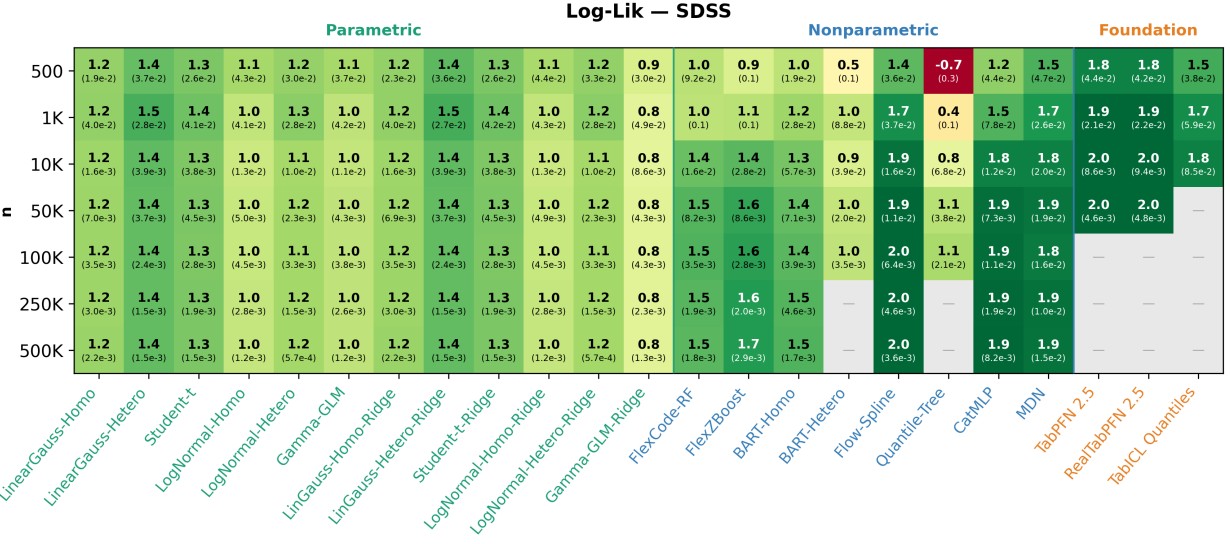

Figure 48: Raw Log-Likelihood – SDSS, real data.

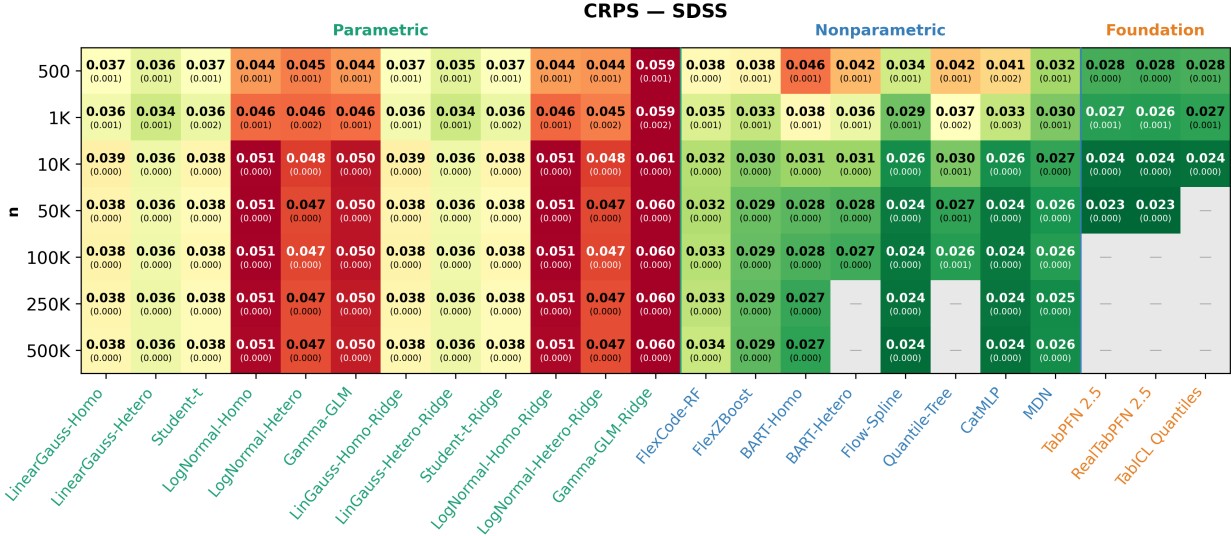

Figure 49: Raw CRPS – SDSS, real data.

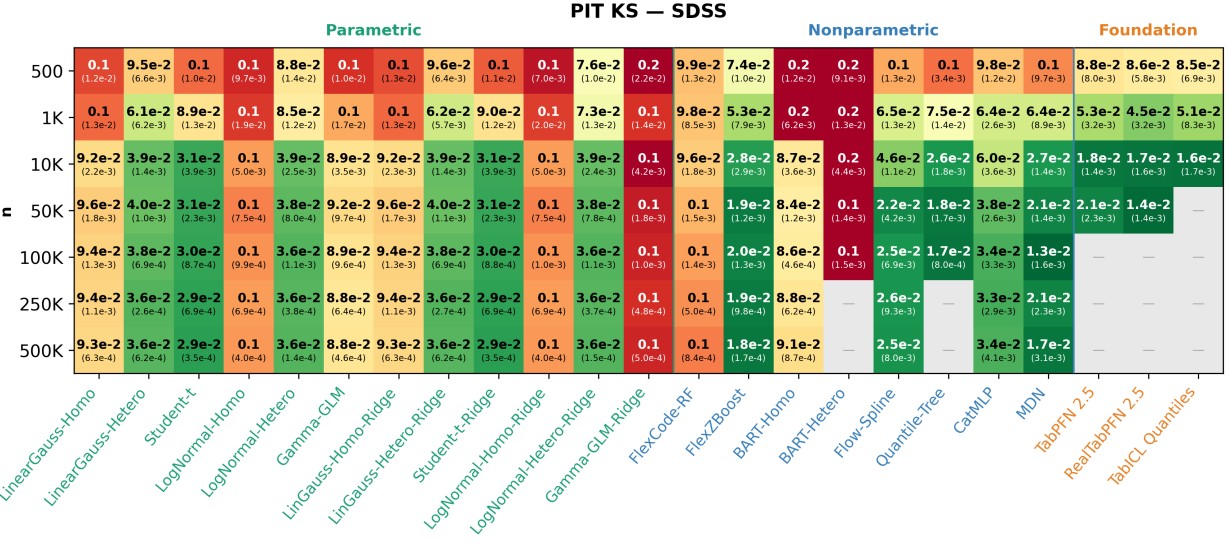

Figure 50: Raw PIT KS – SDSS, real data.

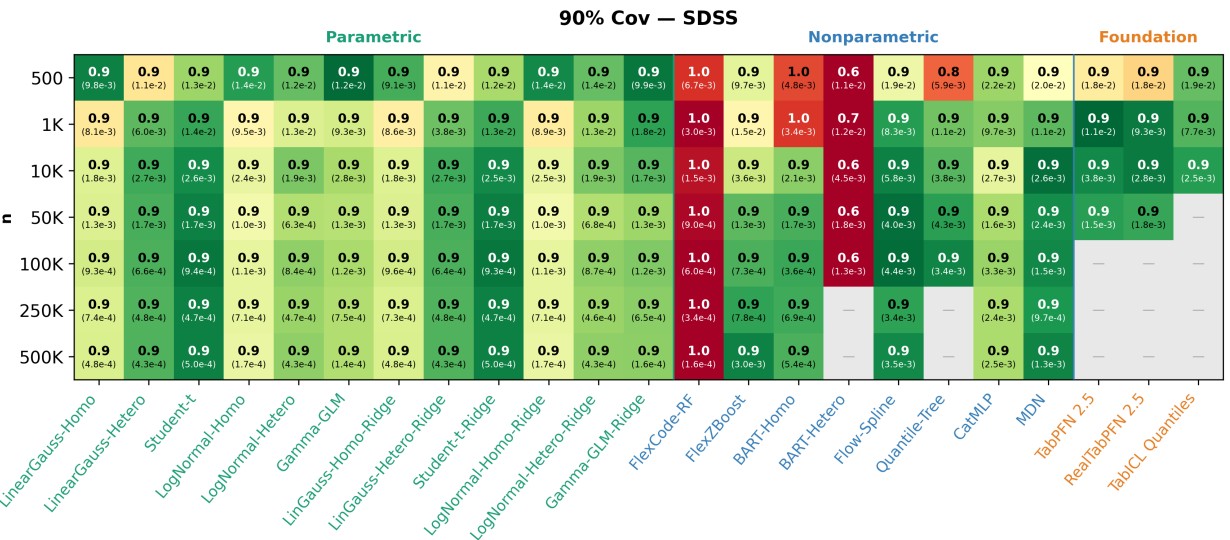

Figure 51: Raw 90% Coverage – SDSS, real data.

