# OpenReview forum: "Benchmarking Tabular Foundation Models for Conditional Density Estimation in Regression"
_TMLR — Under review for TMLR_

### Review · Reviewer_xYSL · 2026-06-05

**Summary Of Contributions:**

This paper benchmarks tabular foundation models (TFMs) like TabPFN-2.5, RealTabPFN-2.5 and TabICL under the Conditional Density Estimation (CDE) framework, testing their ability to predict full probability distributions (which is important e.g. in case of multimodality) rather than simple point estimates. Evaluating across 39 datasets with sample sizes up to 20,000, the authors compare TFMs against established CDE baselines using various metrics. The results reveal that TFMs regularly outperform other baselines.

The paper uses similar to Landsgesell et al. (arxiv 2603.08206, 9th March 2026) that tabular foundation models output predictive distributions. Like Landsgesell et al. the authors identify that most current benchmarks for tabular data only evaluate point forecasts and provide a timely call to action "We hope this benchmark helps shift the evaluation of tabular learning beyond point prediction
and toward a fuller assessment of distributional quality"

**Additional Comments:**

No additional comments.

"training-set sizes" vs "training set sizes", both is used

**Audience:**

Yes

**Audience Explanation:**

Beyond the specific benchmark results, this paper provides a highly timely and necessary call to action that mirrors concurrent movements in the 2026 tabular ML community. For a long time, tabular foundation models (like TabPFN) have been thoroughly evaluated on point predictions, while their intrinsic ability to output entire predictive distributions has been mainly neglected.

By formalizing this evaluation through the lens of Conditional Density Estimation (CDE), this paper aligns with concurrent frameworks like ScoringBench (https://arxiv.org/abs/2603.29928) and its foundation (https://arxiv.org/abs/2603.08206), which advocate for evaluating probabilistic forecasts through rigorous scoring rules rather than point estimates alone. Moving the community toward treating tabular foundation models as native conditional density estimators is a major step forward.

**Broader Impact Concerns:**

No ethical concerns.

**Claims And Evidence:**

Yes

**Claims Explanation:**

Yes, the claims are supported by transparent empirical evidence:

Different TFMs and CDE baseline models are evaluated across 39 datasets. The evaluation spans different metrics, including PIT calibration, Cross Entropy, CRPS and CDE loss. The authors additionally investigate the influence of sample size.

The evaluation shows that TFMs are strong off-the-shelf approaches for density estimation and this aligns with the findings in https://arxiv.org/abs/2603.29928 and https://arxiv.org/abs/2603.08206 which also evaluates several gradient boosted methods.

**Requested Changes:**

Recommendation regarding concurrent work on evaluation metrics:
While the authors do an excellent job of contextualizing their work within the long tradition of Conditional Density Estimation (dating back to Rosenblatt, 1969), the paper would benefit from a brief update by acknowledging concurrent efforts in the 2026 landscape that focus on interpreting and benchmarking distributional regression via proper scoring rules. Specifically, it would be highly valuable for the reader to mention ScoringBench (https://arxiv.org/abs/2603.29928) in addition to its predating paper (https://arxiv.org/abs/2603.08206 9th March 2026). Similar to this paper, ScoringBench highlights the critical gap in evaluating tabular foundation models (such as TabPFN and TabICL) purely on point metrics, providing an open framework for evaluating them under a suite of proper scoring rules (CRPS, CRLS, energy score, etc.). Updates with regard to this concurrent track of work would paint a complete picture of the current state-of-the-art in probabilistic tabular evaluation.

Minor Suggestion:
When referencing the empirical setup of https://arxiv.org/abs/2603.08206, rephrasing the explicit statement of "20 OpenML datasets" to a more flexible descriptor like "several OpenML datasets" might be more timely, as it is conceivable that this benchmark setup could expand its dataset coverage.

---

> ### Author Response · Authors · 2026-07-02
>
> We thank the referee for his/her comments. Here are our point-by-point responses:
>
> > Recommendation regarding concurrent work on evaluation metrics: While the authors do an excellent job of contextualizing their work within the long tradition of Conditional Density Estimation (dating back to Rosenblatt, 1969), the paper would benefit from a brief update by acknowledging concurrent efforts in the 2026 landscape that focus on interpreting and benchmarking distributional regression via proper scoring rules. Specifically, it would be highly valuable for the reader to mention ScoringBench (https://arxiv.org/abs/2603.29928) in addition to its predating paper (https://arxiv.org/abs/2603.08206 9th March 2026). Similar to this paper, ScoringBench highlights the critical gap in evaluating tabular foundation models (such as TabPFN and TabICL) purely on point metrics, providing an open framework for evaluating them under a suite of proper scoring rules (CRPS, CRLS, energy score, etc.). Updates with regard to this concurrent track of work would paint a complete picture of the current state-of-the-art in probabilistic tabular evaluation.
>
> We thank the reviewer for pointing us to this concurrent line of work. We have revised the related-work discussion to mention ScoringBench, together with the earlier proper-scoring-rule paper on distributional regression with tabular foundation models. Indeed, these papers are closely aligned with our motivation: they emphasize that models such as TabPFN and TabICL produce full predictive distributions and should therefore be evaluated with distributional metrics, not only point-prediction metrics. We now clarify that our work is complementary: ScoringBench provides an open, dynamic, proper-scoring-rule benchmark for tabular foundation models, whereas our paper focuses specifically on comparing foundation models against a broad set of purpose-built CDE baselines across several sample sizes and dimensions, and a large SDSS case study.
>
> > Minor Suggestion: When referencing the empirical setup of https://arxiv.org/abs/2603.08206, rephrasing the explicit statement of "20 OpenML datasets" to a more flexible descriptor like "several OpenML datasets" might be more timely, as it is conceivable that this benchmark setup could expand its dataset coverage.
>
> Thanks!

---

### Review · Reviewer_x8k3 · 2026-06-08

**Summary Of Contributions:**

This paper addresses the setting of conditional density estimation given tabular covariates. While the current work focuses on evaluating point-wise estimation, it falls short of the demands of producing predictive conditional distributions. This work proposes a large-scale empirical analysis of the nascent group of tabular foundation models, which can solve the problem.
The analysis offers variety in a set of tasks (39 datasets, ranging from five to 563 covariates, fifty to 563 samples), in a number of commonly-known metrics (CRPS, CDE loss, log-likelihood amongst others), parametric and non-parametric baselines.
One of the important takeaway messages is that while the foundation models perform strongly in terms of the proper scoring rules, the results are much more modest for the calibration results, and they can even be outperformed by the non-foundation models which encode a strong inductive bias.

**Additional Comments:**

I think the paper is well fitting in terms of the scope of TMLR, proposing work that is not necessarily providing methodological or theoretical novelty but rigorous empirical analysis of the existing works of interest to the community.

**Audience:**

Yes

**Audience Explanation:**

The target audience will be interested in conditional density estimation problem, which has not been covered enough yet, including with empirical evaluations.

**Broader Impact Concerns:**

No broader impact concerns: the paper does not propose new methods, and I believe that the ethical implication of this work are sufficiently addressed.

**Claims And Evidence:**

Yes

**Claims Explanation:**

I think the claims are correct and the authors did exactly what they claimed, accomplishing what is described in the summary of contributions.

**Requested Changes:**

1) Clarity: I would also highlight the foundation models in the Results section in bold and/or in colour to make the reading easier.
2) Clarity: in the conclusion, I would propose to group the outcomes by a bullet list of takeaway messages, outlining the main findings of this benchmarking in a way that gives a tl;dr caption and the description of the findings.  Now, the text is a bit hard to follow as it is unstructured.
3) audience: I think to make it even more relevant to the audience, I would add the takeaway messages (best practice) for the practitioners how they could apply the foundation models for the conditional density estimation.
4) finally, it would be an improvement if the authors could include different calibration strategies to show how the choice of calibration could improve the performance of foundation models and of the rest, and whether the same calibration strategies would lead to difference in outcome improvement.

---

> ### Author Response · Authors · 2026-07-02
> **Part 1**
>
> We thank the referee for the useful comments that helped us improved the paper. Here are our responses:
>
> > Clarity: I would also highlight the foundation models in the Results section in bold and/or in colour to make the reading easier.
>
> Great suggestion! Added.
>
> > Clarity: in the conclusion, I would propose to group the outcomes by a bullet list of takeaway messages, outlining the main findings of this benchmarking in a way that gives a tl;dr caption and the description of the findings. Now, the text is a bit hard to follow as it is unstructured.
>
> We thank the reviewer for the helpful suggestion. We agree that the conclusion was hard to follow in its previous form. We have revised it following the referee’s advice, with the following takeaways:
>
>
> - [Tabular foundation models are strong off-the-shelf CDE methods.]
>   Across 39 real-world datasets, training sizes from 50 to 20,000, and six
>   complementary evaluation metrics, the foundation-model variants were the
>   strongest overall performers for density estimation. In particular, the
>    TabPFN variants achieved the best results on CDE loss,
>   log-likelihood, and CRPS on most datasets, while
>   TabICL-Quantiles also ranked among the top-performing
>   methods on most metrics.
>
>  - [Their main empirical advantage is sample efficiency.]
>   Foundation models were already competitive at very small sample sizes and
>   often maintained their advantage as $n$ increased. The SDSS photometric
>   redshift case study illustrates this pattern most clearly:
>   TabPFN trained on 50,000 galaxies outperformed all
>   competing methods trained on the full 500,000-example dataset, indicating
>   that these models can extract substantially more distributional information
>   per labeled example.
>
> - [Calibration is competitive but not uniformly best.]
>   The calibration results are more nuanced than the proper-scoring-rule
>   results. Foundation models performed especially well at small sample sizes,
>   but at larger $n$ they were sometimes overtaken by task-specific neural
>   baselines such as MDN. Post-hoc PIT recalibration substantially
>   improved their absolute calibration while preserving their advantage on CDE
>   loss, log-likelihood, and CRPS.
>
> - [The methods remain constrained by context length and memory.]
>   The TabPFN variants are limited by the amount of data that
>   can be processed in context and typically require GPU resources at larger
>   sample sizes. In our experiments, the combination of large $n$ and high
>   covariate dimension led to out-of-memory failures on CTSlices, showing that
>   direct application to very large or high-dimensional datasets may require
>   subsampling, batching, or other approximation strategies.
>
> - [Specialized models can still win when their inductive bias
>   matches the response structure.]
>   Although such cases were the exception rather than the rule, diagnostic
>   examples showed that non-foundation methods can outperform foundation models
>   for discrete or quasi-discrete responses, or when a simple parametric family
>   provides a particularly well-matched regularizing assumption.
>
> - [Practical recommendation: calibrate, and check computational
>   feasibility.]
>   For practitioners, the results suggest using tabular foundation models as
>   strong default conditional density estimators when the dataset fits within
>   the model's context and memory limits. In applied workflows, we recommend
>   evaluating calibration diagnostics and applying post-hoc PIT recalibration
>   when calibrated uncertainty is important. Users should also account for the
>   joint effect of sample size $n$, covariate dimension $d$, available memory,
>   and total fit-and-predict time, since these factors can determine whether a
>   foundation model is practical on a given problem. This is an active research
>   area, and the computational trade-offs should be revisited as newer tabular
>   foundation models become faster and more memory efficient.
>
>
> > audience: I think to make it even more relevant to the audience, I would add the takeaway messages (best practice) for the practitioners how they could apply the foundation models for the conditional density estimation
>
> We agree that the conclusion should make the practical implications clearer for readers who want to use these models. We now added a bullet to the conclusions that mentions practical recommendations.

---

> ### Author Response · Authors · 2026-07-02
> **Part 2**
>
> > finally, it would be an improvement if the authors could include different calibration strategies to show how the choice of calibration could improve the performance of foundation models and of the rest, and whether the same calibration strategies would lead to difference in outcome improvement.
>
> We thank the reviewer for the suggestion. We have added a new post-hoc recalibration analysis to address this point. In this section, we apply the same distribution-free PIT recalibration procedure to all methods, including both foundation models and baselines, and compare performance before and after recalibration. The results show that recalibration improves absolute calibration for nearly all methods, but not to the same degree. The largest gains occur for methods that were initially most miscalibrated, especially several baselines; foundation models also improve substantially in absolute calibration. Importantly, recalibration improves the calibration of foundation models without removing their advantage on CDE loss, log-likelihood, and CRPS.

---

> > ### Comment · Reviewer_x8k3 · 2026-07-17
> >
> > I would like to thank the authors on the amendments. I think it is an improvement to the paper, and my scores have not changed as a result.

---

### Review · Reviewer_RBQ6 · 2026-06-28

**Summary Of Contributions:**

This paper presents a set of empirical benchmarks for tabular foundation models in the context of conditional density estimation (CDE). This is useful as (i) tabular foundation models are not yet systematically used for CDEs, (ii) most benchmarks in the literature - for TFNs or not - are point estimate benchmarks, not CDEs.

**Additional Comments:**

none

**Audience:**

Yes

**Audience Explanation:**

Yes, I personally would certainly read this paper with interest. The applications of tabular foundation models are not yet evident to a significant cross-section of the community and papers like this are very useful for underscoring their utility in a demonstrated manner.

**Broader Impact Concerns:**

I have no broader impact concerns apart from the critical correction listed above: when real scientific data are used, it is very important to correctly acknowledge the originating work of the data and the selection criteria employed in obtaining a particular subset of those data. Not only is it required for fully understanding the implications of the benchmark, but it is also an acknowledgement of the time, expertise and effort that go into producing those data in the first place.

**Claims And Evidence:**

Yes

**Claims Explanation:**

There are a variety of empirical benchmarks reported. I have been able to approximately* reproduce the SDSS benchmark myself.

*see comments below on data selection.

**Requested Changes:**

[strengthen] Figure 1 caption states that all 3 rows are bimodal distributions, is row 2 bimodal?

[strengthen] why a KS test for calibration? There are a range of methods for making this comparison and it's not immediately clear why a KS test should be preferred in this case, or indeed why a common calibration test should be equally appropriate across all of the benchmarks in this work given their heterogeneous nature. the result that the calibration of the tabular models is dependent on sample size is particularly interesting and it would be good to have the choice of metric better justified. I assume that this behaviour is not replicated across other models? It would be good to have this this explicitly commented upon, because it is very difficult to infer this behaviour from the figures in the appendix.

[critical] the numbers on the SDSS dataset need clarifying. SDSS DR18 contains millions of galaxies, not the 500k stated in the paper. Is this a subset associated with a kaggle challenge? If so it should be stated. Also the citation is incorrect, pointing to a webpage rather than the survey paper. It is important when using real scientific data to properly acknowledge the originating source and to be explicit in the data selection when using a subset of a larger data volume.

---

> ### Author Response · Authors · 2026-07-02
>
> We thank the referee for the useful comments. Here are our responses:
>
> > Figure 1 caption states that all 3 rows are bimodal distributions, is row 2 bimodal?
>
> Good catch. In Figure 1, the DGP is a two-component Gaussian mixture, but not every conditional density shown is necessarily bimodal; Instance 2 is unimodal because the two components highly overlap. We have changed the text and caption to refer to a “two-component Gaussian-mixture DGP” rather than claiming that all rows are bimodal.
>
> > why a KS test for calibration? There are a range of methods for making this comparison and it's not immediately clear why a KS test should be preferred in this case, or indeed why a common calibration test should be equally appropriate across all of the benchmarks in this work given their heterogeneous nature. the result that the calibration of the tabular models is dependent on sample size is particularly interesting and it would be good to have the choice of metric better justified. I assume that this behaviour is not replicated across other models? It would be good to have this this explicitly commented upon, because it is very difficult to infer this behaviour from the figures in the appendix.
>
> We chose the PIT KS statistic because it is a scale-free marginal calibration diagnostic and a standard choice in several existing comparisons of CDE methods. We now cite Dalmasso et al. (2020), Schmidt et al. (2020), and Zhao et al. (2021) as examples where this is done. We also clarify that 90% coverage is an additional calibration diagnostic, so our calibration assessment does not rely only on PIT KS. Finally, we expanded the discussion to note that the sample-size dependence of calibration is not uniform across methods: for example, MDN remains among the best-calibrated methods at larger sample sizes, whereas foundation models improve more strongly on density accuracy than on calibration.
>
> > the numbers on the SDSS dataset need clarifying. SDSS DR18 contains millions of galaxies, not the 500k stated in the paper. Is this a subset associated with a kaggle challenge? If so it should be stated. Also the citation is incorrect, pointing to a webpage rather than the survey paper. It is important when using real scientific data to properly acknowledge the originating source.
>
> Thanks for catching this. The 500,000 objects were obtained via a direct SQL query to the DR18 SkyServer \citep{sdss_dr18}, randomly selecting $500{,}000$ galaxies with reliable spectroscopic redshifts ($\texttt{zWarning}=0$, $0<z<1$) and clean $ugriz$ photometry in the range $14\le r\le 22$. We added this to the paper and updated the reference.

---

### Comment · Reviewer_RBQ6 · 2026-06-19

This paper presents a set of empirical benchmarks for tabular foundation models in the context of conditional density estimation (CDE). This is useful as (i) tabular foundation models are not yet systematically used for CDEs, (ii) most benchmarks in the literature - for TFNs or not - are point estimate benchmarks, not CDEs.

- Figure 1 caption states that all 3 rows are bimodal distributions, is row 2 bimodal?
- why a KS test for calibration? There are a range of methods for making this comparison and it's not immediately clear why a KS test should be preferred in this case, or indeed why a common calibration test should be equally appropriate across all of the benchmarks in this work given their heterogeneous nature. the result that the calibration of the tabular models is dependent on sample size is particularly interesting and it would be good to have the choice of metric better justified. I assume that this behaviour is not replicated across other models? It would be good to have this this explicitly commented upon, because it is very difficult to infer this behaviour from the figures in the appendix.
- the numbers on the SDSS dataset need clarifying. SDSS DR18 contains millions of galaxies, not the 500k stated in the paper. Is this a subset associated with a kaggle challenge? If so it should be stated. Also the citation is incorrect, pointing to a webpage rather than the survey paper. It is important when using real scientific data to properly acknowledge the originating source.